# $p$-LAPLACIAN BASED GRAPH NEURAL NETWORKS

## ABSTRACT

Graph neural networks (GNNs) have demonstrated superior performance for semi-supervised node classification on graphs, as a result of their ability to exploit node features and topological information simultaneously. However, most GNNs implicitly assume that the labels of nodes and their neighbors in a graph are the same or consistent, which does not hold in heterophilic graphs, where the labels of linked nodes are likely to differ. Moreover, when the topology is non-informative for label prediction, ordinary GNNs may work significantly worse than simply applying multi-layer perceptrons (MLPs) on each node. To tackle the above problem, we propose a new $p$-Laplacian based GNN model, termed as $^p$GNN, whose message passing mechanism is derived from a discrete regularization framework and could be theoretically explained as an approximation of a polynomial graph filter defined on the spectral domain of $p$-Laplacians. The spectral analysis shows that the new message passing mechanism works simultaneously as low-pass and high-pass filters, thus making $^p$GNNs are effective on both homophilic and heterophilic graphs. Empirical studies on real-world and synthetic datasets validate our findings and demonstrate that $^p$GNNs significantly outperform several state-of-the-art GNN architectures on heterophilic benchmarks while achieving competitive performance on homophilic benchmarks. Moreover, $^p$GNNs can adaptively learn aggregation weights and are robust to noisy edges.

## 1 INTRODUCTION

In this paper, we explore the usage of graph neural networks (GNNs) for semi-supervised node classification on graphs, especially when the graphs admit strong heterophily or noisy edges. Semi-supervised learning problems on graphs are ubiquitous in a lot of real-world scenarios, such as user classification in social media (Kipf & Welling, 2017), protein classification in biology (Velickovic et al., 2018), molecular property prediction in chemistry (Duvenaud et al., 2015), and many others (Marcheggiani & Titov, 2017; Satorras & Estrach, 2018). Recently, GNNs are becoming the de facto choice for processing graph structured data. They can exploit the node features and the graph topology by propagating and transforming the features over the topology in each layer and thereby learn refined node representations. A series of GNN architectures have been proposed, including graph convolutional networks (Bruna et al., 2014; Henaff et al., 2015; Defferrard et al., 2016; Kipf & Welling, 2017; Wu et al., 2019), graph attention networks (Velickovic et al., 2018; Thekumparampil et al., 2018), and other representatives (Hamilton et al., 2017; Xu et al., 2018; Pei et al., 2020).

Most of the existing GNN architectures work under the homophily assumption, i.e. the labels of nodes and their neighbors in a graph are the same or consistent, which is also commonly used in graph clustering (Bach & Jordan, 2004; von Luxburg, 2007; Liu & Han, 2013) and semi-supervised learning on graphs (Belkin et al., 2004; Hein, 2006; Nadler et al., 2009). However, recent studies (Zhu et al., 2020; 2021; Chien et al., 2021) show that in contrast to their success on homophilic graphs, most GNNs fail to work well on heterophilic graphs, in which linked nodes are more likely to have distinct labels. Moreover, GNNs could even fail on graphs where their topology is not helpful for label prediction. In these cases, propagating and transforming node features over the graph topology could lead to worse performance than simply applying multi-layer perceptrons (MLPs) on each of the nodes independently. Several recent works were proposed to deal with the heterophily issues of GNNs. Zhu et al. (2020) finds that heuristically combining ego-, neighbor, and higher-order embeddings improves GNN performance on heterophilic graphs. Zhu et al. (2021) uses a compatibility matrix to model the graph homophily or heterphily level. Chien et al. (2021) incorporates

the generalized PageRank algorithm with graph convolutions so as to jointly optimize node feature and topological information extraction for both homophilic and heterophilic graphs. However, the problem of GNNs on graphs with non-informative topologies (or noisy edges) remains open.

Unlike previous works, we tackle the above issues of GNNs by proposing the discrete $p$-Laplacian based message passing scheme, termed as $p$-Laplacian message passing. It is derived from a discrete regularization framework and is theoretically verified as an approximation of a polynomial graph filter defined on the spectral domain of the $p$-Laplacian. Spectral analysis of $p$-Laplacian message passing shows that it works simultaneously as low-pass and high-pass filters [1] and thus is applicable to both homophilic and heterophilic graphs. Moreover, when $p \neq 2$, our theoretical results indicate that it can adaptively learn aggregation weights in terms of the variation of node embeddings on edges (measured by the graph gradient (Amghibech, 2003; Zhou & Schölkopf, 2005; Luo et al., 2010)), and work as low-pass or both low-pass and high-pass filters on a node according to the local variation of node embeddings around the node (measured by the norm of graph gradients).

Based on $p$-Laplacian message passing, we propose a new GNN architecture, called $^p$GNN, to enable GNNs to work with heterophilic graphs and graphs with non-informative topologies. Several existing GNN architectures, including SGC (Wu et al., 2019), APPNP (Klicpera et al., 2019) and GPRGNN (Chien et al., 2021), can be shown to be analogical to $^p$GNN with $p = 2$. Our empirical studies on real-world benchmark datasets (homophilic and heterophilic datasets) and synthetic datasets (cSBM (Deshpande et al., 2018)) demonstrate that $^p$GNNs obtain the best performance on heterophilic graphs and competitive performance on homophilic graphs against state-of-the-art GNNs. Moreover, experimental results on graphs with different levels of noisy edges show that $^p$GNNs work much more robustly than GNN baselines and even as well as MLPs on graphs with completely random edges. Additional experiments (reported in Appendix F.5) illustrate that intergrating $^p$GNNs with existing GNN architectures (i.e. GCN (Kipf & Welling, 2017), JKNet (Xu et al., 2018)) can significantly improve their performance on heterophilic graphs. In conclusion, our contributions can be summarized as below:

(1) **New methodologies.** We propose $p$-Laplacian message passing and $^p$GNN to adapt GNNs to heterophilic graphs and graphs where the topology is non-informative for label prediction. (2) **Superior performance.** We empirically demonstrate that $^p$GNNs is superior on heterophilic graphs and competitive on homophilic graphs against state-of-the-art GNNs. Moreover, $^p$GNNs work robustly on graphs with noisy edges or non-informative topologies. (3) **Theoretical justification.** We theoretically demonstrate that $p$-Laplacian message passing works as both low-pass and high-pass filters and the message passing iteration is guarantee to converge with proper settings. (4) **New paradigm of designing GNN architectures.** We bridge the gap between discrete regularization framework and GNNs, which could further inspire researchers to develop new graph convolutions or message passing schemes using other regularization techniques with explicit assumptions on graphs. Due to space limit, we defer the discussions on related work and future work and all proofs to the Appendix.

## 2 PRELIMINARIES AND BACKGROUND

**Notation.** Let $\mathcal{G} = (\mathcal{V}, \mathcal{E}, \mathbf{W})$ be an undirected graph, where $\mathcal{V} = \{1, 2, \ldots, N\}$ is the set of nodes, $\mathcal{E} \subseteq \mathcal{V} \times \mathcal{V}$ is the set of edges, $\mathbf{W} \in \mathbb{R}^{N \times N}$ is the adjacency matrix and $W_{i,j} = W_{j,i}, W_{i,j} > 0$ for $[i, j] \in \mathcal{E}$, $W_{i,j} = 0$, otherwise. $\mathcal{N}_i = \{j\}_{[i,j] \in \mathcal{E}}$ denotes the set of neighbors of node $i$, $\mathbf{D} \in \mathbb{R}^{N \times N} = \mathrm{diag}(D_{1,1}, \ldots D_{N,N})$ denotes the diagonal degree matrix with $D_{i,i} = \sum_{j=1}^{N} W_{i,j}$, for $i = 1, \ldots, N$. $f : \mathcal{V} \to \mathbb{R}$ and $g : \mathcal{E} \to \mathbb{R}$ are functions defined on the vertices and edges of $\mathcal{G}$, respectively. $\mathcal{F}_\mathcal{V}$ denotes the Hilbert space of functions endowed with the inner product $\langle f, \tilde{f} \rangle_{\mathcal{F}_\mathcal{V}} := \sum_{i \in \mathcal{V}} f(i)\tilde{f}(i)$. Similarly define $\mathcal{F}_\mathcal{E}$. We also denote by $[K] = \{1, 2, \ldots, K\}, \forall K \in \mathbb{N}$ and we use $\|\mathbf{x}\| = \|\mathbf{x}\|_2 = (\sum_{i=1}^{d} x_i^2)^{1/2}, \forall \mathbf{x} \in \mathbb{R}^d$ to denote the Frobenius norm of a vector.

**Problem Formulation.** Given a graph $\mathcal{G} = (\mathcal{V}, \mathcal{E}, \mathbf{W})$, each node $i \in \mathcal{V}$ has a feature vector $\mathbf{X}_{i,:}$ which is the $i$-th row of $\mathbf{X}$ and a subset of nodes in $\mathcal{G}$ have labels from a label set $\mathcal{L} = \{1, \ldots, L\}$. The goal of semi-supervised node classification on $\mathcal{G}$ is to learn a mapping $\mathcal{M} : \mathcal{V} \to \mathcal{L}$ and predict the labels of unlabeled nodes.

---

[1] Note that if the low frequencies and high frequencies dominate the middle frequencies (the frequencies that are around the cutoff frequency), we say that the filter works both as low-pass and high-pass filters.

**Homophily and Heterophily.** The homophily or heterophily of a graph is used to describe the relation of labels between linked nodes in the graphs. The level of homophily of a graph can be measured by $\mathcal{H}(\mathcal{G}) = \mathbb{E}_{i \in \mathcal{V}} \left[ \left| \{j\}_{j \in \mathcal{N}_i, y_i = y_j} \right| / |\mathcal{N}_i| \right]$ (Pei et al., 2020; Chien et al., 2021), where $\left| \{j\}_{j \in \mathcal{N}_i, y_i = y_j} \right|$ denotes the number of neighbors of $i \in V$ that share the same label as $i$ and $\mathcal{H}(\mathcal{G}) \to 1$ corresponds to strong homophily while $\mathcal{H}(\mathcal{G}) \to 0$ indicates strong heterophily. We say that a graph is a homophilic (heterophilic) graph if it has strong homophily (heterophily).

**Graph Gradient.** The graph gradient of an edge $[i, j], i, j \in \mathcal{V}$ is defined to be a measurement of the variation of a function $f^2 : \mathcal{V} \to \mathbb{R}$ on the edge $[i, j]$.

**Definition 1** (Graph Gradient). *Given a graph $\mathcal{G} = (\mathcal{V}, \mathcal{E})$ and a function $f : \mathcal{V} \to \mathbb{R}$, the graph gradient is an operator $\nabla : \mathcal{F}_\mathcal{V} \to \mathcal{F}_\mathcal{E}$ defined by*

$$(\nabla f)([i, j]) := \sqrt{\frac{W_{i,j}}{D_{j,j}}} f(j) - \sqrt{\frac{W_{i,j}}{D_{i,i}}} f(i), \quad \text{for all } [i, j] \in \mathcal{E}. \tag{1}$$

For $[i, j] \notin \mathcal{E}$, $(\nabla f)([i, j]) := 0$. The graph gradient of a function $f$ at vertex $i$ is defined to be $\nabla f(i) := ((\nabla f)([i, 1]), \dots, (\nabla f)([i, N]))$ and its Frobenius norm is given by $\|\nabla f(i)\|_2 := (\sum_{j=1}^N (\nabla f)^2([i, j]))^{1/2}$, which measures the variation of $f$ around node $i$. We measure the variation of $f$ over the whole graph $\mathcal{G}$ by $\mathcal{S}_p(f)$ where it is defined to be

$$\mathcal{S}_p(f) := \frac{1}{2} \sum_{i=1}^N \sum_{j=1}^N \|(\nabla f)([i, j])\|^p = \frac{1}{2} \sum_{i=1}^N \sum_{j=1}^N \left\| \sqrt{\frac{W_{i,j}}{D_{j,j}}} f(j) - \sqrt{\frac{W_{i,j}}{D_{i,i}}} f(i) \right\|^p, \text{ for } p \geq 1, \tag{2}$$

Note that the definition of $\mathcal{S}_p$ here is different with the $p$-Dirichlet form in Zhou & Schölkopf (2005).

**Graph Divergence.** The graph divergence is defined to be the adjoint of the graph gradient:

**Definition 2** (Graph Divergence). *Given a graph $\mathcal{G} = (\mathcal{V}, \mathcal{E})$, and functions $f : \mathcal{V} \to \mathbb{R}$, $g : \mathcal{E} \to \mathbb{R}$, the graph divergence is an operator $\text{div} : \mathcal{F}_\mathcal{E} \to \mathcal{F}_\mathcal{V}$ which satisfies*

$$\langle \nabla f, g \rangle = \langle f, -\text{div} g \rangle. \tag{3}$$

*The graph divergence can be computed by*

$$(\text{div} g)(i) = \sum_{j=1}^N \sqrt{\frac{W_{i,j}}{D_{i,i}}} \left( g([i, j]) - g([j, i]) \right). \tag{4}$$

Fig. 4 in Appendix E.1 gives a tiny example of illustration of graph gradient and graph divergence.

**Graph $p$-Laplacian Operator.** By the definitions of graph gradient and graph divergence, we reach the definition of graph $p$-Laplacian operator as below.

**Definition 3** (Graph $p$-Laplacian[3]). *Given a graph $\mathcal{G} = (\mathcal{V}, \mathcal{E})$ and a function $f : \mathcal{V} \to \mathbb{R}$, the graph $p$-Laplacian is an operator $\Delta_p : \mathcal{F}_\mathcal{V} \to \mathcal{F}_\mathcal{V}$ defined by*

$$\Delta_p f := -\frac{1}{2} \text{div}(\|\nabla f\|^{p-2} \nabla f), \text{ for } p \geq 1. \tag{5}$$

*where $\| \cdot \|^{p-2}$ is element-wise, i.e. $\|\nabla f(i)\|^{p-2} = (\|(\nabla f)([i, 1])\|^{p-2}, \dots, \|(\nabla f)([i, N])\|^{p-2})$.*

Substituting Eq. (1) and Eq. (4) into Eq. (5), we obtain

$$(\Delta_p f)(i) = \sum_{j=1}^N \sqrt{\frac{W_{i,j}}{D_{i,i}}} \|(\nabla f)([j, i])\|^{p-2} \left( \sqrt{\frac{W_{i,j}}{D_{i,i}}} f(i) - \sqrt{\frac{W_{i,j}}{D_{j,j}}} f(j) \right) \tag{6}$$

The graph $p$-Laplacian is semi-definite: $\langle f, \Delta_p f \rangle = \mathcal{S}_p(f) \geq 0$ and we have

$$\left. \frac{\partial \mathcal{S}_p(f)}{\partial f} \right|_i = p(\Delta_p f)(i). \tag{7}$$

When $p = 2$, $\Delta_2$ is refered as the ordinary Laplacian operator and $\Delta_2 = \mathbf{I} - \mathbf{D}^{-1/2} \mathbf{W} \mathbf{D}^{-1/2}$ and when $p = 1$, $\Delta_1$ is refered as the Curvature operator and $\Delta_1 f := -\frac{1}{2} \text{div}(\|\nabla f\|^{-1} \nabla f)$. Note that Laplacian $\Delta_2$ is a linear operator, while in general for $p \neq 2$, $p$-Laplacian is nonlinear since $\Delta_p(af) \neq a\Delta_p(f)$ for $a \in \mathbb{R}$.

---

[2] $f$ can be a vector function: $f : \mathcal{V} \to \mathbb{R}^c$ for some $c \in \mathbb{N}$ and here we use $f : \mathcal{V} \to \mathbb{R}$ for better illustration.

[3] Note that the definition adopted is slightly different with the one used in Zhou & Schölkopf (2005) where $\| \cdot \|^{p-2}$ is not element-wise and the one used in some literature such as Amghibech (2003); Bühler & Hein (2009), where $(\Delta_p f)(i) = \sum_{j=1}^N \frac{W_{i,j}}{D_{i,i}} |f(i) - f(j)|^{p-2} (f(i) - f(j))$ for $p > 1$ and $p = 1$ is not allowed.

## 3  $p$-LAPLACIAN BASED GRAPH NEURAL NETWORKS

In this section, we derive the $p$-Laplacian message passing scheme from a $p$-Laplacian regularization framework and present $^p$GNN, a new GNN architecture developed upon the new message passing scheme. We theoretically characterize how $p$-Laplacian message passing adaptively learns aggregation weights and profits $^p$GNN for being effective on both homophilic and heterophilic graphs.

### 3.1  $p$-LAPLACIAN REGULARIZATION FRAMEWORK

Given an undirected graph $\mathcal{G} = (\mathcal{V}, \mathcal{E})$ and a signal function with $c$ ($c \in \mathbb{N}$) channels $f : \mathcal{V} \to \mathbb{R}^c$, let $\mathbf{X} = (\mathbf{X}_{1,:}^\top, \dots, \mathbf{X}_{N,:}^\top)^\top \in \mathbb{R}^{N \times c}$ with $\mathbf{X}_{i,:} \in \mathbb{R}^{1 \times c}, i \in [N]$ denoting the node features of $\mathcal{G}$ and $\mathbf{F} = (\mathbf{F}_{1,:}^\top, \dots, \mathbf{F}_{N,:}^\top)^\top \in \mathbb{R}^{N \times c}$ be a matrix whose $i^{th}$ row vector $\mathbf{F}_{i,:} \in \mathbb{R}^{1 \times c}, i \in [N]$ represents the function value of $f$ at the $i$-th vertex in $\mathcal{G}$. We present a $p$-Laplacian regularization problem whose cost function is defined to be

$$\mathbf{F}^* = \arg\min_{\mathbf{F}} \mathcal{L}(\mathbf{F}) := \arg\min_{\mathbf{F}} \mathcal{S}_p(\mathbf{F}) + \mu \sum_{i=1}^{N} \|\mathbf{F}_{i,:} - \mathbf{X}_{i,:}\|^2, \tag{8}$$

where $\mu \in (0, \infty)$. The first term of the right-hand side in Eq. (8) is a measurement of variation of the signal over the graph based on $p$-Laplacian. As we will discuss later, different choices of $p$ result in different smoothness constraint on the signals. The second term is the constraint that the optimal signals $\mathbf{F}^*$ should not change too much from the input signal $\mathbf{X}$, and $\mu$ provides a trade-off between these two constraints.

**Regularization with $p = 2$.** When $p = 2$, the solution of Eq. (8) satisfies $\Delta_2 \mathbf{F}^* + \mu(\mathbf{F}^* - \mathbf{X}) = \mathbf{0}$ and we can obtain the closed form (Zhou et al., 2003; Zhou & Schölkopf, 2005)

$$\mathbf{F}^* = \mu(\Delta_2 + \mu \mathbf{I}_N)^{-1} \mathbf{X}. \tag{9}$$

Then, we could use the following iteration algorithm to get an approximation of Eq. (9):

$$\mathbf{F}^{(k+1)} = \alpha \mathbf{D}^{-1/2} \mathbf{W} \mathbf{D}^{-1/2} \mathbf{F}^{(k)} + \beta \mathbf{X}, \tag{10}$$

where $k$ represents the iteration index, $\alpha = \frac{1}{1+\mu}$ and $\beta = \frac{\mu}{1+\mu} = 1 - \alpha$. The iteration converges to a closed-form solution as $k$ goes to infinity (Zhou et al., 2003; Zhou & Schölkopf, 2005). We could relate the the result here with the personalized PageRank (PPR) (Page et al., 1999; Klicpera et al., 2019) algorithm (proof defered to Appendix D.1):

**Theorem 1** (Relation to personalized PageRank (Klicpera et al., 2019)). $\mu(\Delta_2 + \mu \mathbf{I}_N)^{-1}$ *in the closed form solution of Eq.* (9) *is equivalent to the personalized PageRank matrix.*

**Regularization with $p > 1$.** For $p > 1$, the solution of Eq. (8) satisfies $p\Delta_p \mathbf{F}^* + 2\mu(\mathbf{F}^* - \mathbf{X}) = \mathbf{0}$. By Eq. (6) we have that, for all $i \in [N]$,

$$\sum_{j=1}^{N} \frac{W_{i,j}}{\sqrt{D_{i,i}}} \|(\nabla f^*)([j,i])\|^{p-2} \left( \frac{1}{\sqrt{D_{i,i}}} \mathbf{F}_{i,:}^* - \frac{1}{\sqrt{D_{j,j}}} \mathbf{F}_{j,:}^* \right) + \frac{2\mu}{p} \left( \mathbf{F}_{i,:}^* - \mathbf{X}_{i,:} \right) = \mathbf{0}.$$

Based on which we can construct a similar iterative algorithm to obtain a solution (Zhou & Schölkopf, 2005):

$$\mathbf{F}_{i,:}^{(k+1)} = \alpha_{i,i}^{(k)} \sum_{j=1}^{N} \frac{M_{i,j}^{(k)}}{\sqrt{D_{i,i} D_{j,j}}} \mathbf{F}_{j,:}^{(k)} + \beta_{i,i}^{(k)} \mathbf{X}_{i,:}, \text{ for all } i \in [N], \tag{11}$$

with $\mathbf{M}^{(k)} \in \mathbb{R}^{N \times N}$, $\boldsymbol{\alpha}^{(k)} = \mathrm{diag}(\alpha_{1,1}^{(k)}, \dots, \alpha_{N,N}^{(k)})$, $\boldsymbol{\beta}^{(k)} = \mathrm{diag}(\beta_{1,1}^{(k)}, \dots, \beta_{N,N}^{(k)})$ updated by

$$M_{i,j}^{(k)} = W_{i,j} \left\| \sqrt{\frac{W_{i,j}}{D_{i,i}}} \mathbf{F}_{i,:}^{(k)} - \sqrt{\frac{W_{i,j}}{D_{j,j}}} \mathbf{F}_{j,:}^{(k)} \right\|^{p-2}, \text{ for all } i, j \in [N], \tag{12}$$

$$\alpha_{i,i}^{(k)} = 1 / \left( \sum_{j=1}^{N} \frac{M_{i,j}^{(k)}}{D_{i,i}} + \frac{2\mu}{p} \right), \quad \beta_{i,i}^{(k)} = \frac{2\mu}{p} \alpha_{i,i}, \quad \text{for all } i \in [N], \tag{13}$$

Note that in Eq. (12), when $\left\| \sqrt{\frac{W_{i,j}}{D_{i,i}}} \mathbf{F}_{i,:}^{(k)} - \sqrt{\frac{W_{i,j}}{D_{j,j}}} \mathbf{F}_{j,:}^{(k)} \right\| = 0$, we set $M_{i,j}^{(k)} = 0$. It is easy to see that Eq. (10) is the special cases of Eq. (14) with $p = 2$.

**Remark 1** (Discussion on $p = 1$). *For $p = 1$, when $f$ is a real-valued function $(c = 1)$, $\Delta_1 f$ is a step function, which could make the stationary condition of the objective function Eq. (8) become problematic. Additionally,$\Delta_1 f$ is not continuous at $\|(\nabla f)([i, j])\| = 0$. Therefore, $p = 1$ is not allowed when $f$ is a real value function. On the other hand, note that there is a Frobenius norm in $\Delta_p f$. When $f$ is a vector-valued function $(c > 1)$, the step function in $\Delta_1 f$ only exists on the axes. The stationary condition will be fine if the node embeddings $\mathbf{F}$ are not a matrix of vectors that has only one non-zero element, which is true for many graphs. $p = 1$ may work for these graphs. Overall, we suggest to use $p > 1$ in practice but $p = 1$ may work for graphs with multiple channel signals as well. We conduct experiments for $p > 1$ (e.g., $p = 1.5, 2, 2.5$) and $p = 1$ in Sec. 5.*

### 3.2 $p$-LAPLACIAN MESSAGE PASSING AND $^p$GNN ARCHITECTURE

**$p$-Laplacian Message Passing.** Rewrite Eq. (11) in a matrix form we obtain

$$\mathbf{F}^{(k+1)} = \boldsymbol{\alpha}^{(k)}\mathbf{D}^{-1/2}\mathbf{M}^{(k)}\mathbf{D}^{-1/2}\mathbf{F}^{(k)} + \boldsymbol{\beta}^{(k)}\mathbf{X}. \tag{14}$$

Eq. (14) provides a new message passing mechanism, named $p$-Laplacian message passing.

**Remark 2.** $\boldsymbol{\alpha}\mathbf{D}^{-1/2}\mathbf{M}\mathbf{D}^{-1/2}$ *in Eq. (14) can be regarded as the learned aggregation weights at each step for message passing. It suggests that $p$-Laplacian message passing could adaptively tune the aggregation weights during the course of learning, which will be demonstrated theoretically and empirically in the sequel of this paper. $\beta\mathbf{X}$ in Eq. (14) can be regarded as a residual unit, which helps the model escape from the oversmoothing issue (Chien et al., 2021).*

We present the following theorem to show the shrinking property of $p$-Laplacian message passing.

**Theorem 2** (Shrinking Property of $p$-Laplacian Message Passing). *Given a graph $\mathcal{G} = (\mathcal{V}, \mathcal{E}, \mathbf{W})$ with node features $\mathbf{X}$, $\boldsymbol{\beta}^{(k)}, \mathbf{F}^{(k)}, \mathbf{M}^{(k)}, \boldsymbol{\alpha}^{(k)}$ are updated accordingly to Equations (11) to (13) for $k = 0, 1, \ldots, K$ and $\mathbf{F}^{(0)} = \mathbf{X}$. Then there exist some positive real value $\mu > 0$ which depends on $\mathbf{X}, \mathcal{G}, p$ and $p > 1$ such that*

$$\mathcal{L}_p(\mathbf{F}^{(k+1)}) \leq \mathcal{L}_p(\mathbf{F}^{(k)}).$$

Proof see Appendix D.2. Thm. 2 shows that with some proper positive real value $\mu$ and $p > 1$, the loss of the objective function Eq. (8) is guaranteed to decline after taking one step $p$-Laplacian message passing. Thm. 2 also demonstrates that the iteration Equations (11) to (13) is guaranteed to converge for $p > 1$ with some proper $\mu$ which is chosen depends on the input graph and $p$.

**$^p$GNN Architecture.** We design the architecture of $^p$GNNs using $p$-Laplacian message passing. Given node features $\mathbf{X} \in \mathbb{R}^{N \times c}$, the number of node labels $L$, the number of hidden units $h$, the maximum number of iterations $K$, and $\mathbf{M}$, $\boldsymbol{\alpha}$, and $\boldsymbol{\beta}$ updated by Equations (12) and (13) respectively, we give the $^p$GNN architecture as following:

$$\mathbf{F}^{(0)} = \text{ReLU}(\mathbf{X}\Theta^{(1)}), \tag{15}$$

$$\mathbf{F}^{(k+1)} = \boldsymbol{\alpha}^{(k)}\mathbf{D}^{-1/2}\mathbf{M}^{(k)}\mathbf{D}^{-1/2}\mathbf{F}^{(k)} + \boldsymbol{\beta}^{(k)}\mathbf{F}^{(0)}, \quad k = 0, 1, \ldots, K - 1, \tag{16}$$

$$\mathbf{Z} = \text{softmax}(\mathbf{F}^{(K)}\Theta^{(2)}), \tag{17}$$

where $\mathbf{Z} \in \mathbb{R}^{N \times L}$ is the output propbability matrix with $Z_{i,j}$ is the estimated probability that the label at node $i \in [N]$ is $j \in [L]$ given the features $\mathbf{X}$ and the graph $\mathcal{G}$, $\Theta^{(1)} \in \mathbb{R}^{c \times h}$ and $\Theta^{(2)} \in \mathbb{R}^{h \times L}$ are the first- and the second-layer parameters of the neural network, respectively.

**Remark 3** (Connection to existing GNN variants). *The message passing scheme of $^p$GNNs is different from that of several GNN variants (say, GCN, GAT, and GraphSage), which repeatedly stack message passing layers. In contrast, it is similar with SGC (Wu et al., 2019), APPNP (Klicpera et al., 2019), and GPRGNN (Chien et al., 2021). SGC is an approximation to the closed-form in Eq. (9) (Fu et al., 2020). By Thm. 1, it is easy to see that APPNP, which uses PPR to propagate the node embeddings, is analogical to $^p$GNN with $p = 2$, termed as $^{2.0}$GNN. APPNP and $^{2.0}$GNN work analogically and effectively on homophilic graphs. $^{2.0}$GNN can also work effectively on heterophilic graphs by letting $\Theta^{(2)}$ be negative. However, APPNP fails on heterophilic graphs as its PPR weights*

*are fixed (Chien et al., 2021). Unlike APPNP, GPRGNN, which adaptively learn the generalized PageRank (GPR) weights, works similarly to $^{2.0}$GNN on both homophilic and heterophilic graphs. However, GPRGNN needs more supervised information in order to learn optimal GPR weights. On the contrary, $^p$GNNs need less supervised information to obtain similar results because $\Theta^{(2)}$ acts like a hyperplane for classification. $^p$GNNs could work better under weak supervised information. Our analysis is also verified by the experimental results in Sec. 5.*

We also provide an upper-bounding risk of $^p$GNNs by Thm. 4 in Appendix C.1 to study the effect of the hyperparameter $\mu$ on the performance of $^p$GNNs. Thm. 4 shows that the risk of $^p$GNNs is upper-bounded by the sum of three terms: the risk of label prediction using only the original node features $\mathbf{X}$, the norm of $p$-Laplacian diffusion on $\mathbf{X}$, and the magnitude of the noise in $\mathbf{X}$. $\mu$ controls the trade-off between these three terms. The smaller $\mu$, the more weights on the $p$-Laplacian diffusion term and the noise term and the less weights on the the other term and vice versa.

## 4 SPECTRAL VIEWS OF $p$-LAPLACIAN MESSAGE PASSING

In this section, we theoretically demonstrate that $p$-Laplacian message passing is an approximation of a polynomial graph filter defined on the spectral domain of $p$-Laplacian. We show by spectral analysis that $p$-Laplacian message passing works simultaneously as low-pass and high-pass filters.

**$p$-Eigenvalues and $p$-Eigenvectors of the Graph $p$-Laplacian.** We first introduce the definitions of $p$-eigenvalues and $p$-eigenvectors of $p$-Laplacian. Let $\phi_p : \mathbb{R} \to \mathbb{R}$ defined as $\phi_p(u) = \|u\|^{p-2}u$, for $u \in \mathbb{R}, u \neq 0$. Note that $\phi_2(u) = u$. For notational simplicity, we denote by $\phi_p(\mathbf{u}) = (\phi_p(u_1), \ldots, \phi_p(u_N))^\top$ for $\mathbf{u} \in \mathbb{R}^N$ and $\Phi_p(\mathbf{U}) = (\phi_p(\mathbf{U}_{:,1}), \ldots, \phi_p(\mathbf{U}_{:,N}))$ for $\mathbf{U} \in \mathbb{R}^{N \times N}$ and $\mathbf{U}_{:,i} \in \mathbb{R}^N$ is the $i^{th}$ column vector of $\mathbf{U}$.

**Definition 4** (*$p$-Eigenvector and $p$-Eigenvalue*)**.** *A vector $\mathbf{u} \in \mathbb{R}^N$ is a $p$-eigenvector of $\Delta_p$ if it satisfies the equation*

$$(\Delta_p \mathbf{u})_i = \lambda \phi_p(u_i), \quad \text{for all } i \in [N],$$

*where $\lambda \in \mathbb{R}$ is a real value referred as a $p$-eigenvalue of $\Delta_p$ associated with the $p$-eigenvector $\mathbf{u}$.*

**Definition 5** (*$p$-Orthogonal (Luo, Huang, Ding, and Nie, 2010)*)**.** *Given two vectors $\mathbf{u}, \mathbf{v} \in \mathbb{R}^N$ with $\mathbf{u}, \mathbf{v} \neq \mathbf{0}$, we call that $\mathbf{u}$ and $\mathbf{v}$ is $p$-orthogonal if $\phi_p(\mathbf{u})^\top \phi_p(\mathbf{v}) = \sum_{i=1}^N \phi_p(u_i)\phi_p(v_i) = 0$.*

Luo et al. (2010) demonstrated that the $p$-eigenvectors of $\Delta_p$ are $p$-orthogonal to each other (see Thm. 5 in Appendix C.2 for details). Therefore, the space spanned by the multiple $p$-eigenvectors of $\Delta_p$ is $p$-orthogonal. Additionally, we demonstrate that the $p$-eigen-decomposition of $\Delta_p$ is given by: $\Delta_p = \Phi_p(\mathbf{U})\mathbf{\Lambda}\Phi_p(\mathbf{U})^\top$ (see Thm. 6 in Appendix C.3 for details), where $\mathbf{U}$ is a matrix of $p$-eigenvectors of $\Delta_p$ and $\mathbf{\Lambda}$ is a diagonal matrix in which the diagonal is the $p$-eigenvalues of $\Delta_p$.

**Graph Convolutions based on $p$-Laplacian.** Based on Thm. 5, the graph Fourier Transform $\hat{f}$ of any function $f$ on the vertices of $\mathcal{G}$ can be defined as the expansion of $f$ in terms of $\Phi(\mathbf{U})$ where $\mathbf{U}$ is the matrix of $p$-eigenvectors of $\Delta_p$: $\hat{f} = \Phi_p(\mathbf{U})^\top f$. Similarly, the inverse graph Fourier transform is then given by: $f = \Phi_p(\mathbf{U})\hat{f}$. Therefore, a signal $\mathbf{X} \in \mathbb{R}^{N \times c}$ being filtered by a spectral filter $g_\theta$ can be expressed formally as: $g_{\boldsymbol{\theta}} \star \mathbf{X} = \Phi_p(\mathbf{U})\hat{g}_{\boldsymbol{\theta}}(\mathbf{\Lambda})\Phi_p(\mathbf{U})^\top \mathbf{X}$, where $\mathbf{\Lambda}$ denotes a diagonal matrix in which the diagonal corresponds to the $p$-eigenvalues $\{\lambda_l\}_{l=0,\ldots,N-1}$ of $\Delta_p$ and $\hat{g}_{\boldsymbol{\theta}}(\mathbf{\Lambda})$ denotes a diagonal matrix in which the diagonal corresponds to spectral filter coefficients. Let $\hat{g}_{\boldsymbol{\theta}}$ be a polynomial filter defined as $\hat{g}_{\boldsymbol{\theta}} = \sum_{k=0}^{K-1} \theta_k \lambda_l^k$, where the parameter $\boldsymbol{\theta} = [\theta_0, \ldots, \theta_{K-1}]^\top \in \mathbb{R}^K$ is a vector of polynomial coefficients. By the $p$-eigen-decomposition of $p$-Laplacian, we have

$$g_{\boldsymbol{\theta}} \star \mathbf{X} \approx \sum_{k=0}^{K-1} \theta_k \Phi_p(\mathbf{U})\mathbf{\Lambda}^k \Phi_p(\mathbf{U})^\top \mathbf{X} = \sum_{k=0}^{K-1} \theta_k \Delta_p^k \mathbf{X}. \tag{18}$$

**Theorem 3.** *The $K$-step $p$-Laplacian message passing is a $K$-order polynomial approximation to the graph filter given by Eq. (18).*

Proof see Appendix D.3. Thm. 3 indicates that $p$-Laplacian message passing mechanism is implicitly a polynomial spectral filter defined on the spectral domain of $p$-Laplacian.

**Spectral Analysis of $p$-Laplacian Message Passing.** Here, we analyze the spectral propecties of $p$-Laplacian message passing. We can approximately view $p$-Laplacian message pasing as a filter of a linear combination of $K$ spectral filters $g(\mathbf{\Lambda})^{(0)}, g(\mathbf{\Lambda})^{(1)}, \ldots, g(\mathbf{\Lambda})^{(K-1)}$ with each spectral filter defined to be $g(\mathbf{\Lambda})^{(k)} := (\boldsymbol{\alpha}\mathbf{D}^{-1/2}\mathbf{M}\mathbf{D}^{-1/2})^k$ where $M_{i,j} = W_{i,j}\|\sqrt{\frac{W_{i,j}}{D_{i,i}}}\mathbf{F}_{i,:} - \sqrt{\frac{W_{i,j}}{D_{j,j}}}\mathbf{F}_{j,:}\|^{p-2}$ for $i, j \in [N]$ and $\mathbf{F}$ is the matrix of node embeddings. We can study the properties of $p$-Laplacian message passing by studying the spectral properties of $\boldsymbol{\alpha}\mathbf{D}^{-1/2}\mathbf{M}\mathbf{D}^{-1/2}$ as given below.

**Proposition 1.** *Given a connected graph $\mathcal{G} = (\mathcal{V}, \mathcal{E}, \mathbf{W})$ with node embeddings $\mathbf{F}$ and the $p$-Laplacian $\Delta_p$ with its $p$-eigenvectors $\{\mathbf{u}^{(l)}\}_{l=0,1,\ldots,N-1}$ and the $p$-eigenvalues $\{\lambda_l\}_{l=0,1,\ldots,N-1}$. Let $g_p(\lambda_{i-1}) := \alpha_{i,i}\sum_j D_{i,i}^{-1/2}M_{i,j}D_{j,j}^{-1/2}$ for $i \in [N]$ be the filters defined on the spectral domain of $\Delta_p$, where $M_{i,j} = W_{i,j}\|\nabla f([i,j])\|^{p-2}$, $(\nabla f)([i,j])$ is the graph gradient of the edge between node $i$ and $j$ and $\|\nabla f(i)\|$ is the norm of graph gradient at $i$. $N_i$ denotes the number of edges connected to $i$, $N_{min} = \min\{N_j\}_{j\in[N]}$, and $k = \arg\max_j(\{|u_j^{(l)}|/\sqrt{D_{l,l}}\}_{j\in[N];l=0,\ldots,N-1})$, then*

1. *When $p = 2$, $g_p(\lambda_{i-1})$ works as both low-pass and high-pass filters.*

2. *When $p > 2$, if $\|\nabla f(i)\| \leq 2^{(p-1)/(p-2)}$, $g_p(\lambda_{i-1})$ works as both low-pass and high-pass filters on node $i$ and $g_p(\lambda_{i-1})$ works as low-pass filters on $i$ when $\|\nabla f(i)\| \geq 2^{(p-1)/(p-2)}$.*

3. *When $1 \leq p < 2$, if $0 \leq \|\nabla f(i)\| \leq 2(2\sqrt{N_k})^{1/(p-2)}$, $g_p(\lambda_{i-1})$ works as low-pass filters on node $i$ and $g_p(\lambda_{i-1})$ works as both low-pass and high-pass filters on $i$ when $\|\nabla f(i)\| \geq 2(2\sqrt{N_k})^{1/(p-2)}$. Specifically, when $p = 1$, $N_k$ can be replaced by $N_{min}$.*

Proof see Appendix D.7. Proposition 1 shows that when $p \neq 2$, $p$-Laplacian message passing adaptively works as low-pass or both low-pass and high-pass filters on node $i$ in terms of the degree of local node embedding variation around $i$, i.e. the norm of the graph gradient $\|\nabla f(i)\|$ at node $i$. When $p = 2$, $p$-Laplacian message passing works as both low-pass and high-pass filters on node $i$ regardless of the value of $\|\nabla f(i)\|$. When $p > 2$, $p$-Laplacian message passing works as low-pass filters on node $i$ for large $\|\nabla f(i)\|$ and works as both low-pass and high-pass filters for small $\|\nabla f(i)\|$. Therefore, $^p$GNNs with $p > 2$ can work very effectively on graphs with strong homophily. When $1 \leq p < 2$, $p$-Laplacian message passing works as low-pass filters for small $\|\nabla f(i)\|$ and works as both low-pass and high-pass filters for large $\|\nabla f(i)\|$. Thus, $^p$GNNs with $1 \leq p < 2$ can work effectively on graphs with low homophily, i.e. heterophilic graphs. The results here confirms our analysis of the aggregation weights of $p$-Laplacian message passing presented in Thm. 2.

## 5 EMPIRICAL STUDIES

In this section, we empirically study the effectiveness of $^p$GNNs for semi-supervised node classification using and real-world benchmark and synthetic datasets with heterophily and strong homophily. The experimental results are also used to validate our theoretical findings presented previously.

**Datasets and Experimental Setup.** We use seven homophilic benchmark datasets: citation graphs Cora, CiteSeer, PubMed (Sen et al., 2008), Amazon co-purchase graphs Computers, Photo, coauthor graphs CS, Physics (Shchur et al., 2018), and six heterophilic benchmark datasets: Wikipedia graphs Chameleon, Squirrel (Rozemberczki et al., 2021), the Actor co-occurrence graph, webpage graphs Wisconsin, Texas, Cornell (Pei et al., 2020). The node classification tasks are conducted in the transductive setting. Following Chien et al. (2021), we use the sparse splitting ($2.5\%/2.5\%/95\%$) and the dense splitting ($60\%/20\%/20\%$) to randomly split the homophilic and heterophilic graphs into training/validation/testing sets, respectively. Dataset statistics and their levels of homophily are presented in Appendix E.

**Baselines.** We compare $^p$GNN with seven models, including MLP, GCN (Kipf & Welling, 2017), SGC (Wu et al., 2019), GAT (Velickovic et al., 2018), JKNet (Xu et al., 2018), APPNP (Klicpera et al., 2019), GPRGNN (Chien et al., 2021). We use the Pytorch Geometric library (Fey & Lenssen, 2019) to implement all baselines except GPRGNN. For GPRGNN, we use the code released by the authors[4]. The details of hyperparameter settings are deferred to Appendix E.3.

[4]https://github.com/jianhao2016/GPRGNN

Table 1: Heterophilious results. Averaged accuracy (%) for 100 runs. Best results outlined in bold and the results within 95% confidence interval of the best results are outlined in underlined bold.

| Method | Chameleon | Squirrel | Actor | Wisconsin | Texas | Cornell |
|---|---|---|---|---|---|---|
| MLP | $48.02_{\pm1.72}$ | $\mathbf{33.80}_{\pm1.05}$ | $39.68_{\pm1.43}$ | $93.56_{\pm3.14}$ | $79.50_{\pm10.62}$ | $80.30_{\pm11.38}$ |
| GCN | $34.54_{\pm2.78}$ | $25.28_{\pm1.55}$ | $31.28_{\pm2.04}$ | $61.93_{\pm3.00}$ | $56.54_{\pm17.02}$ | $51.36_{\pm4.59}$ |
| SGC | $34.76_{\pm4.55}$ | $25.49_{\pm1.63}$ | $30.98_{\pm3.80}$ | $66.94_{\pm2.58}$ | $59.99_{\pm9.95}$ | $44.39_{\pm5.88}$ |
| GAT | $45.16_{\pm2.10}$ | $31.41_{\pm0.98}$ | $34.11_{\pm1.28}$ | $65.64_{\pm6.29}$ | $56.41_{\pm13.01}$ | $43.94_{\pm7.33}$ |
| JKNet | $33.28_{\pm3.59}$ | $25.82_{\pm1.58}$ | $29.77_{\pm2.61}$ | $61.08_{\pm3.71}$ | $59.65_{\pm12.62}$ | $55.34_{\pm4.43}$ |
| APPNP | $36.18_{\pm2.81}$ | $26.85_{\pm1.48}$ | $31.26_{\pm2.52}$ | $64.59_{\pm3.49}$ | $82.90_{\pm5.08}$ | $66.47_{\pm9.34}$ |
| GPRGNN | $43.67_{\pm2.27}$ | $31.27_{\pm1.76}$ | $36.63_{\pm1.22}$ | $88.54_{\pm4.94}$ | $80.74_{\pm6.76}$ | $78.95_{\pm8.52}$ |
| $^{1.0}$GNN | $\mathbf{48.86}_{\pm1.95}$ | $\underline{\mathbf{33.75}}_{\pm1.50}$ | $\mathbf{40.62}_{\pm1.25}$ | $\mathbf{95.37}_{\pm2.06}$ | $84.06_{\pm7.41}$ | $\mathbf{82.16}_{\pm8.62}$ |
| $^{1.5}$GNN | $\underline{\mathbf{48.74}}_{\pm1.62}$ | $33.33_{\pm1.45}$ | $\underline{\mathbf{40.35}}_{\pm1.35}$ | $\underline{\mathbf{95.24}}_{\pm2.01}$ | $84.46_{\pm7.79}$ | $78.47_{\pm6.87}$ |
| $^{2.0}$GNN | $\underline{\mathbf{48.77}}_{\pm1.87}$ | $33.60_{\pm1.47}$ | $40.07_{\pm1.17}$ | $91.15_{\pm2.76}$ | $\mathbf{87.96}_{\pm6.27}$ | $72.04_{\pm8.22}$ |
| $^{2.5}$GNN | $\underline{\mathbf{48.80}}_{\pm1.77}$ | $\underline{\mathbf{33.79}}_{\pm1.45}$ | $39.80_{\pm1.31}$ | $87.08_{\pm2.69}$ | $83.01_{\pm6.80}$ | $70.31_{\pm8.84}$ |

**Superior Performance on Real-World Heterophilic Datasets.** The results on *homophilic* benchmark datasets are deferred to Appendix F.1, which show that $^p$GNNs obtains competitive performance against state-of-the-art GNNs on homophilic datasets. Table 1 summarizes the results on *heterophilic* benchmark datasets. Table 1 shows that $^p$GNNs significantly dominate the baselines and $^{1.0}$GNN obtains the best performance on all heterophilic graphs except the Texas dataset. For Texas, $^{2.0}$GNN is the best. We also observe that MLP works very well and significantly outperforms most GNN baselines, which indicates that the graph topology is not informative for label prediction on these heterophilic graphs. Therefore, propagating and transforming node features over the graph topology could lead to worse performance than MLP. Unlike ordinary GNNs, $^p$GNNs can adaptively learn aggregation weights and ignore edges that are not informative for label prediction and thus could work better. It confirms our theoretical findings presented in previous sections. Note that GAT can also learn aggregation weights, i.e. the attention weights. However, the aggregation weights learned by GAT are significantly distinct from that of $^p$GNNs, as we will show following.

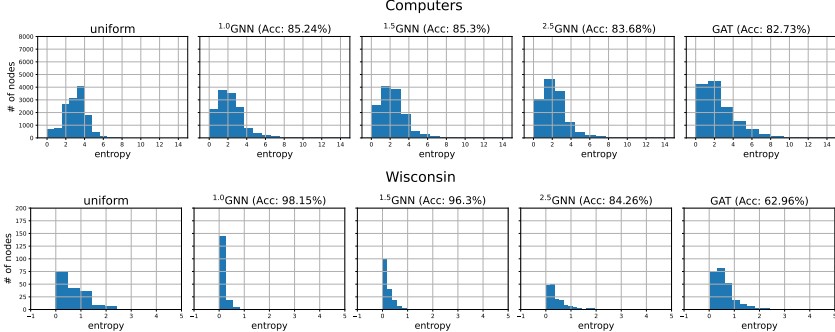

Figure 1: Aggregation weight entropy distribution of graphs. Low entropy means high degree of concentration, vice versa. An entropy of zero means all aggregation weights are on one source node.

**Interpretability of the Learned Aggregation Weights of $^p$GNNs.** We showcase the interpretability of the learned aggregation weights $\alpha_{i,i}D_{i,i}^{-1/2}M_{i,j}D_{j,j}^{-1/2}$ of $^p$GNNs by studying its entropy distribution, along with the attention weights of GAT on real-world datasets. Denote $\{A_{i,j}\}_{j\in\mathcal{N}_i}$ as the aggregation weights of node $i$ and its neighbors. For GAT, $\{A_{i,j}\}_{j\in\mathcal{N}_i}$ are referred as the attention weights (in the first layer) and for $^p$GNNs are $\alpha_{i,i}D_{i,i}^{-1/2}M_{i,j}D_{j,j}^{-1/2}$. For any node $i$, $\{A_{i,j}\}_{j\in\mathcal{N}_i}$ forms a discrete probability distribution over all its neighbors with the entropy given by $H(\{A_{i,j}\}_{j\in\mathcal{N}_i}) = -\sum_{j\in\mathcal{N}_i} A_{i,j}\log(A_{i,j})$. Low entropy means high degree of concentration and vice versa. An entropy of zero means all aggregation weights or attentions are on one source node. The uniform distribution has the highest entropy of $log(D_{i,i})$. Fig. 1 reports the results on Computers, Wisconsin and we defer more results on other datasets to Appendix F.2 due to space limit. Fig. 1 shows that the aggregation weight entropy distributions of GAT and $^p$GNNs on Computers (homophily) are both similar to the uniform case. It indicates the original graph topology of Computers is very helpful for label prediction and therefore GNNs could work very well on Computers. However, for Wisconsin (heterophily), the entropy distribution of $^p$GNNs is significantly different from that of GAT and the uniform case. Most entropy of $^p$GNNs is around zero, which means that most aggregation weights are on one source node. It states that the original graph topology of Wisconsin is not helpful for label prediction, which explains why MLP works well on Wisconsin.

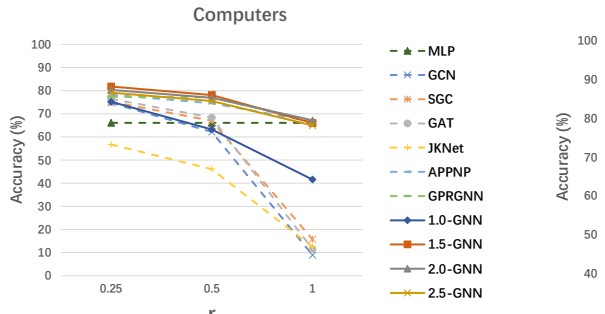
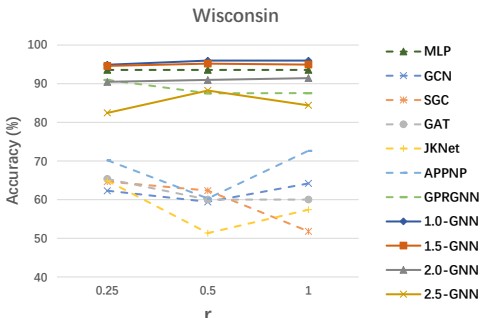

Figure 3: Averaged accuracy (%) on graphs with noisy edges for 20 runs. Best view in colors.

On the contrary, the entropy distribution of GAT is similar to the uniform case and therefore GAT works similarly to GCN and is significantly worse than $^p$GNNs on Wisconsin. Similar results can be observed on the experiments on more datasets in Appendix F.2.

**Results on cSBM Datasets.** We exam the performance of $^p$GNNs on heterophilic graphs whose topology is informative for label prediction using synthetic graphs generated by cSBM (Deshpande et al., 2018) with $\phi \in \{-1, -0.75, \ldots, 1\}$. We use the same settings of cSBM used in Chien et al. (2021). Due to the space limit, we refer the readers to Chien et al. (2021) for more details of cSBM dataset. Fig. 2 reports the results on cSBM using sparse splitting (for results on cSBM with dense splitting see Appendix F.3). Fig. 2 shows that when $\phi \leq -0.5$ (heterophilic graphs), $^{2.0}$GNN obtains the best performance and $^p$GNNs and GPRGNN significantly dominate the others. It validates the effectiveness of $^p$GNNs on het-

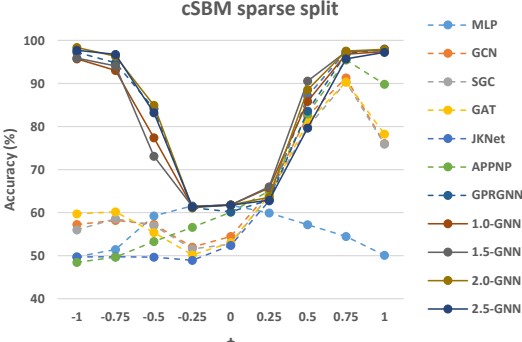

Figure 2: Averaged accuracy (%) on cSBM (sparse split) for 20 runs. Best view in colors.

erophilic graphs. Moreover, $^{2.0}$GNN works better than GPRGNN and it again confirms that $^{2.0}$GNN is more superior under weak supervision (2.5% training rate), as stated in Remark 3. Note that $^{1.0}$GNN and $^{1.5}$GNN are not better than $^{2.0}$GNN, the reason could be the iteration algorithms Eq. (11) with $p = 1, 1.5$ are not as stable as the one with $p = 2$. When the graph topology is almost non-informative for label prediction ($\phi = -0.25, 0$), The performance of $^p$GNNs is close to MLP and they outperform the other baselines. Again, it validates that $^p$GNNs can erase non-informative edges and work as well as MLP and confirms the statements in Thm. 4. When the graph is homophilic ($\phi \geq 0.25$), $^{1.5}$GNN is the best on weak homophilic graphs ($\phi = 0.25, 0.5$) and $^p$GNNs work competitively with all GNN baselines on strong homophilic graphs ($\phi \geq 0.75$).

**Results on Datasets with Noisy Edges.** We conduct experiments to evaluate the performance of $^p$GNNs on graphs with noisy edges by randomly adding edges to the graphs and randomly remove the same number of original edges. We define the random edge rate as $r := \frac{\#\text{random edges}}{\#\text{all edges}}$. The experiments are conducted on 4 homophilic datasets (Computers, Photo, CS, Physics) and 2 heterophilic datasets (Wisconsin, Texas) with $r = 0.25, 0.5, 1$. Fig. 3 reports the results on Computers, Wisconsin and we defer more results to Appendix F.4. Fig. 3 shows that $^p$GNNs significantly outperform all baselines. Specifically, $^{1.5}$GNN obtains the best performance on Computers, and $^{1.5}$GNN and $^{2.0}$GNN even work as well as MLP on Computers with completely random edges ($r = 1$). For Wisconsin, $^{1.0}$GNN is the best, and $^{1.0}$GNN and $^{1.5}$GNN significantly dominate the others. We also observed that APPNP and GPRGNN, whose architectures are analogical to $^{2.0}$GNN, also work better than other GNNs. Nevertheless, they are significantly outperformed by $^p$GNNs overall. Similar results can be observed in the experiments conducted on more datasets as presented in Appendix F.4.

CONCLUSION. We have addressed the problem of generalizing GNNs to heterophilic graphs and graphs with noisy edges. To this end, we derived a novel $p$-Laplacian message passing scheme from a discrete regularization framework and proposed a new $^p$GNN architecture. We theoretically demonstrate our method works as low-pass and high-pass filters and thereby applicable to both homophilic and heterophilic graphs. We empirically validate our theoretical results and show the advantages of our methods on heterophilic graphs and graphs with non-informative topologies.

## REPRODUCIBILITY STATEMENT

In order to ensure reproducibility, we have made the efforts in the following respects: (1) Provide a sampled code as the supplementary material; (2) Provide self-contained proofs of the main claims in Appendices C and D; (3) Provide more details on experimental configurations in Appendix E and experimental results in Appendix F. All the datasets are publicly available as described in the main text. We will fully release our training and evaluation code, as well as our train/validation/test splits.

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

# Appendix

## CONTENTS

# A   RELATED WORK

**Graph Neural Networks.** Graph neural networks (GNNs) are a variant of neural networks for graph-structured data, which can propagate and transform the node features over the graph topology and exploit the information in the graphs. Graph convolutional networks (GCNs) are one type of GNNs whose graph convolution mechanisms or the message passing schemes were mainly inspired by the field of graph signal processing. Bruna et al. (2014) defined a nonparametric graph filter using the Fourier coefficients. Defferrard et al. (2016) introduced Chebyshev polynomial to avoid computational expensive eigen-decomposition of Laplacian and obtain localized spectral filters. GCN (Kipf & Welling, 2017) used the first-order approximation and reparameterized trick to simplify the spectral filters and obtain the layer-wise graph convolution. SGC (Wu et al., 2019) further simplify GCN by removing non-linear transition functions between each layer. Chen et al. (2018) propose importance sampling to design an efficient variant of GCN. Xu et al. (2018) explored a jumping knowledge architecture that flexibly leverages different neighborhood ranges for each node to enable better structure-aware representation. Atwood & Towsley (2016); Liao et al. (2019); Abu-El-Haija et al. (2019) exploited multi-scale information by diffusing multi-hop neighbor information over the graph topology. Wang & Leskovec (2020) used label propagation to improve GCNs. Klicpera et al. (2019) incorporated personalized PageRank with GCNs. Liu et al. (2021) introduced a $l_1$ norm-based graph smoothing term to enhance the local smoothnesss adaptivity of GNNs. Hamilton et al. (2017); Zeng et al. (2020) proposed sampling and aggregation frameworks to extent GCNs to inductive learning settings. Another variant of GNNs is graph attention networks (Velickovic et al., 2018; Thekumparampil et al., 2018; Abu-El-Haija et al., 2018), which use attention mechanisms to adaptively learn aggregation weights based on the nodes features. There are many other works on GNNs (Pei et al., 2020) (Ying et al., 2018; Xinyi & Chen, 2019; Velickovic et al., 2019; Zeng et al., 2020), we refer to Zhou et al. (2020); Battaglia et al. (2018); Wu et al. (2021) for a comprehensive review. Most GNN models implicitly assume that the labels of nodes and their neighbors should be the same or consistent, while it does not hold for heterophilic graphs. Zhu et al. (2020) investigated the issues of GNNs on heterophilic graphs and proposed to separately learn the embeddings of ego-node and its neighborhood. Zhu et al. (2021) proposed a framework to model the heterophily or homophily levels of graphs. Chien et al. (2021) incorporated generalized PageRank with graph convolution to adapt GNNs to heterophilic graphs.

There are also some works on the interpretability of GNNs proposed recently. Li et al. (2018); Ying et al. (2019); Fu et al. (2020) showed that spectral graph convolutions work as conducting Laplacian smoothing on the graph signals and Wu et al. (2019); NT & Maehara (2019) demonstrated that GCN, SGC work as low-pass filters. Gama et al. (2020) studied the stability properties of GNNs. Xu et al. (2019); Oono & Suzuki (2020); Loukas (2020) studied the expressiveness of GNNs. Verma & Zhang (2019); Garg et al. (2020) work on the generalization and representation power of GNNs.

**Graph based Semi-supervised Learning.** Graph-based semi-supervised learning works under the assumption that the labels of a node and its neighbors shall be the same or consistent. Many methods have been proposed in the last decade, such as Smola & Kondor (2003); Zhou et al. (2003); Belkin et al. (2004) use Laplacian regularization techniques to force the labels of linked nodes to be the same or consistent. Zhou & Schölkopf (2005) introduce discrete regularization techniques to impose different regularizations on the node features based on $p$-Laplacian. Lable propagation (Zhu et al., 2003) recursively propagates the labels of labeled nodes over the graph topology and use the convergence results to make predictions. To mention but a few, we refer to Zhou & Schölkopf (2005); van Engelen & Hoos (2020) for a more comprehensive review.

# B    DISCUSSIONS AND FUTURE WORK

In this section, we discuss the future work of $^p$GNNs. Our theoretical results and experimental results could lead to several potential extensions of $^p$GNNs.

**New Paradigm of Designing GNN Architectures.** We bridge the gap between discrete regularization framework, graph-based semi-supervised learning, and GNNs, which provides a new paradigm of designing new GNN architectures. Following the new paradigm, researchers could introduce more regularization techniques, e.g., Laplacian regularization (Smola & Kondor, 2003; Belkin et al., 2004), manifold regularization (Sindhwani et al., 2005; Belkin et al., 2005; Niyogi, 2013), high-order regularization (Zhou & Belkin, 2011), Bayesian regularization (Liu et al., 2014), entropy regularization (Grandvalet & Bengio, 2004), and consider more explicit assumptions on graphs, e.g. the homophily assumption, the low-density region assumption (i.e. the decision boundary is likely to lie in a low data density region), manifold assumption (i.e. the high dimensional data lies on a low-dimensional manifold), to develop new graph convolutions or message passing schemes for graphs with specific properties and generalize GNNs to a much broader range of graphs. Moreover, the paradigm also enables us to explicitly study the behaviors of the designed graph convolutions or message passing schemes from the theory of regularization (Belkin & Niyogi, 2008; Niyogi, 2013; Slepcev & Thorpe, 2017).

**Applications of $^p$GNNs to learn on graphs with noisy topologies.** The empirical results (as shown in Fig. 3 and Tables 6 and 7) on graphs with noisy edges show that $^p$GNNs are very robust to noisy edges, which suggests the applications of $p$-Laplacian message passing and $^p$GNNs on the graph learning scenarios where the graph topology could potentially be seriously intervened.

**Integrating with existing GNN architectures.** As shown in Table 9, the experimental results on heterophilic benchmark datasets illustrate that integrating GCN, JKNet with $^p$GNNs can significantly improve their performance on heterophilic graphs. It shows that $^p$GNN could be used as a plug-and-play component to be integrated into existing GNN architectures and improve their performance on real-world applications.

**Inductive learning for $^p$GNNs.** $^p$GNNs are shown to be very effective for inductive learning on PPI datasets as reported in Table 10. $^p$GNNs even outperforms GAT on PPI, while using much fewer parameters than GAT. It suggests the promising extensions of $^p$GNNs to inductive learning on graphs.

## C  ADDITIONAL THEOREMS

### C.1  THEOREM 4 (UPPER-BOUNDING RISK OF $^p$GNN)

**Theorem 4** (Upper-bounding risks of $^p$GNNs)**.** *Given a graph $\mathcal{G} = (\mathcal{V}, \mathcal{E}, \mathbf{W})$ with $N$ nodes, let $\mathbf{X} \in \mathbb{R}^{N \times c}$ be the node features and $\mathbf{y} \in \mathbb{R}^N$ be the node labels and $\mathbf{M}^{(k)}, \boldsymbol{\alpha}^{(k)}, \boldsymbol{\beta}^k, \mathbf{F}^k$ are updated accordingly by Equations (12) to (14) for $k = 0, 1, \ldots, K - 1$ and $\mathbf{F}^{(0)} = \mathbf{X}$, $K \in \mathbb{N}$. Assume that $\mathcal{G}$ is $d$-regular and the ground-truth node features $\mathbf{X}^* = \mathbf{X} + \boldsymbol{\epsilon}$, where $\boldsymbol{\epsilon} \in \mathbb{R}^{N \times c}$ represents the noise in the node features and there exists a $L$-Lipschitz function $\sigma : \mathbb{R}^{N \times c} \to \mathbb{R}^N$ such that $\sigma(\mathbf{X}^*) = \mathbf{y}$. let $\tilde{\mathbf{y}}^{(k+1)} = \boldsymbol{\alpha}^{(k)} \mathbf{D}^{-1/2} \mathbf{M}^{(k)} \mathbf{D}^{-1/2} \sigma(\mathbf{F}^{(k)}) + \boldsymbol{\beta}^{(k)} \sigma(\mathbf{F}^{(0)})$, we have*

$$\frac{1}{N} \sum_{i=1}^N |y_i - \tilde{y}_i| \le \frac{1}{N} \sum_{i=1}^N \beta_{i,i}^{(K-1)} |y_i - \sigma(\mathbf{X}_{i,:})|$$

$$+ \frac{L}{N} \sum_{i=1}^N \alpha_{i,i}^{(K-1)} \left\| \Delta_p^{(K-1)} \mathbf{F}_{i,:}^{(K-1)} + \sum_{k=0}^{K-2} \prod_{l=k}^{K-2} \left( \sum_{j=1}^N \frac{M_{i,j}^{(l)}}{d} \right) \Delta_p^{(k)} \mathbf{X}_{i,:} \right\|$$

$$+ \frac{L}{N} \sum_{i=1}^N (1 - \beta_{i,i}^{(K-1)}) \left\| \boldsymbol{\epsilon}_{i,:} \right\|.$$

Proof see Appendix D.4. Thm. 4 shows that the risk of $^p$GNNs is upper-bounded by the sum of three terms: The first term of the r.h.s in the above inequation represents the risk of label prediction using only the original node features $\mathbf{X}$, the second term is the norm of $p$-Laplacian diffusion on the node features $\mathbf{X}$, and the third term is the magnitude of the noise in the node features. $\alpha_{i,i}$ and $\beta_{i,i}$ control the trade-off between these three terms and they are related to the hyperparameter $\mu$ in Eq. (10). The smaller $\mu$, the smaller $\beta_{i,i}$ and larger $\alpha_{i,i}$, thus the more important of the $p$-Laplacian diffusion term but also the more effect from the noise. Therefore, for graphs whose topological information is not helpful for label prediction, we could impose more weights on the first term by using a large $\mu$ so that $^p$GNNs work more like MLPs which simply learn on node features. While for graphs whose topological information is helpful for label prediction, we could impose more weights on the second term by using a small $\mu$ so that $^p$GNNs can benefit from $p$-Laplacian smoothing on node features.

In practice, to choose a proper value of $\mu$ one may first simply apply MLPs on the node features to have a glance at the helpfulness of the node features. If MLPs work very well, there is not much space for the graph's topological information to further improve the prediction performance and we may choose a large $\mu$. Otherwise, there could be a large chance for the graph's topological information to further improve the performance and we should choose a small $\mu$.

### C.2  THEOREM 5 ($p$-ORTHOGONAL THEOREM (LUO ET AL., 2010))

**Theorem 5** ($p$-Orthogonal Theorem (Luo et al., 2010))**.** *If $\mathbf{u}^{(l)}$ and $\mathbf{u}^{(r)}$ are two eigenvectors of $p$-Laplacian $\Delta_p$ associated with two different non-zero eigenvalues $\lambda_l$ and $\lambda_r$, $\mathbf{W}$ is symmetric and $p \ge 1$, then $\mathbf{u}^{(l)}$ and $\mathbf{u}^{(r)}$ are $p$-orthogonal up to the second order Taylor expansion.*

Thm. 5 implies that $\phi_p(\mathbf{u})^{(l)\top} \phi_p(\mathbf{u}^{(r)}) \approx 0$, for all $l, r = 0, \ldots, N - 1$ and $\lambda_l \ne \lambda_r$. Therefore, the space spanned by the multiple eigenvectors of the graph $p$-Laplacian is $p$-orthogonal.

### C.3  THEOREM 6 ($p$-EIGEN-DECOMPOSITION OF $\Delta_p$)

**Theorem 6** ($p$-Eigen-Decomposition of $\Delta_p$)**.** *Given the $p$-eigenvalues $\{\lambda_l \in \mathbb{R}\}_{l=0,1,\ldots,N-1}$, and the $p$-eigenvectors $\{\mathbf{u}^{(l)} \in \mathbb{R}^N\}_{l=0,1,\ldots,N-1}$ of $p$-Laplacian $\Delta_p$ and $\|\mathbf{u}^{(l)}\|_p = (\sum_{i=1}^N (u_i^{(l)})^p)^{1/p} = 1$, let $\mathbf{U}$ be a matrix of $p$-eigenvectors with $\mathbf{U} = (\mathbf{u}^{(0)}, \mathbf{u}^{(1)}, \ldots, \mathbf{u}^{(N-1)})$ and $\boldsymbol{\Lambda}$ be a diagonal matrix with $\boldsymbol{\Lambda} = \mathrm{diag}(\lambda_0, \lambda_1, \ldots, \lambda_{N-1})$, then the $p$-eigen-decomposition of $p$-Laplacian $\Delta_p$ is given by*

$$\Delta_p = \Phi_p(\mathbf{U}) \boldsymbol{\Lambda} \Phi_p(\mathbf{U})^\top.$$

*When $p = 2$, it reduces to the standard eigen-decomposition of the Laplacian matrix.*

Proof see Appendix D.5.

### C.4 THEOREM 7 (BOUNDS OF $p$-EIGENVALUES)

**Theorem 7** (Bounds of $p$-Eigenvalues). *Given a graph $\mathcal{G} = (\mathcal{V}, \mathcal{E}, \mathbf{W})$, if $\mathcal{G}$ is connected and $\lambda$ is a $p$-eigenvalue associated with the $p$-eigenvector $\mathbf{u}$ of $\Delta_p$, let $N_i$ denotes the number of edges connected to node $i$, $N_{min} = \min\{N_i\}_{i=1,2,...,N}$, and $k = \arg\max(\{|u_i|/\sqrt{D_{i,i}}\}_{i=1,2,...,N})$, then*

    *1. for $p \geq 2$, $0 \leq \lambda \leq 2^{p-1}$;*

    *2. for $1 < p < 2$, $0 \leq \lambda \leq 2^{p-1}\sqrt{N_k}$;*

    *3. for $p = 1$, $0 \leq \lambda \leq \sqrt{N_{min}}$.*

Proof see Appendix D.6.

## D PROOF OF THEOREMS

### D.1 PROOF OF THEOREM 1

*Proof.* Let $\boldsymbol{i}$ be the one-hot indicator vector whose $i$-th element is one and the other elements are zero. Then, we can obtain the personalized PageRank on node $i$, denoted as $\boldsymbol{\pi}_{\text{PPR}}(\boldsymbol{i})$, by using the recurrent equation (Klicpera et al., 2019):

$$\boldsymbol{\pi}_{\text{PPR}}^{(k+1)}(\boldsymbol{i}) = \alpha \mathbf{D}^{-1/2}\mathbf{W}\mathbf{D}^{-1/2}\boldsymbol{\pi}_{\text{PPR}}^{(k)}(\boldsymbol{i}) + \beta\boldsymbol{i},$$

where $k$ is the iteration step, $0 < \alpha < 1$ and $\beta = (1-\alpha)$ represents the restart probability. Without loss of generality, suppose $\boldsymbol{\pi}_{\text{PPR}}^{(0)}(\boldsymbol{i}) = \boldsymbol{i}$. Then we have,

$$\begin{aligned}
\boldsymbol{\pi}_{\text{PPR}}^{(k)}(\boldsymbol{i}) &= \alpha \mathbf{D}^{-1/2}\mathbf{W}\mathbf{D}^{-1/2}\boldsymbol{\pi}_{\text{PPR}}^{(k-1)}(\boldsymbol{i}) + \beta\boldsymbol{i} \\
&= \alpha \mathbf{D}^{-1/2}\mathbf{W}\mathbf{D}^{-1/2}\left(\alpha \boldsymbol{D}^{-1/2}\boldsymbol{W}\boldsymbol{D}^{-1/2}\boldsymbol{\pi}_{\text{PPR}}^{(k-2)}(\boldsymbol{i}) + \beta\boldsymbol{i}\right) + \beta\boldsymbol{i} \\
&= \left(\alpha \mathbf{D}^{-1/2}\mathbf{W}\mathbf{D}^{-1/2}\right)^2 \boldsymbol{\pi}_{\text{PPR}}^{(t-2)}(\boldsymbol{i}) + \beta\alpha \mathbf{D}^{-1/2}\mathbf{W}\mathbf{D}^{-1/2}\boldsymbol{i} + \beta\boldsymbol{i} \\
&= \left(\alpha \mathbf{D}^{-1/2}\mathbf{W}\mathbf{D}^{-1/2}\right)^k \boldsymbol{\pi}_{\text{PPR}}^{(0)}(\boldsymbol{i}) + \beta\sum_{t=0}^{k-1}\left(\alpha \mathbf{D}^{-1/2}\mathbf{W}\mathbf{D}^{-1/2}\right)^t \boldsymbol{i} \\
&= \left(\alpha \mathbf{D}^{-1/2}\mathbf{W}\mathbf{D}^{-1/2}\right)^k \boldsymbol{i} + \beta\sum_{t=0}^{k-1}\left(\alpha \mathbf{D}^{-1/2}\mathbf{W}\mathbf{D}^{-1/2}\right)^t \boldsymbol{i}
\end{aligned}$$

Since $0 < \alpha < 1$ and the eigenvalues of $\mathbf{D}^{-1/2}\mathbf{W}\mathbf{D}^{-1/2}$ in $[-1, 1]$, we have

$$\lim_{k\to\infty}\left(\alpha \mathbf{D}^{-1/2}\mathbf{W}\mathbf{D}^{-1/2}\right)^k = 0,$$

and we also have

$$\lim_{k\to\infty}\sum_{t=0}^{k-1}\left(\alpha \mathbf{D}^{-1/2}\mathbf{W}\mathbf{D}^{-1/2}\right)^t = \left(\mathbf{I}_N - \alpha \mathbf{D}^{-1/2}\mathbf{W}\mathbf{D}^{-1/2}\right)^{-1}.$$

Therefore,

$$\begin{aligned}
\boldsymbol{\pi}_{\text{PPR}}(\boldsymbol{i}) = \lim_{k\to\infty}\boldsymbol{\pi}_{\text{PPR}}^{(k)}(\boldsymbol{i}) &= \beta\left(\mathbf{I}_N - \alpha \mathbf{D}^{-1/2}\mathbf{W}\mathbf{D}^{-1/2}\right)^{-1}\boldsymbol{i} \\
&= \beta\left(\alpha\Delta_2 + (1-\alpha)\mathbf{I}_N\right)^{-1}\boldsymbol{i} \\
&= \mu(\Delta_2 + \mu\mathbf{I}_N)^{-1}\boldsymbol{i},
\end{aligned}$$

where we let $\alpha = \frac{1}{1+\mu}$ and $\beta = \frac{\mu}{1+\mu}$, $\mu > 0$. Then the fully personalized PageRank matrix can be obtained by substituting $\boldsymbol{i}$ with $\mathbf{I}_N$:

$$\boldsymbol{\Pi}_{\text{PPR}} = \mu(\Delta_2 + \mu\mathbf{I}_N)^{-1}.$$

$\square$

## D.2 PROOF OF THEOREM 2

*Proof.* By the definition of $\mathcal{L}_p(f)$ in Eq. (8), we have for some positive real value $\mu, \mu > 0$

$$\mathcal{L}_p(\mathbf{F}) = \frac{1}{2} \sum_{i=1}^{N} \sum_{j=1}^{N} \left\| \sqrt{\frac{W_{i,j}}{D_{i,i}}} \mathbf{F}_{i,:} - \sqrt{\frac{W_{i,j}}{D_{j,j}}} \mathbf{F}_{j,:} \right\|^p + \mu \sum_{i=1}^{N} \|\mathbf{F}_{i,:} - \mathbf{X}_{i,:}\|^2.$$

and by Eq. (12),

$$M_{i,j}^{(k)} := W_{i,j} \left\| \sqrt{\frac{W_{i,j}}{D_{i,i}}} \mathbf{F}_{i,:}^{(k)} - \sqrt{\frac{W_{i,j}}{D_{j,j}}} \mathbf{F}_{j,:}^{(k)} \right\|^{p-2}$$

Then, we have

$$
\begin{aligned}
\frac{\partial \mathcal{L}_p(\mathbf{F}^{(k)})}{\partial \mathbf{F}_{i,:}^{(k)}} &= p \sum_{j=1}^{N} \sqrt{\frac{W_{i,j}}{D_{i,i}}} \left\| \sqrt{\frac{W_{i,j}}{D_{i,i}}} \mathbf{F}_{i,:}^{(k)} - \sqrt{\frac{W_{i,j}}{D_{j,j}}} \mathbf{F}_{j,:}^{(k)} \right\|^{p-2} \left( \sqrt{\frac{W_{i,j}}{D_{i,i}}} \mathbf{F}_{i,:}^{(k)} - \sqrt{\frac{W_{i,j}}{D_{j,j}}} \mathbf{F}_{j,:}^{(k)} \right) + 2\mu(\mathbf{F}_{i,:}^{(k)} - \mathbf{X}_{i,:}) \\
&= p \left( \sum_{j=1}^{N} \frac{M_{i,j}^{(k)}}{D_{i,i}} \mathbf{F}_{i,:}^{(k)} - \sum_{j=1}^{N} \frac{M_{i,j}^{(k)}}{\sqrt{D_{i,i}D_{j,j}}} \mathbf{F}_{j,:}^{(k)} \right) + 2\mu(\mathbf{F}_{i,:}^{(k)} - \mathbf{X}_{i,:}) \\
&= p \left( \left( \sum_{j=1}^{N} \frac{M_{i,j}^{(k)}}{D_{i,i}} + \frac{2\mu}{p} \right) \mathbf{F}_{i,:}^{(k)} - \left( \sum_{j=1}^{N} \frac{M_{i,j}^{(k)}}{\sqrt{D_{i,i}D_{j,j}}} \mathbf{F}_{j,:}^{(k)} + \frac{2\mu}{p} \mathbf{X}_{i,:} \right) \right) \\
&= \frac{p}{\alpha_{i,i}^{(k)}} \left( \mathbf{F}_{i,:}^{(k)} - \left( \alpha_{i,i}^{(k)} \sum_{j=1}^{N} \frac{M_{i,j}^{(k)}}{\sqrt{D_{i,i}D_{j,j}}} \mathbf{F}_{j,:}^{(k)} + \beta_{i,i}^{(k)} \mathbf{X}_{i,:} \right) \right) \\
&= \frac{p}{\alpha_{i,i}^{(k)}} \left( \mathbf{F}_{i,:}^{(k)} - \mathbf{F}_{i,:}^{(k+1)} \right),
\end{aligned}
$$

which indicates that

$$\mathbf{F}_{i,:}^{(k)} - \mathbf{F}_{i,:}^{(k+1)} = \frac{\alpha_{i,i}^{(k)}}{p} \cdot \frac{\partial \mathcal{L}_p(\mathbf{F}^{(k)})}{\partial \mathbf{F}_{i,:}^{(k)}}.$$

For all $i, j \in [N], \mathbf{v} \in \mathbb{R}^{1 \times c}$, denote by

$$\partial \mathcal{L}_p(\mathbf{F}_{i,:}^{(k)}) := \frac{\partial \mathcal{L}_p(\mathbf{F}^{(k)})}{\partial \mathbf{F}_{i,:}^{(k)}},$$

$$M_{i,j}^{\prime(k)} := W_{i,j} \left\| \sqrt{\frac{W_{i,j}}{D_{i,i}}} (\mathbf{F}_{i,:}^{(k)} + \mathbf{v}) - \sqrt{\frac{W_{i,j}}{D_{j,j}}} \mathbf{F}_{j,:}^{(k)} \right\|^{p-2},$$

$$\alpha_{i,i}^{\prime(k)} := 1 \bigg/ \left( \sum_{j=1}^{N} \frac{M_{i,j}^{\prime(k)}}{D_{i,i}} + \frac{2\mu}{p} \right),$$

$$\beta_{i,i}^{\prime(k)} := \frac{2\mu}{p} \alpha_{i,i}^{\prime(k)}$$

$$\mathbf{F}_{i,:}^{\prime(k+1)} := \alpha_{i,i}^{\prime(k)} \sum_{j=1}^{N} \frac{M_{i,j}^{\prime(k)}}{\sqrt{D_{i,i}D_{j,j}}} \mathbf{F}_{j,:}^{(k)} + \beta_{i,i}^{\prime} \mathbf{X}_{i,:}.$$

Then

$$
\left\| \partial \mathcal{L}_p(\mathbf{F}_{i,:}^{(k)} + \mathbf{v}) - \partial \mathcal{L}_p(\mathbf{F}_{i,:}^{(k)}) \right\|
$$

$$
= \left\| \frac{p}{\alpha_{i,i}^{\prime(k)}} \left( \mathbf{F}_{i,:}^{(k)} + \mathbf{v} - \mathbf{F}_{i,:}^{\prime(k+1)} \right) - \frac{p}{\alpha_{i,i}^{(k)}} \left( \mathbf{F}_{i,:}^{(k)} - \mathbf{F}_{i,:}^{(k+1)} \right) \right\|
$$

$$
\leq \frac{p}{\alpha_{i,i}^{\prime(k)}} \|\mathbf{v}\| + \left\| \frac{p}{\alpha_{i,i}^{\prime(k)}} \left( \mathbf{F}_{i,:}^{(k)} - \mathbf{F}_{i,:}^{\prime(k)} \right) - \frac{p}{\alpha_{i,i}^{(k)}} \left( \mathbf{F}_{i,:}^{(k)} - \mathbf{F}_{i,:}^{(k+1)} \right) \right\|
$$

$$
= \frac{p}{\alpha_{i,i}^{\prime(k)}} \|\mathbf{v}\| + \left\| \left( \frac{p}{\alpha_{i,i}^{\prime(k)}} - \frac{p}{\alpha_{i,i}^{(k)}} \right) \mathbf{F}_{i,:}^{(k)} - \frac{p}{\alpha_{i,i}^{\prime(k)}} \mathbf{F}_{i,:}^{\prime(k+1)} + \frac{p}{\alpha_{i,i}^{(k)}} \mathbf{F}_{i,:}^{(k+1)} \right\|
$$

$$
= \frac{p}{\alpha_{i,i}^{\prime(k)}} \|\mathbf{v}\| + p \left\| \left( \sum_{j=1}^{N} \frac{M_{i,j}^{\prime(k)}}{D_{i,i}} - \sum_{j=1}^{N} \frac{M_{i,j}^{(k)}}{D_{i,i}} \right) \mathbf{F}_{i,:}^{(k)} - \sum_{j=1}^{N} \frac{M_{i,j}^{\prime(k)}}{\sqrt{D_{i,i}D_{j,j}}} \mathbf{F}_{j,:}^{(k)} - \frac{2\mu}{p} \mathbf{X}_{i,:} + \sum_{j=1}^{N} \frac{M_{i,j}^{(k)}}{\sqrt{D_{i,i}D_{j,j}}} \mathbf{F}_{j,:}^{(k)} + \frac{2\mu}{p} \mathbf{X}_{i,:} \right\|
$$

$$
= \frac{p}{\alpha_{i,i}^{\prime(k)}} \|\mathbf{v}\| + p \left\| \left( \sum_{j=1}^{N} \frac{M_{i,j}^{\prime(k)}}{D_{i,i}} - \sum_{j=1}^{N} \frac{M_{i,j}^{(k)}}{D_{i,i}} \right) \mathbf{F}_{i,:}^{(k)} - \sum_{j=1}^{N} \frac{M_{i,j}^{\prime(k)}}{\sqrt{D_{i,i}D_{j,j}}} \mathbf{F}_{j,:}^{(k)} + \sum_{j=1}^{N} \frac{M_{i,j}^{(k)}}{\sqrt{D_{i,i}D_{j,j}}} \mathbf{F}_{j,:}^{(k)} \right\|
$$

$$
= \left( p \sum_{j=1}^{N} \frac{M_{i,j}^{(k)}}{D_{i,i}} + 2\mu \right) \|\mathbf{v}\| + p \sum_{j=1}^{N} \frac{M_{i,j}^{\prime(k)} - M_{i,j}^{(k)}}{D_{i,i}} \|\mathbf{v}\| + p \left\| \sum_{j=1}^{N} \frac{M_{i,j}^{\prime(k)} - M_{i,j}^{(k)}}{D_{i,i}} \mathbf{F}_{i,:}^{(k)} - \sum_{j=1}^{N} \frac{M_{i,j}^{\prime(k)} - M_{i,j}^{(k)}}{\sqrt{D_{i,i}D_{j,j}}} \mathbf{F}_{j,:}^{(k)} \right\|
$$

$$
= p \left( \sum_{j=1}^{N} \frac{M_{i,j}^{(k)}}{D_{i,i}} + \frac{2\mu}{p} + o\left( p, \mathbf{v}, \mathbf{X}, \mathcal{G} \right) \right) \|\mathbf{v}\|.
$$

Therefore, there exists some real positive value $\mu \in o\left( p, \mathbf{v}, \mathbf{X}, \mathcal{G} \right) > 0$ such that

$$
\left\| \partial \mathcal{L}_p(\mathbf{F}_{i,:}^{(k)} + \mathbf{v}) - \partial \mathcal{L}_p(\mathbf{F}_{i,:}^{(k)}) \right\| \leq p \left( \sum_{j=1}^{N} \frac{M_{i,j}^{(k)}}{D_{i,i}} + \frac{2\mu}{p} \right) \|\mathbf{v}\| = \frac{p}{\alpha_{i,i}^{(k)}} \|\mathbf{v}\|. \tag{19}
$$

Let $\boldsymbol{\gamma} = (\gamma_1, \ldots, \gamma_N)^\top \in \mathbb{R}^N$ and $\boldsymbol{\eta} \in \mathbb{R}^{N \times c}$. By Taylor's theorem, we have:

$$
\mathcal{L}_p(\mathbf{F}_{i,:}^{(k)} + \gamma_i \boldsymbol{\eta}_{i,:})
$$

$$
= \mathcal{L}_p(\mathbf{F}_{i,:}^{(k)}) + \gamma_i \int_0^1 \langle \partial \mathcal{L}_p(\mathbf{F}_{i,:}^{(k)} + \epsilon \gamma_i \boldsymbol{\eta}_{i,:}), \boldsymbol{\eta}_{i,:} \rangle \mathrm{d}\epsilon
$$

$$
= \mathcal{L}_p(\mathbf{F}_{i,:}^{(k)}) + \gamma_i \langle \boldsymbol{\eta}_{i,:}, \partial \mathcal{L}_p(\mathbf{F}_{i,:}^{(k)}) \rangle + \gamma_i \int_0^1 \langle \partial \mathcal{L}_p(\mathbf{F}_{i,:}^{(k)} + \epsilon \gamma_i \boldsymbol{\eta}_{i,:}) - \partial \mathcal{L}_p(\mathbf{F}_{i,:}^{(k)}), \boldsymbol{\eta}_{i,:} \rangle \mathrm{d}\epsilon
$$

$$
\leq \mathcal{L}_p(\mathbf{F}_{i,:}^{(k)}) + \gamma_i \langle \boldsymbol{\eta}_{i,:}, \partial \mathcal{L}_p(\mathbf{F}_{i,:}^{(k)}) \rangle + \gamma_i \int_0^1 \|\partial \mathcal{L}_p(\mathbf{F}_{i,:}^{(k)} + \epsilon \gamma_i \boldsymbol{\eta}_{i,:}) - \partial \mathcal{L}_p(\mathbf{F}_{i,:}^{(k)})\| \|\boldsymbol{\eta}_{i,:}\| \mathrm{d}\epsilon
$$

$$
\leq \mathcal{L}_p(\mathbf{F}_{i,:}^{(k)}) + \gamma_i \langle \boldsymbol{\eta}_{i,:}, \partial \mathcal{L}_p(\mathbf{F}_{i,:}^{(k)}) \rangle + \frac{p}{2\alpha_{i,i}^{(k)}} \gamma_i^2 \|\boldsymbol{\eta}_{i,:}\|^2
$$

Let $\boldsymbol{\eta} = -\nabla \mathcal{L}_p(\mathbf{F}^{(k)})$ and choose some positive real value $\mu$ which depends on $\mathbf{X}, \mathcal{G}, p$ and $p > 1$, i.e. $\mu \in o\,(p, \mathbf{X}, \mathcal{G})$. By Eq. (19), we have for all $i \in [N]$,

$$\mathcal{L}_p(\mathbf{F}_{i,:}^{(k)} - \gamma \partial \mathcal{L}_p(\mathbf{F}_{i,:}^{(k)})) \leq \mathcal{L}_p(\mathbf{F}_{i,:}^{(k)}) - \langle \gamma_i \partial \mathcal{L}_p(\mathbf{F}_{i,:}^{(k)}), \partial \mathcal{L}_p(\mathbf{F}_{i,:}^{(k)}) \rangle + \frac{p}{2\alpha_{i,i}^{(k)}} \gamma_i^2 \|\partial \mathcal{L}_p(\mathbf{F}_{i,:}^{(k)})\|^2$$

$$= \mathcal{L}_p(\mathbf{F}_{i,:}^{(k)}) - \frac{p}{2\alpha_{i,i}^{(k)}} \left( \frac{2\alpha_{i,i}^{(k)} \gamma_i}{p} - \gamma_i^2 \right) \|\partial \mathcal{L}_p(\mathbf{F}_{i,:})\|^2$$

$$= \mathcal{L}_p(\mathbf{F}_{i,:}^{(k)}) - \frac{p}{2\alpha_{i,i}^{(k)}} \left( \frac{\left(\alpha_{i,i}^{(k)}\right)^2}{p^2} - \left(\gamma_i - \frac{\alpha_{i,i}^{(k)}}{p}\right)^2 \right) \|\partial \mathcal{L}_p(\mathbf{F}_{i,:}^{(k)})\|^2.$$

Then for all $i \in [N]$, when $0 \leq \gamma_i \leq \frac{2\alpha_{i,i}^{(k)}}{p}$, we have $\mathcal{S}_p(\mathbf{F}_{i,:}^{(k)} - \gamma_i \partial \mathcal{S}_p(\mathbf{F}_{i,:}^{(k)})) \leq \mathcal{S}_p(\mathbf{F}_{i,:}^{(k)})$ and $\gamma_i = \frac{\alpha_{i,i}^{(k)}}{p}$ minimizes $\mathcal{S}_p(\mathbf{F}_{i,:}^{(k)} - \gamma_i \partial \mathcal{S}_p(\mathbf{F}_{i,:}^{(k)}))$. Therefore,

$$\mathcal{L}_p(\mathbf{F}^{(k+1)}) = \mathcal{L}_p(\mathbf{F}^{(k)} - \frac{1}{p} \cdot \boldsymbol{\alpha}^{(k)} \nabla \mathcal{L}_p(\mathbf{F}^{(k)})) \leq \mathcal{L}_p(\mathbf{F}^{(k)}).$$

$\square$

### D.3 PROOF OF THEOREM 3

*Proof.* Without loss of generality, suppose $\mathbf{F}^{(0)} = \mathbf{X}$. Denote $\tilde{\mathbf{M}}^{(k)} = \mathbf{D}^{-1/2}\mathbf{M}^{(k)}\mathbf{D}^{-1/2}$, by Eq. (14), we have for $K \geq 2$,

$$\mathbf{F}^{(K)} = \boldsymbol{\alpha}^{(K-1)} \mathbf{D}^{-1/2} \mathbf{M}^{(K-1)} \mathbf{D}^{-1/2} \mathbf{F}^{(K-1)} + \boldsymbol{\beta}^{(K-1)} \mathbf{X}$$

$$= \boldsymbol{\alpha}^{(K-1)} \tilde{\mathbf{M}}^{K-1} \mathbf{F}^{(K-1)} + \boldsymbol{\beta}^{(K-1)} \mathbf{X}$$

$$= \boldsymbol{\alpha}^{(K-1)} \tilde{\mathbf{M}}^{K-1} \left( \boldsymbol{\alpha}^{(K-2)} \tilde{\mathbf{M}}^{K-2} \mathbf{F}^{(K-2)} + \boldsymbol{\beta}^{(K-2)} \mathbf{X} \right) + \boldsymbol{\beta}^{(K-1)} \mathbf{X}$$

$$= \boldsymbol{\alpha}^{(K-1)} \boldsymbol{\alpha}^{(K-2)} \tilde{\mathbf{M}}^{K-1} \tilde{\mathbf{M}}^{K-2} \mathbf{F}^{(K-2)} + \boldsymbol{\alpha}^{(K-1)} \tilde{\mathbf{M}}^{K-1} \boldsymbol{\beta}^{(K-2)} \mathbf{X} + \boldsymbol{\beta}^{(K-1)} \mathbf{X}$$

$$= \left( \prod_{k=0}^{K-1} \boldsymbol{\alpha}^{(k)} \right) \left( \prod_{k=0}^{K-1} \tilde{\mathbf{M}}^{(k)} \right) \mathbf{F}^{(0)} + \sum_{k=1}^{K-1} \left( \prod_{l=K-k}^{K-1} \boldsymbol{\alpha}^{(l)} \tilde{\mathbf{M}}^{(l)} \right) \boldsymbol{\beta}^{(K-1-k)} \mathbf{X} + \boldsymbol{\beta}^{(K-1)} \mathbf{X}$$

$$= \left( \prod_{k=0}^{K-1} \boldsymbol{\alpha}^{(k)} \right) \left( \prod_{k=0}^{K-1} \tilde{\mathbf{M}}^{(k)} \right) \mathbf{X} + \sum_{k=1}^{K-1} \left( \prod_{l=K-k}^{K-1} \boldsymbol{\alpha}^{(l)} \tilde{\mathbf{M}}^{(l)} \right) \boldsymbol{\beta}^{(K-1-k)} \mathbf{X} + \boldsymbol{\beta}^{(K-1)} \mathbf{X}.$$

(20)

Recall Equations (12) and (13), we have

$$\tilde{M}_{i,j}^{(k)} = \frac{W_{i,j}}{\sqrt{D_{i,i}D_{j,j}}} \left\| \sqrt{\frac{W_{i,j}}{D_{i,i}}} \mathbf{F}_{i,:}^{(k)} - \sqrt{\frac{W_{i,j}}{D_{j,j}}} \mathbf{F}_{j,:}^{(k)} \right\|^{p-2}, \quad \text{for all } i, j = 1, 2, \ldots, N,$$

and

$$\alpha_{i,i}^{(k)} = \frac{1}{\sum_{j=1}^{N} \frac{M_{i,j}^{(k)}}{D_{i,i}} + \frac{2\mu}{p}}, \quad \text{for all } i = 1, 2, \ldots, N.$$

Note that the eigenvalues of $\tilde{\mathbf{M}}$ are not infinity and $0 < \alpha_{i,i} < 1$ for all $i = 1, \ldots, N$. Then we have

$$\lim_{K \to \infty} \prod_{k=0}^{K-1} \boldsymbol{\alpha}^{(k)} = 0,$$

and

$$\lim_{K \to \infty} \left( \prod_{k=0}^{K-1} \boldsymbol{\alpha}^{(k)} \right) \left( \prod_{k=0}^{K-1} \tilde{\mathbf{M}}^{(k)} \right) = 0.$$

Therefore,

$$\lim_{K\to\infty} \mathbf{F}^{(K)} = \lim_{K\to\infty} \left( \sum_{k=1}^{K-1} \left( \prod_{l=K-k}^{K-1} \boldsymbol{\alpha}^{(l)} \tilde{\mathbf{M}}^{(l)} \right) \boldsymbol{\beta}^{(K-1-k)} \mathbf{X} + \boldsymbol{\beta}^{(K-1)} \mathbf{X} \right). \tag{21}$$

By Equations (6) and (12), we have

$$\Delta_p f(i) = \sum_{j=1}^{N} \frac{W_{i,j}}{D_{i,i}} \|(\nabla f)([j,i])\|^{p-2} f(i) - \sum_{j=1}^{N} \frac{W_{i,j}}{\sqrt{D_{i,i} D_{j,j}}} \|(\nabla f)([j,i])\|^{p-2} f(j)$$

$$= \sum_{j=1}^{N} \frac{M_{i,j}}{D_{i,i}} f(i) - \sum_{j=1}^{N} \frac{M_{i,j}}{\sqrt{D_{i,i} D_{j,j}}} f(j). \tag{22}$$

By Eq. (13), we have

$$\sum_{j=1}^{N} \frac{M_{i,j}^{(k)}}{D_{i,i}} = \frac{1}{\alpha_{i,i}^{(k)}} - \frac{2\mu}{p}. \tag{23}$$

Equations (22) and (23) show that

$$\Delta_p^{(k)} = \left( \left( \boldsymbol{\alpha}^{(k)} \right)^{-1} - \frac{2\mu}{p} \mathbf{I}_N \right) - \tilde{\mathbf{M}}^{(k)}, \tag{24}$$

which indicates

$$\boldsymbol{\alpha}^{(k)} \tilde{\mathbf{M}}^{(k)} = \mathbf{I}_N - \frac{2\mu}{p} \boldsymbol{\alpha}^{(k)} - \boldsymbol{\alpha}^{(k)} \Delta_p^{(k)}. \tag{25}$$

Eq. (25) shows that $\boldsymbol{\alpha}^{(k)} \tilde{\mathbf{M}}^{(k)}$ is linear w.r.t $\Delta_p$ and therefore can be expressed by a linear combination in terms of $\Delta_p$:

$$\boldsymbol{\alpha}^{(k)} \tilde{\mathbf{M}}^{(k)} = \boldsymbol{\theta}'^{(k)} \Delta_p, \tag{26}$$

where $\boldsymbol{\theta}' = \mathrm{diag}(\theta_0', \theta_1', \ldots, \theta_{N-1}')$ are the parameters. Therefore, we have

$$\lim_{K\to\infty} \mathbf{F}^{(K)} = \lim_{K\to\infty} \left( \sum_{k=1}^{K-1} \left( \prod_{l=K-k}^{K-1} \boldsymbol{\alpha}^{(l)} \tilde{\mathbf{M}}^{(l)} \right) \boldsymbol{\beta}^{(K-1-k)} \mathbf{X} + \boldsymbol{\beta}^{(K-1)} \mathbf{X} \right)$$

$$= \lim_{K\to\infty} \left( \sum_{k=1}^{K-1} \left( \prod_{l=K-k}^{K-1} \boldsymbol{\theta}'^{(l)} \Delta_p \right) \boldsymbol{\beta}^{(K-1-k)} \mathbf{X} + \boldsymbol{\beta}^{(K-1)} \mathbf{X} \right)$$

$$= \lim_{K\to\infty} \left( \sum_{k=1}^{K-1} \boldsymbol{\beta}^{(K-1-k)} \left( \prod_{l=K-k}^{K-1} \boldsymbol{\theta}'^{(l)} \right) \Delta_p^k \mathbf{X} + \boldsymbol{\beta}^{(K-1)} \mathbf{X} \right)$$

$$= \lim_{K\to\infty} \sum_{k=0}^{K-1} \boldsymbol{\theta}''^{(k)} \Delta_p^k \mathbf{X},$$

where $\boldsymbol{\theta}''^{(k)} = \mathrm{diag}(\theta_1''^{(k)}, \theta_2''^{(k)}, \ldots, \theta_N''^{(k)})$ defined as $\boldsymbol{\theta}''^{(0)} = \boldsymbol{\beta}^{(K-1)}$ and

$$\theta_i''^{(k)} = \beta_{i,i}^{(K-1-k)} \prod_{l=K-k}^{K-1} \theta_i'^{(l)}, \text{ for } k = 1, 2, \ldots, K-1.$$

Let $\boldsymbol{\theta} = (\theta_0, \theta_1, \ldots, \theta_{K-1})$ defined as $\theta_k = \sum_{i=1}^{N} \theta_i''^{(k)}$ for all $k = 0, 1, \ldots, K-1$, then

$$\lim_{K\to\infty} \mathbf{F}^{(K)} = \lim_{K\to\infty} \left( \sum_{k=1}^{K-1} \theta_k \Delta_p^k \mathbf{X} + \theta_0 \mathbf{X} \right)$$

$$= \lim_{K\to\infty} \sum_{k=0}^{K-1} \theta_k \Delta_p^k \mathbf{X}.$$

Therefore complete the proof. □

### D.4 PROOF OF THEOREM 4

*Proof.* The first-order Taylor expansion with Peano's form of remainder for $\sigma$ at $\mathbf{X}_{i,:}^*$ is given by:

$$\sigma(\mathbf{F}_{j,:}^{(K-1)}) = \sigma(\mathbf{X}_{i,:}^*) + \frac{\partial\sigma(\mathbf{X}_{i,:}^*)}{\partial\mathbf{X}}\left(\mathbf{F}_{j,:}^{(K-1)} - \mathbf{X}_{i,:}^*\right)^\top + o(\|\mathbf{F}_{j,:}^{(K-1)} - \mathbf{X}_{i,:}^*\|).$$

Note that in general the output non-linear layer $\sigma(\cdot)$ is simple. Here we assume that it can be well approximated by the first-order Taylor expansion and we can ignore the Peano's form of remainder. For all $i = 1, \ldots, N$, $D_{i,i} = D_{j,j} = d$, we have $\alpha_{i,i}\sum_{j=1}^N D_{i,i}^{-1/2}M_{i,j}^{(K-1)}D_{j,j}^{-1/2} + \beta_{i,i}^{(K-1)} = 1$. Then

$$\left|y_i - \tilde{y}_i^{(K)}\right|$$

$$= \left|y_i - \alpha_{i,i}^{(K-1)}\sum_{j=1}^N \frac{M_{i,j}^{(K-1)}}{\sqrt{D_{i,i}D_{j,j}}}\sigma\left(\mathbf{F}_{j,:}^{K-1}\right) - \beta_{i,i}^{(K-1)}\sigma\left(\mathbf{X}_{i,:}\right)\right|$$

$$= \left|y_i - \beta_{i,i}^{(K-1)}\sigma\left(\mathbf{X}_{i,:}\right) - \alpha_{i,i}^{(K-1)}\sum_{j=1}^N \frac{M_{i,j}^{(K-1)}}{\sqrt{D_{i,i}D_{j,j}}}\left(\sigma\left(\mathbf{X}_{i,:}^*\right) + \frac{\partial\sigma\left(\mathbf{X}_{i,:}^*\right)}{\partial\mathbf{X}}\left(\mathbf{F}_{j,:}^{(K-1)} - \mathbf{X}_{i,:}^*\right)^\top\right)\right|$$

$$= \left|y_i - \alpha_{i,i}^{(K-1)}\sum_{j=1}^N \frac{M_{i,j}^{(K-1)}}{\sqrt{D_{i,i}D_{j,j}}}y_i - \beta_{i,i}^{(K-1)}\sigma(\mathbf{X}_{i,:}) - \alpha_{i,i}^{(K-1)}\sum_{j=1}^N \frac{M_{i,j}^{(K-1)}}{\sqrt{D_{i,i}D_{j,j}}}\left(\frac{\partial\sigma(\mathbf{X}_{i,:}^*)}{\partial\mathbf{X}}\left(\mathbf{F}_{j,:}^{(K-1)} - \mathbf{X}_{i,:}^*\right)^\top\right)\right|$$

$$= \left|\beta_{i,i}^{(K-1)}(y_i - \sigma(\mathbf{X}_{i,:})) - \alpha_{i,i}^{(K-1)}\sum_{j=1}^N \frac{M_{i,j}^{(K-1)}}{\sqrt{D_{i,i}D_{j,j}}}\left(\frac{\partial\sigma(\mathbf{X}_{i,:}^*)}{\partial\mathbf{X}}\left(\mathbf{F}_{j,:}^{(K-1)} - \mathbf{X}_{i,:} - \boldsymbol{\epsilon}_{i,:}\right)^\top\right)\right|$$

$$\leq \beta_{i,i}^{(K-1)}|y_i - \sigma(\mathbf{X}_{i,:})| + \alpha_{i,i}^{(K-1)}\left|\sum_{j=1}^N \frac{M_{i,j}^{(K-1)}}{\sqrt{D_{i,i}D_{j,j}}}\frac{\partial\sigma(\mathbf{X}_{i,:}^*)}{\partial\mathbf{X}}\left(\mathbf{F}_{j,:}^{(K-1)} - \mathbf{X}_{i,:}\right)^\top\right| + (1 - \beta_{i,i}^{(K-1)})\left|\frac{\partial\sigma(\mathbf{X}_{i,:}^*)}{\partial\mathbf{X}}\boldsymbol{\epsilon}_{i,:}^\top\right|$$

$$\leq \beta_{i,i}^{(K-1)}|y_i - \sigma(\mathbf{X}_{i,:})| + \alpha_{i,i}^{(K-1)}\left\|\frac{\partial\sigma(\mathbf{X}_{i,:}^*)}{\partial\mathbf{X}}\right\|\left\|\sum_{j=1}^N \frac{M_{i,j}^{(K-1)}}{\sqrt{D_{i,i}D_{j,j}}}\left(\mathbf{F}_{j,:}^{(K-1)} - \mathbf{X}_{i,:}\right)\right\| + (1 - \beta_{i,i}^{(K-1)})\left\|\frac{\partial\sigma(\mathbf{X}_{i,:}^*)}{\partial\mathbf{X}}\right\|\|\boldsymbol{\epsilon}_{i,:}\|$$

$$\leq \beta_{i,i}^{(K-1)}|y_i - \sigma(\mathbf{X}_{i,:})| + \alpha_{i,i}^{(K-1)}L\left\|\sum_{j=1}^N \frac{M_{i,j}^{(K-1)}}{d}\left(\mathbf{F}_{j,:}^{(K-1)} - \mathbf{X}_{i,:}\right)\right\| + (1 - \beta_{i,i}^{(K-1)})L\|\boldsymbol{\epsilon}_{i,:}\|$$

$$= \beta_{i,i}^{(K-1)}|y_i - \sigma(\mathbf{X}_{i,:})| + \alpha_{i,i}^{(K-1)}L\left\|\sum_{j=1}^N \frac{M_{i,j}^{(K-1)}}{d}\left(\mathbf{F}_{j,:}^{(K-1)} - \mathbf{F}_{i,:}^{(K-1)} + \mathbf{F}_{i,:}^{(K-1)} - \mathbf{X}_{i,:}\right)\right\|$$
$$+ (1 - \beta_{i,i}^{(K-1)})L\|\boldsymbol{\epsilon}_{i,:}\|$$

$$= \beta_{i,i}^{(K-1)}|y_i - \sigma(\mathbf{X}_{i,:})| + \alpha_{i,i}^{(K-1)}L\left\|\Delta_p\mathbf{F}_{i,:}^{(K-1)} + \sum_{j=1}^N \frac{M_{i,j}^{(K-1)}}{d}\cdot\sum_{j=1}^N \frac{M_{i,j}^{(K-2)}}{d}\left(\mathbf{F}_{j,:}^{(K-2)} - \mathbf{X}_{i,:}\right)\right\|$$
$$+ (1 - \beta_{i,i}^{(K-1)})L\|\boldsymbol{\epsilon}_{i,:}\|$$

$$= \beta_{i,i}^{(K-1)}|y_i - \sigma(\mathbf{X}_{i,:})| + \alpha_{i,i}^{(K-1)}L\left\|\Delta_p^{(K-1)}\mathbf{F}_{i,:}^{(K-1)} + \sum_{k=0}^{K-2}\prod_{l=k}^{K-2}\left(\sum_{j=1}^N \frac{M_{i,j}^{(l)}}{d}\right)\Delta_p^{(k)}\mathbf{X}_{i,:}\right\| + (1 - \beta_{i,i}^{(K-1)})L\|\boldsymbol{\epsilon}_{i,:}\|$$

$$= \beta_{i,i}^{(K-1)}|y_i - \sigma(\mathbf{X}_{i,:})| + \alpha_{i,i}^{(K-1)}L\left\|\Delta_p^{(K-1)}\mathbf{F}_{i,:}^{(K-1)} + \sum_{k=0}^{K-2}\prod_{l=k}^{K-2}\left(\sum_{j=1}^N \frac{M_{i,j}^{(l)}}{d}\right)\Delta_p^{(k)}\mathbf{X}_{i,:}\right\| + (1 - \beta_{i,i}^{(K-1)})L\|\boldsymbol{\epsilon}_{i,:}\|$$

Therefore,

$$
\begin{aligned}
\frac{1}{N}\sum_{i=1}^{N}|y_i - \tilde{y}_i| \leq {} & \frac{1}{N}\sum_{i=1}^{N}\beta_{i,i}^{(K-1)}|y_i - \sigma(\mathbf{X}_{i,:})| \\
& + \frac{L}{N}\sum_{i=1}^{N}\alpha_{i,i}^{(K-1)}\left\|\Delta_p^{(K-1)}\mathbf{F}_{i,:}^{(K-1)} + \sum_{k=0}^{K-2}\prod_{l=k}^{K-2}\left(\sum_{j=1}^{N}\frac{M_{i,j}^{(l)}}{d}\right)\Delta_p^{(k)}\mathbf{X}_{i,:}\right\| \\
& + \frac{L}{N}\sum_{i=1}^{N}(1-\beta_{i,i}^{(K-1)})\|\boldsymbol{\epsilon}_{i,:}\|
\end{aligned}
$$

$\square$

## D.5 PROOF OF THEOREM 6

*Proof.* Note that

$$
\phi_p(\mathbf{u})^\top\mathbf{u} = \sum_{i=1}^{N}\phi_p(u_i)u_i = \sum_{i=1}^{N}\|u_i\|^{p-2}u_i^2 = \sum_{i=1}^{N}\|u_i\|^p = \sum_{i=1}^{N}|u_i|^p = \|\mathbf{u}\|_p^p = 1,
$$

then we have

$$
\Delta_p\mathbf{U} = \Phi_p(\mathbf{U})\boldsymbol{\Lambda} = \Phi_p(\mathbf{U})\boldsymbol{\Lambda}\Phi(\mathbf{U})^\top\mathbf{U}.
$$

Therefore, $\Delta_p = \Phi_p(\mathbf{U})\boldsymbol{\Lambda}\Phi_p(\mathbf{U})^\top$.

When $p=2$, by $\Phi_2(\mathbf{U}) = \mathbf{U}$, we get $\Delta_2 = \Phi_2(\mathbf{U})\boldsymbol{\Lambda}\Phi_2(\mathbf{U})^\top = \mathbf{U}\boldsymbol{\Lambda}\mathbf{U}^\top$. $\square$

## D.6 PROOF OF THEOREM 7

*Proof.* By the definition of graph $p$-Laplacian, we have for all $i=1,2,\ldots,N$,

$$
(\Delta_p\mathbf{u})_i = \sum_{j=1}^{N}\sqrt{\frac{W_{i,j}}{D_{i,i}}}\left\|\sqrt{\frac{W_{i,j}}{D_{i,i}}}u_i - \sqrt{\frac{W_{i,j}}{D_{j,j}}}u_j\right\|^{p-2}\left(\sqrt{\frac{W_{i,j}}{D_{i,i}}}u_i - \sqrt{\frac{W_{i,j}}{D_{j,j}}}u_j\right) = \lambda\phi_p(u_i).
$$

Then, for all $i=1,2,\ldots,N$,

$$
\begin{aligned}
\lambda &= \frac{1}{\phi_p(u_i)}\sum_{j=1}^{N}\sqrt{\frac{W_{i,j}}{D_{i,i}}}\left\|\sqrt{\frac{W_{i,j}}{D_{i,i}}}u_i - \sqrt{\frac{W_{i,j}}{D_{j,j}}}u_j\right\|^{p-2}\left(\sqrt{\frac{W_{i,j}}{D_{i,i}}}u_i - \sqrt{\frac{W_{i,j}}{D_{j,j}}}u_j\right) \\
&= \frac{1}{\|u_i\|^{p-2}u_i}\sum_{j=1}^{N}\sqrt{\frac{W_{i,j}}{D_{i,i}}}\left\|\sqrt{\frac{W_{i,j}}{D_{i,i}}}u_i - \sqrt{\frac{W_{i,j}}{D_{j,j}}}u_j\right\|^{p-2}\left(\sqrt{\frac{W_{i,j}}{D_{i,i}}}u_i - \sqrt{\frac{W_{i,j}}{D_{j,j}}}u_j\right) \\
&= \sum_{j=1}^{N}\sqrt{\frac{W_{i,j}}{D_{i,i}}}\frac{\left\|\sqrt{\frac{W_{i,j}}{D_{i,i}}}u_i - \sqrt{\frac{W_{i,j}}{D_{j,j}}}u_j\right\|^{p-2}}{\|u_i\|^{p-2}}\left(\sqrt{\frac{W_{i,j}}{D_{i,i}}} - \sqrt{\frac{W_{i,j}}{D_{j,j}}}\frac{u_j}{u_i}\right) \\
&= \sum_{j=1}^{N}\left(\frac{W_{i,j}}{D_{i,i}} - \frac{W_{i,j}}{\sqrt{D_{i,i}D_{j,j}}}\frac{u_j}{u_i}\right)\left(\frac{\left\|\sqrt{\frac{W_{i,j}}{D_{i,i}}}u_i - \sqrt{\frac{W_{i,j}}{D_{j,j}}}u_j\right\|}{\|u_i\|}\right)^{p-2} \\
&= \sum_{j=1}^{N}\left(\frac{W_{i,j}}{D_{i,i}} - \frac{W_{i,j}}{\sqrt{D_{i,i}D_{j,j}}}\frac{u_j}{u_i}\right)\left\|\frac{\sqrt{\frac{W_{i,j}}{D_{i,i}}}u_i - \sqrt{\frac{W_{i,j}}{D_{j,j}}}u_j}{u_i}\right\|^{p-2} \\
&= \sum_{j=1}^{N}\left(\frac{W_{i,j}}{D_{i,i}} - \frac{W_{i,j}}{\sqrt{D_{i,i}D_{j,j}}}\frac{u_j}{u_i}\right)\left\|\sqrt{\frac{W_{i,j}}{D_{i,i}}} - \sqrt{\frac{W_{i,j}}{D_{j,j}}}\frac{u_j}{u_i}\right\|^{p-2}
\end{aligned}
$$

Let $l = \arg\max\{\|u_i\|\}_{i=1,2,\ldots,N}$, the above equation holds for all $i = 1, 2, \ldots, N$, then

$$\lambda = \sum_{j=1}^{N} \left( \frac{W_{l,j}}{D_{l,l}} - \frac{W_{l,j}}{\sqrt{D_{l,l}D_{j,j}}} \frac{u_j}{u_l} \right) \left\| \sqrt{\frac{W_{l,j}}{D_{l,l}}} - \sqrt{\frac{W_{l,j}}{D_{j,j}}} \frac{u_j}{u_l} \right\|^{p-2}$$

$$\geq \sum_{j=1}^{N} \left( \frac{W_{l,j}}{D_{l,l}} - \frac{W_{l,j}}{\sqrt{D_{l,l}D_{j,j}}} \left| \frac{u_j}{u_l} \right| \right) \left\| \sqrt{\frac{W_{l,j}}{D_{l,l}}} - \sqrt{\frac{W_{l,j}}{D_{j,j}}} \frac{u_j}{u_l} \right\|^{p-2}$$

$$\geq \sum_{j=1}^{N} \left( \frac{W_{l,j}}{D_{l,l}} - \frac{W_{l,j}}{\sqrt{D_{l,l}D_{j,j}}} \right) \left\| \sqrt{\frac{W_{l,j}}{D_{l,l}}} - \sqrt{\frac{W_{l,j}}{D_{j,j}}} \frac{u_j}{u_l} \right\|^{p-2}$$

$$\geq 0.$$

When $p = 1$,

$$\lambda = \sum_{j=1}^{N} \left( \frac{W_{i,j}}{D_{i,i}} - \frac{W_{i,j}}{\sqrt{D_{i,i}D_{j,j}}} \frac{u_j}{u_i} \right) \left\| \sqrt{\frac{W_{i,j}}{D_{i,i}}} - \sqrt{\frac{W_{i,j}}{D_{j,j}}} \frac{u_j}{u_i} \right\|^{-1}$$

$$\leq \sum_{j=1}^{N} \sqrt{\frac{W_{i,j}}{D_{i,i}}} \leq \sqrt{N_i \sum_{j=1}^{N} \frac{W_{i,j}}{D_{i,i}}} = \sqrt{N_i},$$

where the last inequality holds by using the Cauchy-Schwarz inequality. The above inequality holds for all $i = 1, 2, \ldots, N$, therefore,

$$\lambda \leq \sum_{j=1}^{N} \sqrt{\frac{W_{i,j}}{D_{i,i}}} \leq \sqrt{N_{min}}.$$

When $p > 1$, we have for $i = 1, 2, \ldots, N$,

$$\lambda = \sum_{j=1}^{N} \left( \frac{W_{i,j}}{D_{i,i}} - \frac{W_{i,j}}{\sqrt{D_{i,i}D_{j,j}}} \frac{u_j}{u_i} \right) \left\| \sqrt{\frac{W_{i,j}}{D_{i,i}}} - \sqrt{\frac{W_{i,j}}{D_{j,j}}} \frac{u_j}{u_i} \right\|^{p-2}$$

$$\leq \sum_{j=1}^{N} \sqrt{\frac{W_{i,j}}{D_{i,i}}} \left\| \sqrt{\frac{W_{i,j}}{D_{i,i}}} - \sqrt{\frac{W_{i,j}}{D_{j,j}}} \frac{u_j}{u_i} \right\|^{p-1}$$

$$\leq \sum_{j=1}^{N} \sqrt{\frac{W_{i,j}}{D_{i,i}}} \left\| \sqrt{\frac{W_{i,j}}{D_{i,i}}} + \sqrt{\frac{W_{i,j}}{D_{j,j}}} \left| \frac{u_j}{u_i} \right| \right\|^{p-1}$$

$$= \sum_{j=1}^{N} \left( \sqrt{\frac{W_{i,j}}{D_{i,i}}} \right)^p \left\| 1 + \sqrt{\frac{D_{i,i}}{D_{j,j}}} \left| \frac{u_j}{u_i} \right| \right\|^{p-1}.$$

Without loss of generality, let $k = \arg\max(\{|u_i|/\sqrt{D_{i,i}}\}_{i=1,2,\ldots,N})$. Because the above inequality holds for all $i = 1, 2, \ldots, N$, then we have

$$\lambda \leq \sum_{j=1}^{N} \left( \sqrt{\frac{W_{k,j}}{D_{k,k}}} \right)^p \left( 1 + \sqrt{\frac{D_{k,k}}{D_{j,j}}} \left| \frac{u_j}{u_k} \right| \right)^{p-1}$$

$$\leq 2^{p-1} \sum_{j=1}^{N} \left( \sqrt{\frac{W_{k,j}}{D_{k,k}}} \right)^p.$$

For $p \geq 2$,

$$\lambda \leq 2^{p-1} \sum_{j=1}^{N} \left( \sqrt{\frac{W_{k,j}}{D_{k,k}}} \right)^p \leq 2^{p-1} \sum_{j=1}^{N} \left( \sqrt{\frac{W_{k,j}}{D_{k,k}}} \right)^2 = 2^{p-1}.$$

For $1 < p < 2$,

$$\lambda \le 2^{p-1} \sum_{j=1}^{N} \left( \sqrt{\frac{W_{k,j}}{D_{k,k}}} \right)^p \le 2^{p-1} \sum_{j=1}^{N} \sqrt{\frac{W_{k,j}}{D_{k,k}}} \le 2^{p-1} \sqrt{N_k \sum_{j=1}^{N} \frac{W_{k,j}}{D_{k,k}}} = 2^{p-1} \sqrt{N_k}.$$

□

### D.7 PROOF OF PROPOSITION 1

*Proof.* We proof Proposition 1 based on the bounds of $p$-eigenvalues as demonstrated in Thm. 7. By Eq. (6) and Eq. (12), we have

$$\Delta_p f(i) = \sum_{j=1}^{N} \frac{W_{i,j}}{D_{i,i}} \|(\nabla f)([j,i])\|^{p-2} f(i) - \sum_{j=1}^{N} \frac{W_{i,j}}{\sqrt{D_{i,i}D_{j,j}}} \|(\nabla f)([j,i])\|^{p-2} f(j)$$

$$= \sum_{j=1}^{N} \frac{M_{i,j}}{D_{i,i}} f(i) - \sum_{j=1}^{N} \frac{M_{i,j}}{\sqrt{D_{i,i}D_{j,j}}} f(j). \tag{27}$$

By Eq. (13), we have

$$\sum_{j=1}^{N} \frac{M_{i,j}^{(k)}}{D_{i,i}} = \frac{1}{\alpha_{i,i}^{(k)}} - \frac{2\mu}{p}. \tag{28}$$

Equations (27) and (28) show that

$$\Delta_p^{(k)} = \left( \left( \boldsymbol{\alpha}^{(k)} \right)^{-1} - \frac{2\mu}{p} \mathbf{I}_N \right) - \mathbf{D}^{-1/2} \mathbf{M}^{(k)} \mathbf{D}^{-1/2}, \tag{29}$$

which indicates

$$\boldsymbol{\alpha}^{(k)} \mathbf{D}^{-1/2} \mathbf{M}^{(k)} \mathbf{D}^{-1/2} = \mathbf{I}_N - \frac{2\mu}{p} \boldsymbol{\alpha}^{(k)} - \boldsymbol{\alpha}^{(k)} \Delta_p^{(k)}. \tag{30}$$

For $i = 1, 2, \ldots, N$, let $\tilde{\boldsymbol{\alpha}} := (\tilde{\alpha}_1, \ldots, \tilde{\alpha}_N)$, $\tilde{\alpha}_i := 1/\sum_{j=1}^{N} \frac{M_{i,j}}{D_{i,i}}$, then

$$\alpha_{i,i} \sum_{j=1}^{N} \frac{M_{i,j}}{\sqrt{D_{i,i}D_{j,j}}} = (1 - \frac{2\mu}{p} \alpha_{i,i}) - \alpha_{i,i} \lambda_i$$

$$= \frac{\sum_{j=1}^{N} \frac{M_{i,j}}{D_{i,i}}}{\sum_{j=1}^{N} \frac{M_{i,j}}{D_{i,i}} + \frac{2\mu}{p}} \left( 1 - \frac{1}{\sum_{j=1}^{N} \frac{M_{i,j}}{D_{i,i}}} \lambda_i \right)$$

$$= \frac{1}{1 + \frac{2\mu\tilde{\alpha}_i}{p}} (1 - \tilde{\alpha}_i \lambda_i), \tag{31}$$

Recall the Eq. (12) that

$$M_{i,j} = W_{i,j} \left\| \sqrt{\frac{W_{i,j}}{D_{i,i}}} \mathbf{F}_{i,:} - \sqrt{\frac{W_{i,j}}{D_{j,j}}} \mathbf{F}_{j,:} \right\|^{p-2} = W_{i,j} \|(\nabla f)([i,j])\|^{p-2},$$

1. When $p = 2$, for all $i = 1, \ldots, N$, $\tilde{\alpha}_i = 1$ and $0 \le \lambda_{i-1} \le 2$, $g_2(\lambda_{i-1})$ works as both low-pass and high-pass filters.

2. When $p > 2$, by Thm. 7 we have for all $i = 1, \ldots, N$, $0 \le \lambda_{i-1} \le 2^{p-1}$. If $0 \le \tilde{\alpha}_i \le 2^{1-p}$, then $0 \le 1 - \tilde{\alpha}_i \lambda_i \le 1$, which indicates that $g_p(\lambda_{i-1})$ works as a low-pass filter; If

$\tilde{\alpha}_i > 2^{1-p}$, then $g_p(\lambda_{i-1})$ works as both low-pass and high-pass filters. Since

$$\sum_{j=1}^{N} \frac{M_{i,j}}{D_{i,i}} = \sum_{j=1}^{N} \frac{W_{i,j}\|(\nabla f)([i,j])\|^{p-2}}{D_{i,i}}$$

$$\leq \sqrt{\sum_{j=1}^{N} \left(\frac{W_{i,j}}{D_{i,i}}\right)^2 \sum_{j=1}^{N} \|(\nabla f)([i,j])\|^{2(p-2)}}$$

$$\leq \sqrt{\sum_{j=1}^{N} \|(\nabla f)([i,j])\|^{2(p-2)}}$$

$$\leq \|\nabla f(i)\|^{p-2},$$

which indicates that $\tilde{\alpha}_i \geq \|\nabla f(i)\|^{2-p}$. $0 \leq \tilde{\alpha}_i \leq 2^{1-p}$ directly implies that $0 \leq \|\nabla f(i)\|^{2-p} \leq 2^{1-p}$, i.e. $\|\nabla f(i)\| \geq 2^{(p-1)/(p-2)}$ and when $\|\nabla f(i)\|^{2-p} \geq 2^{1-p}$, i.e. $\|\nabla f(i)\| \leq 2^{(p-1)/(p-2)}$, $\tilde{\alpha}_i \geq 2^{1-p}$ always holds. Therefore, if $\|\nabla f(i)\| \leq 2^{(p-1)/(p-2)}$, $g_p(\lambda_{i-1})$ works as both low-pass and high-pass filters on node $i$; If $g_p(\lambda_{i-1})$ works as a low-pass filter, $\|\nabla f(i)\| \geq 2^{(p-1)/(p-2)}$.

3. When $1 \leq p < 2$, by Thm. 7 we have for all $i = 1, \ldots, N$, $0 \leq \lambda_{i-1} \leq 2^{p-1}\sqrt{N_k}$. If $0 \leq \tilde{\alpha}_i \leq 2^{1-p}/\sqrt{N_k}$, $0 \leq 1 - \tilde{\alpha}_i\lambda_i \leq 1$, which indicates that $g_p(\lambda_{i-1})$ work as low-pass filters; If $\tilde{\alpha}_i \geq 2^{1-p}/\sqrt{N_k}$, $g_p(\lambda_{i-1})$ work as both low-pass and high-pass filters. By

$$\sum_{j=1}^{N} \frac{W_{i,j}}{D_{i,i}} \frac{1}{\|(\nabla f)([i,j])\|^{p-2}} \Bigg/ \frac{1}{\sum_{j=1}^{N} \frac{W_{i,j}}{D_{i,i}} \|(\nabla f)([i,j])\|^{p-2}}$$

$$= \sum_{j=1}^{N} \frac{W_{i,j}}{D_{i,i}} \frac{1}{\|(\nabla f)([i,j])\|^{p-2}} \cdot \sum_{j=1}^{N} \frac{W_{i,j}}{D_{i,i}} \|(\nabla f)([i,j])\|^{p-2}$$

$$\geq \sum_{j=1}^{N} \frac{W_{i,j}}{D_{i,i}} \left(\frac{1}{\|(\nabla f)([i,j])\|^{p-2}} \cdot \|(\nabla f)([i,j])\|^{p-2}\right)$$

$$= 1,$$

we have

$$\tilde{\alpha}_i = \frac{1}{\sum_{i=1}^{N} \frac{M_{i,j}}{D_{i,i}}} = \frac{1}{\sum_{j=1}^{N} \frac{W_{i,j}}{D_{i,i}} \|(\nabla f)([i,j])\|^{p-2}}$$

$$\leq \sum_{j=1}^{N} \frac{W_{i,j}}{D_{i,i}} \frac{1}{\|(\nabla f)([i,j])\|^{p-2}}$$

$$= \sum_{j=1}^{N} \frac{W_{i,j}}{D_{i,i}} \|(\nabla f)([i,j])\|^{2-p}$$

$$\leq \|\nabla f(i)\|^{2-p}.$$

$\tilde{\alpha}_i \geq 2^{1-p}/\sqrt{N_k}$ directly implies that $\|\nabla f(i)\|^{2-p} \geq 2^{1-p}/\sqrt{N_k}$, i.e. $\|\nabla f(i)\| \geq 2(2\sqrt{N_k})^{1/(p-2)}$ and when $0 \leq \|\nabla f(i)\|^{2-p} \leq 2^{1-p}/\sqrt{N_k}$, i.e. $0 \leq \|\nabla f(i)\| \leq 2(2\sqrt{N_k})^{1/(p-2)}$, $0 \leq \tilde{\alpha}_i \leq 2^{1-p}/\sqrt{N_k}$ always holds. Therefore, if $0 \leq \|\nabla f(i)\| \leq 2(2\sqrt{N_k})^{1/(p-2)}$, $g_p(\lambda_{i-1})$ work as low-pass filters; If $g_p(\lambda_{i-1})$ work as both low-pass and high-pass filters, $\|\nabla f(i)\| \geq 2\left(2\sqrt{N_k}\right)^{1/(p-2)}$.

Specifically, when $p = 1$, by Thm. 7 we have for all $i = 1, \ldots, N$, $0 \leq \lambda_{i-1} \leq 2^{p-1}\sqrt{N_{\min}}$. Following the same derivation above we attain if $0 \leq \|\nabla f(i)\| \leq 2(2\sqrt{N_{\min}})^{1/(p-2)}$, $g_p(\lambda_{i-1})$ work as low-pass filters; If $g_p(\lambda_{i-1})$ work as both low-pass and high-pass filters, $\|\nabla f(i)\| \geq 2\left(2\sqrt{N_{\min}}\right)^{1/(p-2)}$.

$\square$

# E DATASET STATISTICS AND HYPERPARAMETERS

## E.1 ILLUSTRATION OF GRAPH GRADIENT AND GRAPH DIVERGENCE

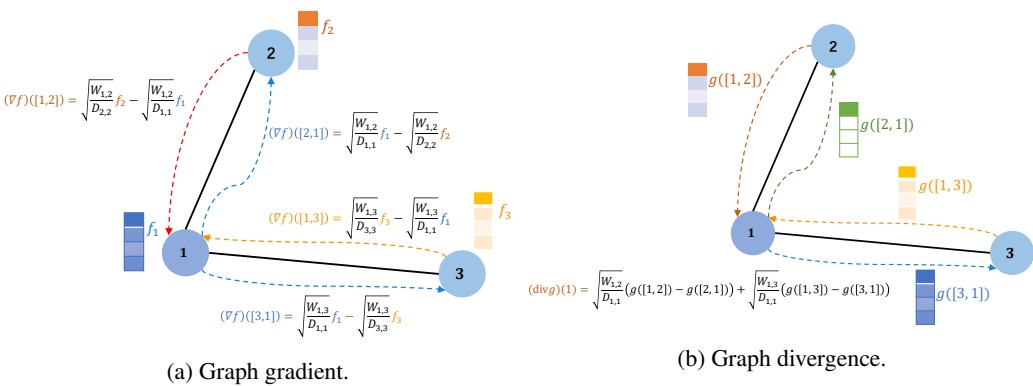

(a) Graph gradient.

(b) Graph divergence.

Figure 4: A tiny example of illustration of graph gradient and graph divergence. Best view in colors.

## E.2 DATASET STATISTICS

Table 2 summarizes the dataset statistics and the levels of homophily $\mathcal{H}(\mathcal{G})$ of all benchmark datasets. Note that the homophily scores here is different with the scores reported by Chien et al. (2021). There is a bug in their code when computing the homophily scores (doing division with torch integers) which caused their homophily scores to be smaller.

Table 2: Statistics of datasets.

| Dataset | #Class | #Feature | #Node | #Edge | Training | Validation | Testing | $\mathcal{H}(\mathcal{G})$ |
|---|---|---|---|---|---|---|---|---|
| Cora | 7 | 1433 | 2708 | 5278 | 2.5% | 2.5% | 95% | 0.825 |
| CiteSeer | 6 | 3703 | 3327 | 4552 | 2.5% | 2.5% | 95% | 0.717 |
| PubMed | 3 | 500 | 19717 | 44324 | 2.5% | 2.5% | 95% | 0.792 |
| Computers | 10 | 767 | 13381 | 245778 | 2.5% | 2.5% | 95% | 0.802 |
| Photo | 8 | 745 | 7487 | 119043 | 2.5% | 2.5% | 95% | 0.849 |
| CS | 15 | 6805 | 18333 | 81894 | 2.5% | 2.5% | 95% | 0.832 |
| Physics | 5 | 8415 | 34493 | 247962 | 2.5% | 2.5% | 95% | 0.915 |
| Chameleon | 5 | 2325 | 2277 | 31371 | 60% | 20% | 20% | 0.247 |
| Squirrel | 5 | 2089 | 5201 | 198353 | 60% | 20% | 20% | 0.216 |
| Actor | 5 | 932 | 7600 | 26659 | 60% | 20% | 20% | 0.221 |
| Wisconsin | 5 | 251 | 499 | 1703 | 60% | 20% | 20% | 0.150 |
| Texas | 5 | 1703 | 183 | 279 | 60% | 20% | 20% | 0.097 |
| Cornell | 5 | 1703 | 183 | 277 | 60% | 20% | 20% | 0.386 |

## E.3 HYPERPARAMETER SETTINGS

We set the number of layers as 2, the maximum number of epochs as 1000, the number for early stopping as 200, the weight decay as 0 or 0.0005 for all models. The other hyperparameters for each model are listed as below:

- $^{1.0}$GNN, $^{1.5}$GNN, $^{2.0}$GNN, $^{2.5}$GNN:
    - Number of hidden units: 16
    - Learning rate: $\{0.001, 0.01, 0.05\}$
    - Dropout rate: $\{0, 0.5\}$
    - $\mu$: $\{0.01, 0.1, 0.2, 1, 10\}$
    - $K$: $4, 6, 8$

- MLP:
  - Number of hidden units: 16
  - Learning rate: $\{0.001, 0.01\}$
  - Dropout rate: $\{0, 0.5\}$
- GCN:
  - Number of hidden units: 16
  - Learning rate: $\{0.001, 0.01\}$
  - Dropout rate: $\{0, 0.5\}$
- SGC:
  - Number of hidden units: 16
  - Learning rate: $\{0.2, 0.01\}$
  - Dropout rate: $\{0, 0.5\}$
  - $K$: 2
- GAT:
  - Number of hidden units: 8
  - Number of attention heads: 8
  - Learning rate: $\{0.001, 0.005\}$
  - Dropout rate: $\{0, 0.6\}$
- JKNet:
  - Number of hidden units: 16
  - Learning rate: $\{0.001, 0.01\}$
  - Dropout rate: $\{0, 0.5\}$
  - $K$: 10
  - $\alpha$: $\{0.1, 0.5, 0.7, 1\}$
  - The number of GCN based layers: 2
  - The layer aggregation: LSTM with 16 channels and 4 layers
- APPNP:
  - Number of hidden units: 16
  - Learning rate: $\{0.001, 0.01\}$
  - Dropout rate: $\{0, 0.5\}$
  - $K$: 10
  - $\alpha$: $\{0.1, 0.5, 0.7, 1\}$
- GPRGNN:
  - Number of hidden units: 16
  - Learning rate: $\{0.001, 0.01, 0.05\}$
  - Dropout rate: $\{0, 0.5\}$
  - $K$: 10
  - $\alpha$: $\{0, 0.1, 0.2, 0.5, 0.7, 0.9, 1\}$
  - dprate: $\{0, 0.5, 0.7\}$

## F  ADDITIONAL EXPERIMENTS

### F.1  EXPERIMENTAL RESULTS ON HOMOPHILIC BENCHMARK DATASETS

**Competitive Performance on Real-World Homophilic Datasets.** Table 3 summarizes the averaged accuracy (the micro-F1 score) and standard deviation of semi-supervised node classification on homophilic benchmark datasets. Table 3 shows that the performance of $^p$GNN is very close to APPNP, JKNet, GCN on Cora, CiteSeer, PubMed datasets and slightly outperforms all baselines on Computers, Photo, CS, Physics datasets. Moreover, we observe that $^p$GNNs outperform GPRGNN on all homophilic datasets, which confirms that $^p$GNNs work better under weak supervised information (2.5% training rate) as discussed in Remark 3. We also see that all GNN models work significantly better than MLP on all homophilic datasets. It illustrates that the graph topological information is helpful for the label prediction tasks. Notably, $^{1.0}$GNN is slightly worse than the other $^p$GNNs with larger $p$, which suggests to use $p \approx 2$ for homophilic graphs. Overall, the results of Table 3 indicates that $^p$GNNs obtain competitive performance against all baselines on homophilic datasets.

Table 3: Results on homophilic benchmark datasets. Averaged accuracy (%) for 100 runs. Best results are outlined in bold and the results within $95\%$ confidence interval of the best results are outlined in underlined bold. OOM denotes out of memory.

| Method | Cora | CiteSeer | PubMed | Computers | Photo | CS | Physics |
|--------|------|----------|--------|-----------|-------|-----|---------|
| MLP | $43.47_{\pm 3.82}$ | $46.95_{\pm 2.15}$ | $78.95_{\pm 0.49}$ | $66.11_{\pm 2.70}$ | $76.44_{\pm 2.83}$ | $86.24_{\pm 1.43}$ | $92.58_{\pm 0.83}$ |
| GCN | $76.23_{\pm 0.79}$ | $62.43_{\pm 0.81}$ | $83.72_{\pm 0.27}$ | $84.17_{\pm 0.59}$ | $90.46_{\pm 0.48}$ | $90.33_{\pm 0.36}$ | $94.46_{\pm 0.08}$ |
| SGC | $77.19_{\pm 1.47}$ | $\mathbf{64.10}_{\pm 1.36}$ | $79.26_{\pm 0.69}$ | $84.32_{\pm 0.59}$ | $89.81_{\pm 0.57}$ | $91.06_{\pm 0.05}$ | OOM |
| GAT | $75.62_{\pm 1.01}$ | $61.28_{\pm 1.09}$ | $83.60_{\pm 0.22}$ | $82.72_{\pm 1.29}$ | $90.48_{\pm 0.57}$ | $89.96_{\pm 0.27}$ | $93.96_{\pm 0.21}$ |
| JKNet | $77.19_{\pm 0.98}$ | $63.32_{\pm 0.95}$ | $82.54_{\pm 0.43}$ | $79.94_{\pm 2.47}$ | $88.29_{\pm 1.64}$ | $89.69_{\pm 0.66}$ | $93.92_{\pm 0.32}$ |
| APPNP | $\mathbf{79.58}_{\pm 0.59}$ | $63.02_{\pm 1.10}$ | $\mathbf{84.80}_{\pm 0.22}$ | $83.32_{\pm 1.11}$ | $90.42_{\pm 0.53}$ | $91.54_{\pm 0.24}$ | $\mathbf{94.93}_{\pm 0.06}$ |
| GPRGNN | $76.10_{\pm 1.30}$ | $61.60_{\pm 1.69}$ | $83.16_{\pm 0.84}$ | $82.78_{\pm 1.87}$ | $89.81_{\pm 0.66}$ | $90.59_{\pm 0.38}$ | $\underline{\mathbf{94.72}}_{\pm 0.16}$ |
| $^{1.0}$GNN | $77.59_{\pm 0.69}$ | $63.19_{\pm 0.98}$ | $83.21_{\pm 0.30}$ | $84.46_{\pm 0.89}$ | $\underline{\mathbf{90.69}}_{\pm 0.66}$ | $91.46_{\pm 0.50}$ | $\underline{\mathbf{94.72}}_{\pm 0.37}$ |
| $^{1.5}$GNN | $78.86_{\pm 0.75}$ | $\underline{\mathbf{63.80}}_{\pm 0.79}$ | $83.65_{\pm 0.17}$ | $\mathbf{85.03}_{\pm 0.90}$ | $\mathbf{90.91}_{\pm 0.50}$ | $\mathbf{92.12}_{\pm 0.40}$ | $\mathbf{94.90}_{\pm 0.16}$ |
| $^{2.0}$GNN | $78.93_{\pm 0.60}$ | $\underline{\mathbf{63.65}}_{\pm 1.08}$ | $84.19_{\pm 0.22}$ | $84.39_{\pm 0.85}$ | $90.40_{\pm 0.63}$ | $\mathbf{92.28}_{\pm 0.47}$ | $\mathbf{94.93}_{\pm 0.14}$ |
| $^{2.5}$GNN | $78.87_{\pm 0.57}$ | $63.28_{\pm 0.97}$ | $\underline{\mathbf{84.45}}_{\pm 0.18}$ | $83.85_{\pm 0.87}$ | $89.82_{\pm 0.64}$ | $91.94_{\pm 0.40}$ | $\underline{\mathbf{94.87}}_{\pm 0.11}$ |

### F.2  EXPERIMENTAL RESULTS OF AGGREGATION WEIGHT ENTROPY DISTRIBUTION

Here we present the visualization results of the learned aggregation weight entropy distribution of $^p$GNNs and GAT on all benchmark datasets. Fig. 5 and Fig. 6 show the results obtained on homophilic and heterophilic benchmark datasets, respectively.

We observe from Fig. 5 that the aggregation weight entropy distributions learned by $^p$GNNs and GAT on homophilic benchmark datasets are similar to the uniform cases, which indicates that aggregating and transforming node features over the original graph topology is very helpful for label prediction. It explains why $^p$GNNs and GNN baselines obtained similar performance on homophilic benchmark datasets and all GNN models significantly outperform MLP.

Contradict to the results on homophilic graphs shown in Fig. 5, Fig. 6 shows that the aggregation weight entropy distributions of $^p$GNNs on heterophilic benchmark datasets are very different from that of GAT and the uniform cases. We observe from Fig. 6 that the entropy of most of the aggregation weights learned by $^p$GNNs are around zero, which means that most aggregation weights are on one source node. It indicates that the graph topological information in these heterophilic benchmark graphs is not helpful for label prediction. Therefore, propagating and transforming node features over the graph topology could lead to worse performance than MLPs, which validates the results in Table 3 that the performance of MLP is significantly better most GNN baselines on all heterophilic graphs and closed to $^p$GNNs.

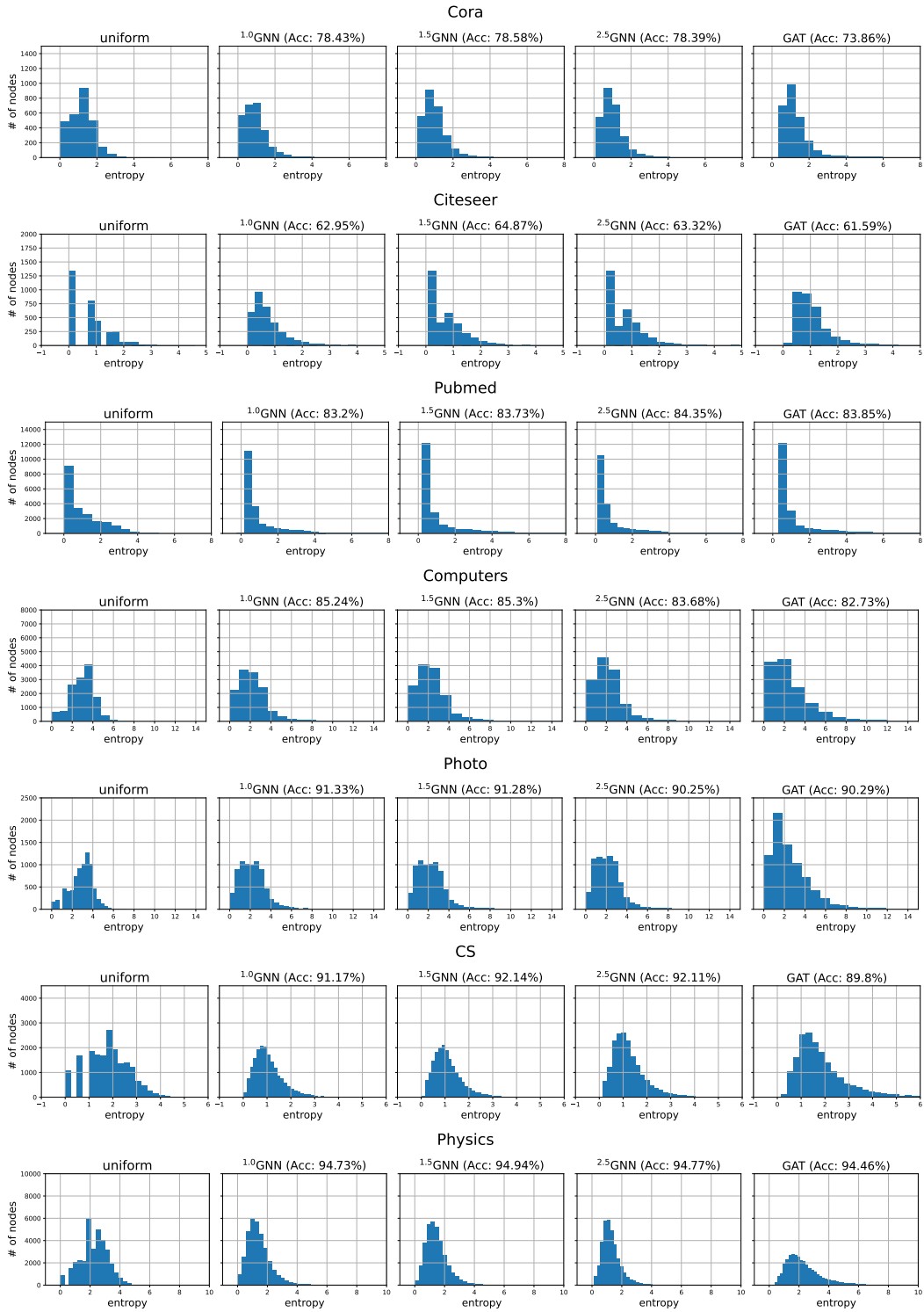

Figure 5: Aggregation weight entropy distribution of homophilic benchmark graphs. Low entropy means high degree of concentration, vice versa. An entropy of zero means all aggregation weights are on one source node.

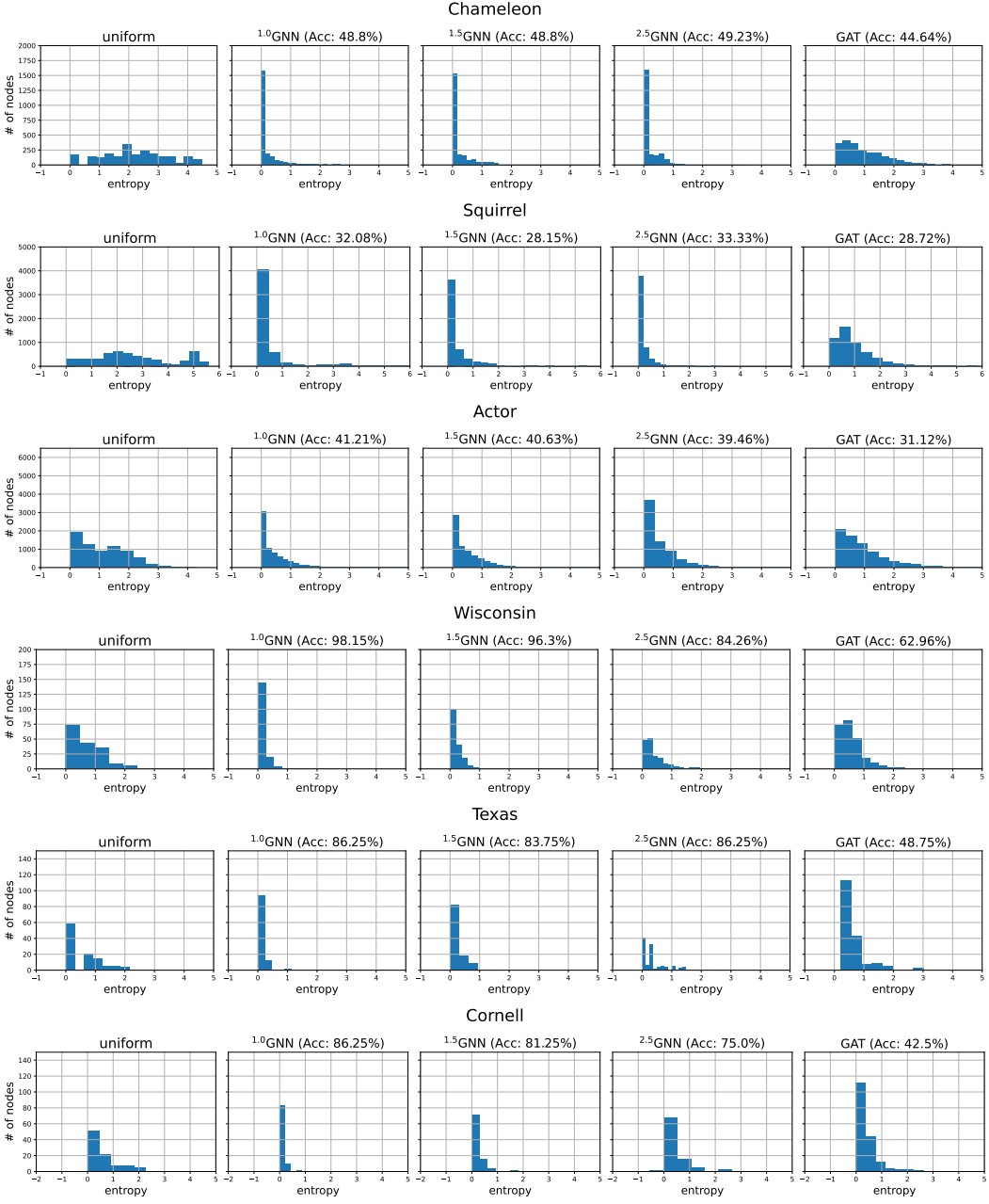

Figure 6: Aggregation weight entropy distribution of heterophilic benchmark graphs. Low entropy means high degree of concentration and vice versa. An entropy of zero means all aggregation weights are on one source node.

### F.3 EXPERIMENTAL RESULTS ON CSBM

In this section we present the experimental results on cSBM using sparse splitting and dense splitting, respectively. We used the same settings in Chien et al. (2021) in which the number of nodes $n = 5000$, the number of features $f = 2000$, $\epsilon = 3.25$ for all experiments. Table 4 reports the results on cSBM with sparse splitting setting, which also are presented in Fig. 2 and discussed in Sec. 5. Table 5 reports the results on cSBM with dense splitting settings.

Table 4: Results on cSBM with sparse splitting setting. Average accuracy (%) for 20 runs. Best results are outlined in bold and the results within 95% confidence interval of the best results are outlined in underlined bold.

| Method | $\phi = -1$ | $\phi = -0.75$ | $\phi = -0.5$ | $\phi = -0.25$ | $\phi = 0$ | $\phi = 0.25$ | $\phi = 0.5$ | $\phi = 0.75$ | $\phi = 1$ |
|---|---|---|---|---|---|---|---|---|---|
| MLP | $49.72_{\pm0.36}$ | $51.42_{\pm1.83}$ | $59.21_{\pm1.01}$ | $\mathbf{61.57}_{\pm0.38}$ | $\underline{\mathbf{61.70}}_{\pm0.30}$ | $59.92_{\pm1.88}$ | $57.20_{\pm0.62}$ | $54.48_{\pm0.48}$ | $50.09_{\pm0.51}$ |
| GCN | $57.24_{\pm1.15}$ | $58.19_{\pm1.46}$ | $57.30_{\pm1.30}$ | $51.97_{\pm0.44}$ | $54.45_{\pm1.38}$ | $64.70_{\pm2.38}$ | $82.45_{\pm1.35}$ | $91.31_{\pm0.54}$ | $76.07_{\pm3.30}$ |
| SGC | $55.98_{\pm1.48}$ | $58.56_{\pm1.40}$ | $56.97_{\pm0.54}$ | $51.54_{\pm0.22}$ | $52.69_{\pm2.36}$ | $64.14_{\pm1.05}$ | $79.88_{\pm1.57}$ | $90.37_{\pm0.09}$ | $75.94_{\pm0.92}$ |
| GAT | $59.72_{\pm2.23}$ | $60.20_{\pm2.14}$ | $55.38_{\pm1.96}$ | $50.15_{\pm0.55}$ | $53.05_{\pm1.40}$ | $64.00_{\pm2.03}$ | $81.04_{\pm1.71}$ | $90.37_{\pm1.33}$ | $78.24_{\pm1.95}$ |
| JKNet | $49.70_{\pm0.39}$ | $49.75_{\pm0.79}$ | $49.65_{\pm0.52}$ | $49.65_{\pm0.58}$ | $52.36_{\pm2.09}$ | $62.76_{\pm2.54}$ | $87.10_{\pm1.52}$ | $\underline{\mathbf{97.43}}_{\pm0.36}$ | $\mathbf{97.69}_{\pm0.52}$ |
| APPNP | $48.45_{\pm0.98}$ | $49.65_{\pm0.46}$ | $53.31_{\pm0.89}$ | $56.58_{\pm0.58}$ | $60.10_{\pm0.65}$ | $65.02_{\pm2.23}$ | $82.95_{\pm1.38}$ | $95.49_{\pm0.43}$ | $89.85_{\pm0.60}$ |
| GPRGNN | $97.26_{\pm0.66}$ | $94.81_{\pm0.91}$ | $82.14_{\pm0.47}$ | $61.15_{\pm2.55}$ | $60.20_{\pm0.76}$ | $62.90_{\pm2.22}$ | $83.61_{\pm1.28}$ | $96.96_{\pm0.41}$ | $\mathbf{98.01}_{\pm0.71}$ |
| $^{1.0}$GNN | $95.75_{\pm1.21}$ | $93.06_{\pm1.13}$ | $77.39_{\pm4.21}$ | $\underline{\mathbf{61.38}}_{\pm0.39}$ | $61.80_{\pm0.29}$ | $\underline{\mathbf{65.73}}_{\pm2.11}$ | $85.85_{\pm3.24}$ | $96.80_{\pm0.87}$ | $97.40_{\pm1.10}$ |
| $^{1.5}$GNN | $95.90_{\pm3.01}$ | $94.10_{\pm4.57}$ | $73.08_{\pm2.59}$ | $\underline{\mathbf{61.44}}_{\pm0.30}$ | $\underline{\mathbf{61.77}}_{\pm0.35}$ | $66.01_{\pm1.88}$ | $90.57_{\pm0.71}$ | $\underline{\mathbf{97.38}}_{\pm0.43}$ | $\mathbf{97.76}_{\pm0.86}$ |
| $^{2.0}$GNN | $\mathbf{98.37}_{\pm0.78}$ | $\mathbf{96.32}_{\pm1.50}$ | $84.93_{\pm0.39}$ | $61.13_{\pm0.51}$ | $\underline{\mathbf{61.79}}_{\pm0.34}$ | $63.55_{\pm1.73}$ | $88.55_{\pm1.05}$ | $\mathbf{97.56}_{\pm0.16}$ | $\mathbf{97.94}_{\pm0.39}$ |
| $^{2.5}$GNN | $97.74_{\pm0.99}$ | $\mathbf{96.78}_{\pm0.44}$ | $83.21_{\pm2.12}$ | $\underline{\mathbf{61.30}}_{\pm0.41}$ | $\underline{\mathbf{61.74}}_{\pm0.34}$ | $62.88_{\pm2.31}$ | $79.64_{\pm2.15}$ | $95.71_{\pm0.34}$ | $97.25_{\pm0.58}$ |

Table 5: Results on cSBM with dense splitting setting. Average accuracy (%) for 20 runs. Best results are outlined in bold and the results within 95% confidence interval of the best results are outlined in underlined bold.

| Method | $\phi = -1$ | $\phi = -0.75$ | $\phi = -0.5$ | $\phi = -0.25$ | $\phi = 0$ | $\phi = 0.25$ | $\phi = 0.5$ | $\phi = 0.75$ | $\phi = 1$ |
|---|---|---|---|---|---|---|---|---|---|
| MLP | $50.37_{\pm0.60}$ | $65.22_{\pm0.92}$ | $75.82_{\pm0.65}$ | $81.18_{\pm0.55}$ | $79.86_{\pm0.69}$ | $79.97_{\pm0.57}$ | $75.03_{\pm0.89}$ | $67.53_{\pm0.68}$ | $51.96_{\pm0.69}$ |
| GCN | $83.14_{\pm0.49}$ | $82.59_{\pm0.48}$ | $77.17_{\pm0.59}$ | $58.58_{\pm0.41}$ | $61.18_{\pm1.06}$ | $82.59_{\pm0.50}$ | $92.20_{\pm0.27}$ | $97.21_{\pm0.27}$ | $97.10_{\pm0.12}$ |
| SGC | $78.35_{\pm0.36}$ | $82.13_{\pm0.09}$ | $77.76_{\pm0.12}$ | $59.14_{\pm0.57}$ | $60.31_{\pm0.63}$ | $81.96_{\pm0.34}$ | $91.68_{\pm0.13}$ | $96.56_{\pm0.09}$ | $96.87_{\pm0.05}$ |
| GAT | $92.99_{\pm0.86}$ | $90.89_{\pm0.60}$ | $87.02_{\pm0.80}$ | $68.40_{\pm1.60}$ | $61.98_{\pm1.16}$ | $82.92_{\pm0.51}$ | $92.05_{\pm0.73}$ | $97.28_{\pm0.25}$ | $98.04_{\pm0.46}$ |
| JKNet | $68.95_{\pm9.05}$ | $79.21_{\pm7.67}$ | $67.97_{\pm5.22}$ | $56.12_{\pm4.10}$ | $58.33_{\pm1.70}$ | $80.15_{\pm0.80}$ | $91.21_{\pm0.50}$ | $\underline{\mathbf{97.62}}_{\pm0.25}$ | $\mathbf{98.32}_{\pm0.21}$ |
| APPNP | $49.86_{\pm0.39}$ | $50.47_{\pm0.89}$ | $65.28_{\pm0.68}$ | $73.98_{\pm0.64}$ | $79.37_{\pm0.66}$ | $86.60_{\pm0.73}$ | $\mathbf{92.45}_{\pm0.39}$ | $97.67_{\pm0.14}$ | $97.65_{\pm0.49}$ |
| GPRGNN | $\mathbf{99.06}_{\pm0.25}$ | $\underline{\mathbf{97.14}}_{\pm0.31}$ | $\mathbf{94.59}_{\pm0.32}$ | $\mathbf{83.84}_{\pm0.69}$ | $78.81_{\pm1.30}$ | $85.85_{\pm1.01}$ | $92.08_{\pm0.81}$ | $\underline{\mathbf{97.49}}_{\pm0.22}$ | $98.46_{\pm0.15}$ |
| $^{1.0}$GNN | $98.19_{\pm0.28}$ | $94.38_{\pm0.44}$ | $86.40_{\pm1.00}$ | $80.57_{\pm0.43}$ | $\underline{\mathbf{80.21}}_{\pm0.42}$ | $\mathbf{87.32}_{\pm0.47}$ | $\mathbf{92.42}_{\pm0.62}$ | $\underline{\mathbf{97.52}}_{\pm0.33}$ | $\mathbf{98.37}_{\pm0.26}$ |
| $^{1.5}$GNN | $\underline{\mathbf{98.88}}_{\pm0.16}$ | $95.62_{\pm0.21}$ | $86.87_{\pm1.22}$ | $80.70_{\pm0.71}$ | $\underline{\mathbf{80.28}}_{\pm0.31}$ | $86.29_{\pm0.43}$ | $\underline{\mathbf{92.40}}_{\pm0.24}$ | $\underline{\mathbf{97.56}}_{\pm0.25}$ | $\mathbf{98.24}_{\pm0.32}$ |
| $^{2.0}$GNN | $99.21_{\pm0.09}$ | $\underline{\mathbf{96.91}}_{\pm0.16}$ | $92.96_{\pm0.31}$ | $80.83_{\pm0.61}$ | $\underline{\mathbf{80.04}}_{\pm0.49}$ | $84.96_{\pm0.60}$ | $91.18_{\pm0.27}$ | $\underline{\mathbf{97.41}}_{\pm0.14}$ | $\mathbf{98.45}_{\pm0.14}$ |
| $^{2.5}$GNN | $99.21_{\pm0.14}$ | $\underline{\mathbf{96.94}}_{\pm0.16}$ | $93.28_{\pm0.37}$ | $80.93_{\pm0.44}$ | $\underline{\mathbf{80.28}}_{\pm0.38}$ | $83.83_{\pm0.70}$ | $86.10_{\pm0.39}$ | $96.28_{\pm0.43}$ | $97.76_{\pm0.18}$ |

Table 5 shows that $^p$GNNs obtain the best performance on weak homophilic graphs ($\phi = 0, 0.25$) while competitive performance against GPRGNN on strong heterophilic graphs ($\phi = -0.75, -1$) and competitive performance with state-of-the-art GNNs on strong homophilic graphs ($\phi = 0.75, 1$). We also observe that GPRGNN is slightly better than $^p$GNNs on weak heterophilic graphs ($\phi = -0.25, -0.5$), which suggests that GPRGNN could work very well using strong supervised information (60% training rate and 20% validation rate). However, as shown in Table 4, $^p$GNNs work better than GPRGNN under weak supervised information (2.5% training rate and 2.5%) on all heterophilic graphs. The result is reasonable, as discussed in Remark 3 in Sec. 3.2, GPRGNN can adaptively learn the generalized PageRank (GPR) weights and it works similarly to $^{2.0}$GNN on both homophilic and heterophilic graphs. However, it needs more supervised information in order to learn optimal GPR weights. On the contrary, $^p$GNNs need less supervised information to obtain similar results because $\Theta^{(2)}$ acts like a hyperplane for classification. Therefore, $^p$GNNs can work better under weak supervised information.

### F.4 EXPERIMENTAL RESULTS ON GRAPHS WITH NOISY EDGES

Here we present more experimental results on graph with noisy edges. Table 6 reports the results on homophilic graphs (Computers, Photo, CS, Physics) and Table 7 reports the results on heterophilic graphs (Wisconsin Texas). We observe from Tables 6 and 7 that $^p$GNNs dominate all baselines. Moreover, $^p$GNNs even slightly better than MLP when the graph topologies are completely random, i.e. the noisy edge rate $r = 1$. We also observe that the performance of GCN, SGC, JKNet on homophilic graphs dramatically degrades as the noisy edge rate $r$ increases while they do not change

a lot for the cases on heterophilic graphs. It is reasonable since the original graph topological information is very helpful for label prediction on these homophilic graphs. Adding noisy edges and remove the same number of original edges could significantly degrade the performance of ordinary GNNs. On the other hand, since we find that the original graph topological information in Wisconsin and Texas is not helpful for label prediction. Therefore, adding noisy edges and removing original edges on these heterophilic graphs would not affect too much their performance.

Table 6: Results on homophilic graphs with random edges. Average accuracy (%) for 20 runs. Best results are outlined in bold and the results within 95% confidence interval of the best results are outlined in underlined bold. OOM denotes out of memory.

| Method | Computers | | | Photo | | |
|---|---|---|---|---|---|---|
| | $r = 0.25$ | $r = 0.5$ | $r = 1$ | $r = 0.25$ | $r = 0.5$ | $r = 1$ |
| MLP | $66.11_{\pm2.70}$ | $66.11_{\pm2.70}$ | $66.11_{\pm2.70}$ | $76.44_{\pm2.83}$ | $76.44_{\pm2.83}$ | $76.44_{\pm2.83}$ |
| GCN | $74.70_{\pm1.72}$ | $62.16_{\pm2.76}$ | $8.95_{\pm6.90}$ | $81.43_{\pm0.76}$ | $75.52_{\pm3.59}$ | $12.78_{\pm5.20}$ |
| SGC | $75.15_{\pm1.08}$ | $66.96_{\pm1.05}$ | $15.79_{\pm7.47}$ | $82.22_{\pm0.36}$ | $77.80_{\pm0.49}$ | $13.57_{\pm3.63}$ |
| GAT | $76.44_{\pm1.81}$ | $68.34_{\pm2.61}$ | $11.58_{\pm7.70}$ | $82.70_{\pm1.31}$ | $77.20_{\pm2.10}$ | $13.74_{\pm5.14}$ |
| JKNet | $56.74_{\pm6.48}$ | $46.11_{\pm8.43}$ | $12.50_{\pm6.56}$ | $73.46_{\pm6.74}$ | $64.18_{\pm4.06}$ | $15.66_{\pm6.10}$ |
| APPNP | $78.23_{\pm1.84}$ | $74.57_{\pm2.25}$ | $66.67_{\pm2.68}$ | $87.63_{\pm1.05}$ | $86.22_{\pm1.73}$ | $75.55_{\pm1.72}$ |
| GPRGNN | $77.30_{\pm2.24}$ | $77.11_{\pm1.80}$ | $66.85_{\pm1.65}$ | $85.95_{\pm1.05}$ | $85.64_{\pm1.22}$ | $\mathbf{77.46_{\pm1.44}}$ |
| $^{1.0}$GNN | $75.14_{\pm14.95}$ | $63.26_{\pm20.67}$ | $41.60_{\pm16.17}$ | $\underline{\mathbf{87.97}}_{\pm0.70}$ | $84.47_{\pm3.05}$ | $41.17_{\pm18.15}$ |
| $^{1.5}$GNN | $\mathbf{81.79_{\pm1.33}}$ | $\mathbf{78.12_{\pm2.08}}$ | $66.04_{\pm2.73}$ | $\mathbf{88.09_{\pm1.18}}$ | $86.20_{\pm1.61}$ | $68.78_{\pm8.97}$ |
| $^{2.0}$GNN | $80.34_{\pm1.07}$ | $76.90_{\pm1.93}$ | $\mathbf{67.17_{\pm1.63}}$ | $87.65_{\pm0.94}$ | $\mathbf{87.06_{\pm1.50}}$ | $\underline{\mathbf{77.07}}_{\pm1.83}$ |
| $^{2.5}$GNN | $79.14_{\pm1.51}$ | $75.49_{\pm1.25}$ | $64.95_{\pm2.27}$ | $87.38_{\pm0.85}$ | $86.11_{\pm1.10}$ | $76.65_{\pm1.46}$ |

| Method | CS | | | Physics | | |
|---|---|---|---|---|---|---|
| | $r = 0.25$ | $r = 0.5$ | $r = 1$ | $r = 0.25$ | $r = 0.5$ | $r = 1$ |
| MLP | $86.24_{\pm1.43}$ | $86.24_{\pm1.43}$ | $\underline{\mathbf{86.24}}_{\pm1.43}$ | $92.58_{\pm0.83}$ | $92.58_{\pm0.83}$ | $92.58_{\pm0.83}$ |
| GCN | $81.05_{\pm0.59}$ | $68.37_{\pm0.85}$ | $7.72_{\pm2.39}$ | $89.02_{\pm0.16}$ | $80.45_{\pm0.34}$ | $19.78_{\pm3.94}$ |
| SGC | $83.41_{\pm0.01}$ | $71.98_{\pm0.12}$ | $8.00_{\pm1.43}$ | OOM | OOM | OOM |
| GAT | $80.11_{\pm0.67}$ | $68.66_{\pm1.42}$ | $8.49_{\pm2.39}$ | $88.72_{\pm0.61}$ | $82.05_{\pm1.86}$ | $22.39_{\pm5.04}$ |
| JKNet | $81.35_{\pm0.74}$ | $71.30_{\pm2.14}$ | $11.43_{\pm1.18}$ | $87.98_{\pm0.97}$ | $81.90_{\pm2.27}$ | $26.38_{\pm5.80}$ |
| APPNP | $88.63_{\pm0.68}$ | $87.56_{\pm0.51}$ | $76.90_{\pm0.96}$ | $93.46_{\pm0.12}$ | $92.81_{\pm0.24}$ | $90.49_{\pm0.33}$ |
| GPRGNN | $85.77_{\pm0.81}$ | $83.89_{\pm1.54}$ | $72.79_{\pm2.24}$ | $92.18_{\pm0.29}$ | $90.96_{\pm0.48}$ | $91.77_{\pm0.41}$ |
| $^{1.0}$GNN | $90.27_{\pm0.86}$ | $89.56_{\pm0.81}$ | $\mathbf{86.60_{\pm1.22}}$ | $\mathbf{94.35_{\pm0.39}}$ | $\mathbf{94.23_{\pm0.27}}$ | $\mathbf{92.97_{\pm0.36}}$ |
| $^{1.5}$GNN | $\mathbf{91.27_{\pm0.40}}$ | $\mathbf{90.50_{\pm0.71}}$ | $84.40_{\pm1.84}$ | $\underline{\mathbf{94.34}}_{\pm0.21}$ | $93.77_{\pm0.29}$ | $92.51_{\pm0.35}$ |
| $^{2.0}$GNN | $90.97_{\pm0.49}$ | $89.98_{\pm0.50}$ | $80.84_{\pm1.48}$ | $\underline{\mathbf{94.14}}_{\pm0.18}$ | $93.30_{\pm0.31}$ | $91.72_{\pm0.44}$ |
| $^{2.5}$GNN | $89.90_{\pm0.45}$ | $89.00_{\pm0.59}$ | $76.82_{\pm2.11}$ | $93.61_{\pm0.30}$ | $92.77_{\pm0.26}$ | $91.16_{\pm0.47}$ |

Table 7: Results on heterophilic graphs with random edges. Average accuracy (%) for 20 runs. Best results are outlined in bold and the results within 95% confidence interval of the best results are outlined in underlined bold.

| Method | Wisconsin | | | Texas | | |
|---|---|---|---|---|---|---|
| | $r = 0.25$ | $r = 0.5$ | $r = 1$ | $r = 0.25$ | $r = 0.5$ | $r = 1$ |
| MLP | $93.56_{\pm3.14}$ | $93.56_{\pm3.14}$ | $93.56_{\pm3.14}$ | $79.50_{\pm10.62}$ | $79.50_{\pm10.62}$ | $79.50_{\pm10.62}$ |
| GCN | $62.31_{\pm8.12}$ | $59.44_{\pm5.76}$ | $64.21_{\pm4.49}$ | $41.56_{\pm8.89}$ | $44.69_{\pm23.05}$ | $40.31_{\pm18.26}$ |
| SGC | $64.68_{\pm7.34}$ | $62.36_{\pm2.64}$ | $51.81_{\pm2.63}$ | $42.50_{\pm5.49}$ | $40.94_{\pm18.34}$ | $23.81_{\pm14.54}$ |
| GAT | $65.37_{\pm9.04}$ | $60.05_{\pm9.12}$ | $60.05_{\pm7.46}$ | $39.50_{\pm9.29}$ | $34.88_{\pm21.59}$ | $29.38_{\pm11.53}$ |
| JKNet | $64.91_{\pm13.07}$ | $51.39_{\pm10.36}$ | $57.41_{\pm2.57}$ | $47.75_{\pm7.30}$ | $46.62_{\pm23.23}$ | $40.69_{\pm13.57}$ |
| APPNP | $70.19_{\pm9.04}$ | $60.32_{\pm4.70}$ | $72.64_{\pm4.73}$ | $66.69_{\pm13.46}$ | $63.25_{\pm9.87}$ | $69.81_{\pm7.76}$ |
| GPRGNN | $90.97_{\pm3.83}$ | $87.50_{\pm3.86}$ | $87.55_{\pm2.97}$ | $74.25_{\pm7.25}$ | $76.75_{\pm14.05}$ | $80.69_{\pm5.87}$ |
| $^{1.0}$GNN | $\mathbf{94.91_{\pm2.73}}$ | $\mathbf{95.97_{\pm2.00}}$ | $\mathbf{95.97_{\pm2.27}}$ | $81.50_{\pm9.24}$ | $\mathbf{82.12_{\pm11.09}}$ | $\mathbf{81.81_{\pm5.67}}$ |
| $^{1.5}$GNN | $\underline{\mathbf{94.58}}_{\pm1.25}$ | $95.19_{\pm2.18}$ | $94.95_{\pm2.79}$ | $82.50_{\pm6.39}$ | $78.12_{\pm5.30}$ | $78.50_{\pm7.98}$ |
| $^{2.0}$GNN | $90.46_{\pm2.79}$ | $90.97_{\pm4.22}$ | $91.44_{\pm2.27}$ | $\mathbf{86.06_{\pm5.17}}$ | $69.38_{\pm11.47}$ | $63.50_{\pm8.90}$ |
| $^{2.5}$GNN | $82.45_{\pm3.93}$ | $88.24_{\pm2.79}$ | $84.40_{\pm1.98}$ | $80.00_{\pm10.83}$ | $56.62_{\pm10.01}$ | $52.31_{\pm10.58}$ |

### F.5 Experimental Results of Intergrating $^p$GNNs with GCN and JKNet

Here we further conduct experiments to study whether $^p$GNNs can be intergrated into existing GNN architectures and improve their performance on heterophilic graphs. We use two popular GNN architectures: GCN (Kipf & Welling, 2017) and JKNet (Xu et al., 2018).

To incorporate $^p$GNNs with GCN, we use the $^p$GNN layers as the first layer of the combined models, termed as $^p$GNN + GCN, and GCN layer as the second layer. Specifically, we use the aggregation weights $\alpha \mathbf{D}^{-1/2}\mathbf{M}\mathbf{D}^{-1/2}$ learned by the $^p$GNN in the first layer as the input edge weights of GCN layer in the second layer. To combine $^p$GNN with JKNet, we use the $^p$GNN layer as the GNN layers in the JKNet framework, termed as $^p$GNN + JKNet. Tables 8 and 9 report the experimental results on homophilic and heterophilic benchmark datasets, respectively.

Table 8: The results of $^p$GNNs + GCN and $^p$GNNs + JKNet on homophilic benchmark dataset. Averaged accuracy (%) for 20 runs. Best results are outlined in bold and the results within 95% confidence interval of the best results are outlined in underlined bold.

| Method | Cora | CiteSeer | PubMed | Computers | Photo | CS | Physics |
|---|---|---|---|---|---|---|---|
| GCN | $\mathbf{76.23}_{\pm0.79}$ | $\mathbf{62.43}_{\pm0.81}$ | $\mathbf{83.72}_{\pm0.27}$ | $\mathbf{84.17}_{\pm0.59}$ | $\mathbf{90.46}_{\pm0.48}$ | $\mathbf{90.33}_{\pm0.36}$ | $94.46_{\pm0.08}$ |
| $^{1.0}$GNN + GCN | $72.37_{\pm1.35}$ | $60.56_{\pm1.59}$ | $82.14_{\pm0.31}$ | $83.75_{\pm1.05}$ | $\underline{\mathbf{90.24}}_{\pm1.12}$ | $89.60_{\pm0.46}$ | $\underline{\mathbf{94.59}}_{\pm0.33}$ |
| $^{1.5}$GNN + GCN | $72.72_{\pm1.39}$ | $60.23_{\pm1.80}$ | $82.21_{\pm0.22}$ | $\underline{\mathbf{83.89}}_{\pm0.74}$ | $90.00_{\pm0.68}$ | $89.48_{\pm0.45}$ | $94.70_{\pm0.18}$ |
| $^{2.0}$GNN + GCN | $72.39_{\pm1.55}$ | $60.19_{\pm1.60}$ | $82.24_{\pm0.23}$ | $\underline{\mathbf{83.92}}_{\pm1.09}$ | $\underline{\mathbf{90.17}}_{\pm0.83}$ | $89.60_{\pm0.71}$ | $94.51_{\pm0.39}$ |
| $^{2.5}$GNN + GCN | $72.85_{\pm1.19}$ | $59.68_{\pm1.85}$ | $82.23_{\pm0.34}$ | $83.69_{\pm0.92}$ | $90.02_{\pm1.09}$ | $89.53_{\pm0.68}$ | $\underline{\mathbf{94.58}}_{\pm0.31}$ |
| JKNet | $\mathbf{77.19}_{\pm0.98}$ | $\mathbf{63.32}_{\pm0.95}$ | $\underline{\mathbf{82.54}}_{\pm0.43}$ | $79.94_{\pm2.47}$ | $88.29_{\pm1.64}$ | $89.69_{\pm0.66}$ | $93.92_{\pm0.32}$ |
| $^{1.0}$GNN+JKNet | $75.67_{\pm1.54}$ | $60.38_{\pm1.65}$ | $81.68_{\pm0.44}$ | $\mathbf{83.19}_{\pm1.36}$ | $89.71_{\pm1.05}$ | $90.26_{\pm0.72}$ | $94.27_{\pm0.69}$ |
| $^{1.5}$GNN+JKNet | $76.40_{\pm1.59}$ | $60.67_{\pm1.93}$ | $\underline{\mathbf{82.42}}_{\pm0.35}$ | $82.78_{\pm2.09}$ | $\mathbf{90.25}_{\pm1.03}$ | $\mathbf{90.76}_{\pm0.75}$ | $\mathbf{94.82}_{\pm0.34}$ |
| $^{2.0}$GNN+JKNet | $76.75_{\pm1.26}$ | $61.05_{\pm1.48}$ | $\underline{\mathbf{82.50}}_{\pm0.53}$ | $82.36_{\pm2.39}$ | $89.31_{\pm1.39}$ | $90.33_{\pm0.63}$ | $\underline{\mathbf{94.70}}_{\pm0.33}$ |
| $^{2.5}$GNN+JKNet | $76.48_{\pm1.28}$ | $60.97_{\pm0.97}$ | $82.56_{\pm1.04}$ | $81.45_{\pm1.55}$ | $89.21_{\pm1.10}$ | $89.66_{\pm0.68}$ | $94.29_{\pm0.59}$ |

Table 9: The results of $^p$GNNs + GCN and $^p$GNNs + JKNet on heterophilic benchmark dataset. Averaged accuracy (%) for 20 runs. Best results are outlined in bold and the results within 95% confidence interval of the best results are outlined in underlined bold.

| Method | Chameleon | Squirrel | Actor | Wisconsin | Texas | Cornell |
|---|---|---|---|---|---|---|
| GCN | $34.54_{\pm2.78}$ | $25.28_{\pm1.55}$ | $31.28_{\pm2.04}$ | $61.93_{\pm3.00}$ | $56.54_{\pm17.02}$ | $51.36_{\pm4.59}$ |
| $^{1.0}$GNN + GCN | $48.52_{\pm1.89}$ | $\underline{\mathbf{34.78}}_{\pm1.11}$ | $32.37_{\pm3.12}$ | $\mathbf{68.52}_{\pm3.75}$ | $\underline{\mathbf{67.94}}_{\pm12.60}$ | $\underline{\mathbf{67.81}}_{\pm7.61}$ |
| $^{1.5}$GNN + GCN | $48.85_{\pm2.13}$ | $34.61_{\pm1.11}$ | $32.37_{\pm2.48}$ | $66.25_{\pm3.95}$ | $65.62_{\pm11.99}$ | $64.88_{\pm9.19}$ |
| $^{2.0}$GNN + GCN | $48.71_{\pm2.24}$ | $\mathbf{35.06}_{\pm1.18}$ | $\mathbf{32.72}_{\pm2.02}$ | $66.34_{\pm4.51}$ | $65.94_{\pm7.63}$ | $\mathbf{68.62}_{\pm6.55}$ |
| $^{2.5}$GNN + GCN | $\mathbf{49.53}_{\pm2.19}$ | $34.40_{\pm1.60}$ | $32.40_{\pm3.23}$ | $67.18_{\pm3.50}$ | $\mathbf{68.31}_{\pm9.18}$ | $66.06_{\pm9.56}$ |
| JKNet | $33.28_{\pm3.59}$ | $25.82_{\pm1.58}$ | $29.77_{\pm2.61}$ | $61.08_{\pm3.71}$ | $59.65_{\pm12.62}$ | $55.34_{\pm4.43}$ |
| $^{1.0}$GNN + JKNet | $\underline{\mathbf{49.00}}_{\pm2.09}$ | $35.56_{\pm1.34}$ | $\mathbf{40.74}_{\pm0.98}$ | $\mathbf{95.23}_{\pm2.43}$ | $80.25_{\pm6.87}$ | $\mathbf{78.38}_{\pm8.14}$ |
| $^{1.5}$GNN + JKNet | $48.77_{\pm2.22}$ | $\mathbf{35.98}_{\pm0.93}$ | $40.22_{\pm1.27}$ | $94.86_{\pm2.00}$ | $80.38_{\pm9.79}$ | $72.25_{\pm9.83}$ |
| $^{2.0}$GNN + JKNet | $48.88_{\pm1.63}$ | $35.77_{\pm1.73}$ | $40.16_{\pm1.31}$ | $88.84_{\pm2.78}$ | $\mathbf{86.12}_{\pm5.59}$ | $74.75_{\pm7.81}$ |
| $^{2.5}$GNN + JKNet | $\mathbf{49.04}_{\pm1.95}$ | $35.78_{\pm1.87}$ | $40.00_{\pm1.12}$ | $85.42_{\pm3.86}$ | $79.06_{\pm7.60}$ | $76.81_{\pm7.66}$ |

We observe from Table 8 that intergrating $^p$GNNs with GCN and JKNet does not improve their performance on homophilic graphs. The performance of GCN slightly degrade after incorporating $^p$GNNs. The performance of JKNet also slightly degrade on Cora, CiteSeer, and PubMed but is improved on Computers, Photo, CS, Physics. It is reasonable since GCN and JKNet can predict well on these homophilic benchmark datasets based on their original graph topology.

However, for heterophilic benchmark datasets, Table 9 shows that there are significant improvements over GCN, and JKNet after intergrating with $^p$GNNs. Moreover, $^p$GNNs + JKNet obtain advanced performance on all heterophilic benchmark datasets and even better than $^p$GNNs on Squirrel. The results of Table 9 demonstrate that intergrating $^p$GNNs with GCN and JKNet can sigificantly improve their performance on heterophilic graphs.

### F.6 Experimental Results of $^p$GNNs on PPI Dataset for Inductive Learning

Additionally, we conduct comparison experiments of $^p$GNNs against GAT on PPI dataset (Zitnik & Leskovec, 2017) using the inductive learning settings as in Velickovic et al. (2018) (20 graphs for training, 2 graphs for validation, 2 graphs for testing). We use three layers of GAT architecture with 256 hidden units, use 1 attention head for GAT (1 head) and 4 attention heads for GAT (4 heads). We use three $^p$GNN layers and a MLP layer as the first layer for $^p$GNNs, set $\mu = 0.01$, $K = 1$, and use 256 hidden units for $^p$GNN-256 and 512 hidden units for $^p$GNN-512. The experimental results are reported in Table 10.

Table 10: Results on PPI datasets. Averaged micro-F1 scores for 10 runs. Best results are outlined in bold.

| Method | PPI |
|---|---|
| GAT (1 head) | $0.917 \pm 0.041$ |
| GAT (4 heads) | $0.972 \pm 0.002$ |
| $^{1.0}$GNN-256 | $0.961 \pm 0.003$ |
| $^{1.5}$GNN-256 | $0.967 \pm 0.008$ |
| $^{2.0}$GNN-256 | $0.968 \pm 0.006$ |
| $^{2.5}$GNN-256 | $0.973 \pm 0.002$ |
| $^{1.0}$GNN-512 | $0.978 \pm 0.005$ |
| $^{1.5}$GNN-512 | $0.977 \pm 0.008$ |
| $^{2.0}$GNN-512 | $\mathbf{0.981} \pm 0.006$ |
| $^{2.5}$GNN-512 | $0.978 \pm 0.005$ |

From Appendix F.6 we observe that the results of $^{2.5}$GNN on PPI slightly better than GAT with 4 attention heads and other $^p$GNNs are very close to it. Moreover, all results of $^p$GNNs significantly outperform GAT with one attention head. The results of $^p$GNNs on PPI is impressive. $^p$GNNs have much less parameters than GAT with 4 attention heads while obtain very completitive performance on PPI. When we use more hidden units, 512 hidden units, $^p$GNNs-512 significantly outperform GAT, while $^p$GNNs-512 still have less parameters. It illustrates the superior potential of applying $^p$GNNs to inducting learning on graphs.

### F.7 Experimental Results of $^p$GNNs on OGBN arXiv Dataset

Table 11: Results on OGBN arXiv dataset. Average accuracy (%) for 10 runs. Best results are outlined in bold.

| Method | OGBN arXiv |
|---|---|
| MLP | $55.50 \pm 0.23$ |
| GCN | $71.74 \pm 0.29$ |
| JKNet (GCN-based) | $72.19 \pm 0.21$ |
| DeepGCN | $71.92 \pm 0.16$ |
| GCN + residual (6 layers) | $72.86 \pm 0.16$ |
| GCN + residual (8 layers) + C&S | $72.97 \pm 0.22$ |
| GCN + residual (8 layers) + C&S v2 | $73.13 \pm 0.17$ |
| $^1$GNN | $72.40 \pm 0.19$ |
| $^2$GNN | $72.45 \pm 0.20$ |
| $^3$GNN | $72.58 \pm 0.23$ |
| $^1$GNN + residual (6 layers) + C&S | $72.96 \pm 0.22$ |
| $^2$GNN + residual (6 layers) + C&S | $73.13 \pm 0.20$ |
| $^3$GNN + residual (6 layers) + C&S | $\mathbf{73.23} \pm 0.16$ |

Here we present the experimental of $^p$GNNs on OGBN arXiv dataset (Hu et al., 2020). We use the official data split setting of OGBN arXiv. We use three layers $^p$GNN architecture and 256 hidden units with $\mu = 0.5$, $K = 2$. We also combine $^p$GNNs with correct and smooth model (C&S) (Huang et al., 2021) and introduce residual units. The results of MLP, GCN, JKNet, DeepGCN (Li et al.,

2019), GCN with residual units, C&S model are extracted from the leaderboard for OGBN arXiv. dataset[5]. Table 11 summaries the results of $^p$GNNs against the baselines.

We observe from Table 11 that $^p$GNNs outperform MLP, GCN, JKNet, and DeepGCN. The performance of $^p$GNNs can be further improved by combining it with C&S model and residual units and $^3$GNN + residual (6 layers) + C&S obtains the best performance against the baselines.

## F.8 Running Time of $^p$GNNs

Tables 12 and 13 report the averaged running time of $^p$GNNs and baselines on homophilic and heterophilic benchmark datasets, respectively.

Table 12: Efficiency on homophilic benchmark datasests. Averaged running time per epoch (ms) / averaged total running time (s). OOM denotes out of memory.

| Method | Cora | CiteSeer | PubMed | Computers | Photo | CS | Physics |
|---|---|---|---|---|---|---|---|
| MLP | 7.7 ms / 5.27s | 8.1 ms / 5.37s | 7.8 ms / 5.52s | 8.8 ms / 5.45s | 8.4 ms / 5.34s | 10.5 ms / 8.18s | 14.6 ms / 12.78s |
| GCN | 82.2 ms / 6.1s | 84.2 ms / 6.1s | 85 ms / 6.13s | 85.2 ms / 7.07s | 83.6 ms / 6.08s | 85 ms / 9.68s | 90 ms / 13.8s |
| SGC | 89.5 ms / 4.96s | 74.7 ms / 4.86s | 80.6 ms / 5.28s | 109 ms / 5.21s | 85.9 ms / 4.96s | 213.6 ms / 8.01s | OOM |
| GAT | 534.8 ms / 13.06s | 313.6 ms / 13.36s | 314.6 ms / 13.97s | 441.3 ms / 24.62s | 309.8 ms / 15.96s | 454 ms / 21.87s | 436.9 ms / 40.9s |
| JKNet | 95.4 ms / 20.07s | 101.1 ms / 19.58s | 105.4 ms / 20.8s | 106.1 ms / 29.72s | 97.9 ms / 21.18s | 102.7 ms / 24.94s | 119.2 ms / 40.83s |
| APPNP | 86.7 ms / 11.6s | 86.3 ms / 11.98s | 85.5 ms / 11.97s | 92.1 ms / 15.75s | 86 ms / 12.19s | 90.5 ms / 17.36s | 99.6 ms / 25.89s |
| GPRGNN | 86.5 ms / 12.42s | 195.8 ms / 12.6s | 88.6 ms / 12.59s | 93.3 ms / 15.98s | 86.7 ms / 12.65s | 92 ms / 17.8s | 217.1 ms / 26.33s |
| $^{1.0}$GNN | 96 ms / 20.12s | 98.1 ms / 19.81s | 100.2 ms / 21.74s | 151.4 ms / 64.08s | 121.3 ms / 34.07s | 109.7 ms / 25.03s | 122.9 ms / 49.59s |
| $^{1.5}$GNN | 98.2 ms / 20.19s | 97 ms / 20.26s | 100.2 ms / 22.6s | 140.3 ms / 64.08s | 120 ms / 34.22s | 112.3 ms / 25.11s | 127.9 ms / 49.54s |
| $^{2.0}$GNN | 98.1 ms / 20.11s | 96.3 ms / 19.97s | 99.3 ms / 22.17s | 141 ms / 64.04s | 129.3 ms / 34.14s | 104.7 ms / 24.93s | 124.6 ms / 49.35s |
| $^{2.5}$GNN | 96.6 ms / 20.12s | 92.9 ms / 20.16s | 103 ms / 22.17s | 141.6 ms / 64.01s | 128.1 ms / 34.22s | 110.8 ms / 25.07s | 124 ms / 49.39s |

Table 13: Efficiency on heterophilic benchmark datasests. Averaged running time per epoch (ms) / averaged total running time (s).

| Method | Chameleon | Squirrel | Actor | Wisconsin | Texas | Cornell |
|---|---|---|---|---|---|---|
| MLP | 7.7 ms / 5.29s | 8 ms / 5.44s | 8.6 ms / 5.4s | 7.7 ms / 5.16s | 7.9 ms / 5.22s | 7.6 ms / 5.19s |
| GCN | 83.4 ms / 6.1s | 83.2 ms / 6.2s | 90.7 ms / 6.07s | 83.5 ms / 5.94s | 80.7 ms / 5.96s | 87.1 ms / 5.92s |
| SGC | 78.1 ms / 4.93s | 110.9 ms / 5.21s | 77.1 ms / 4.71s | 73.2 ms / 4.52s | 74.2 ms / 4.79s | 71.3 ms / 4.8s |
| GAT | 374.9 ms / 13.49s | 324.2 ms / 17.15s | 420 ms / 13.82s | 317.5 ms / 12.68s | 357.9 ms / 12.38s | 383.3 ms / 12.45s |
| JKNet | 102.4 ms / 21.15s | 101 ms / 22.84s | 97.2 ms / 21.24s | 98.5 ms / 21.07s | 103.6 ms / 20.92s | 102.2 ms / 20.79s |
| APPNP | 87.1 ms / 12.12s | 98.8 ms / 12.41s | 87.2 ms / 11.81s | 84.2 ms / 11.83s | 86 ms / 11.9s | 83.1 ms / 11.94s |
| GPRGNN | 93 ms / 12.98s | 86.1 ms / 13.01s | 94.2 ms / 13.01s | 84.3 ms / 12.66s | 92 ms / 12.64s | 89.1 ms / 12.6s |
| $^{1.0}$GNN | 107.3 ms / 22.43s | 116.3 ms / 30.92s | 117.8 ms / 23.6s | 94.5 ms / 18.47s | 92 ms / 18.83s | 92.7 ms / 18.97s |
| $^{1.5}$GNN | 97.2 ms / 22.54s | 115 ms / 31.04s | 119.2 ms / 23.47s | 93.3 ms / 18.64s | 90.8 ms / 19.09s | 94.9 ms / 18.88s |
| $^{2.0}$GNN | 98.7 ms / 22.37s | 114.8 ms / 31.14s | 100.8 ms / 23.73s | 92.2 ms / 19.09s | 92.5 ms / 18.72s | 98 ms / 18.64s |
| $^{2.5}$GNN | 97.9 ms / 22.38s | 115.9 ms / 31.09s | 97.3 ms / 23.77s | 92.8 ms / 19.03s | 91 ms / 18.84s | 90.7 ms / 18.83s |

## F.9 Experimental Results on Benchmark Datasets for 64 Hidden Units

Table 14: Results on heterophilic benchmark datasets for 64 hidden units. Averaged accuracy (%) for 20 runs. Best results outlined in bold and the results within 95% confidence interval of the best results are outlined in underlined bold.

| Method | Chameleon | Squirrel | Actor | Wisconsin | Texas | Cornell |
|---|---|---|---|---|---|---|
| MLP | $46.55_{\pm0.90}$ | $\mathbf{33.83}_{\pm0.59}$ | $38.40_{\pm0.76}$ | $93.91_{\pm2.47}$ | $\mathbf{87.51}_{\pm8.53}$ | $\mathbf{86.75}_{\pm8.22}$ |
| GCN | $34.74_{\pm2.62}$ | $25.68_{\pm1.17}$ | $30.86_{\pm1.51}$ | $65.93_{\pm5.47}$ | $58.56_{\pm13.28}$ | $46.81_{\pm4.28}$ |
| SGC | $34.57_{\pm4.71}$ | $24.39_{\pm1.54}$ | $35.50_{\pm2.09}$ | $62.87_{\pm8.92}$ | $50.62_{\pm5.60}$ | $29.44_{\pm14.83}$ |
| GAT | $43.33_{\pm1.53}$ | $30.07_{\pm0.99}$ | $33.44_{\pm2.45}$ | $66.57_{\pm4.69}$ | $50.69_{\pm12.89}$ | $42.62_{\pm13.37}$ |
| JKNet | $32.69_{\pm4.47}$ | $27.18_{\pm0.76}$ | $25.72_{\pm2.75}$ | $66.57_{\pm10.53}$ | $43.88_{\pm17.10}$ | $47.69_{\pm3.25}$ |
| APPNP | $35.09_{\pm3.18}$ | $28.15_{\pm0.93}$ | $32.28_{\pm1.75}$ | $66.30_{\pm1.60}$ | $69.00_{\pm4.53}$ | $54.88_{\pm3.85}$ |
| GPRGNN | $34.65_{\pm2.86}$ | $28.56_{\pm1.35}$ | $34.58_{\pm1.45}$ | $93.70_{\pm3.12}$ | $86.50_{\pm6.04}$ | $84.75_{\pm8.38}$ |
| $^{1.0}$GNN | $\underline{\mathbf{49.51}}_{\pm1.32}$ | $32.67_{\pm1.00}$ | $\mathbf{40.70}_{\pm0.88}$ | $\mathbf{95.23}_{\pm1.60}$ | $84.12_{\pm7.39}$ | $82.56_{\pm6.97}$ |
| $^{1.5}$GNN | $\mathbf{49.52}_{\pm1.15}$ | $33.14_{\pm1.10}$ | $39.82_{\pm1.54}$ | $94.03_{\pm2.26}$ | $86.94_{\pm6.99}$ | $\mathbf{86.89}_{\pm6.63}$ |
| $^{2.0}$GNN | $49.19_{\pm0.81}$ | $\underline{\mathbf{33.78}}_{\pm0.87}$ | $39.75_{\pm1.26}$ | $94.49_{\pm1.81}$ | $\mathbf{87.62}_{\pm6.64}$ | $85.56_{\pm7.25}$ |
| $^{2.5}$GNN | $48.93_{\pm0.74}$ | $33.31_{\pm1.27}$ | $39.47_{\pm1.20}$ | $92.13_{\pm2.16}$ | $\underline{\mathbf{87.25}}_{\pm5.57}$ | $80.56_{\pm5.28}$ |

---

[5]https://ogb.stanford.edu/docs/leader_nodeprop/#ogbn-arxiv

Table 15: Results on homophilic benchmark datasets for 64 hidden units. Averaged accuracy (%) for 20 runs. Best results are outlined in bold and the results within 95% confidence interval of the best results are outlined in underlined bold. OOM denotes out of memory.

| Method | Cora | CiteSeer | PubMed | Computers | Photo | CS | Physics |
|---|---|---|---|---|---|---|---|
| MLP | $49.05_{\pm0.82}$ | $50.67_{\pm1.25}$ | $80.32_{\pm0.40}$ | $70.58_{\pm0.82}$ | $79.44_{\pm0.79}$ | $89.48_{\pm0.50}$ | $92.84_{\pm0.62}$ |
| GCN | $77.65_{\pm0.42}$ | $64.72_{\pm0.52}$ | $84.13_{\pm0.12}$ | $\underline{\mathbf{84.56}}_{\pm0.79}$ | $90.16_{\pm0.88}$ | $91.14_{\pm0.10}$ | $94.75_{\pm0.04}$ |
| SGC | $70.32_{\pm1.87}$ | $\mathbf{65.77}_{\pm0.99}$ | $76.27_{\pm0.94}$ | $83.24_{\pm0.81}$ | $89.43_{\pm1.03}$ | $91.11_{\pm0.10}$ | OOM |
| GAT | $76.97_{\pm1.18}$ | $61.28_{\pm1.62}$ | $83.57_{\pm0.23}$ | $83.84_{\pm1.93}$ | $\underline{\mathbf{90.54}}_{\pm0.56}$ | $89.68_{\pm0.42}$ | $93.91_{\pm0.20}$ |
| JKNet | $78.77_{\pm0.79}$ | $64.62_{\pm0.80}$ | $82.82_{\pm0.16}$ | $82.22_{\pm1.32}$ | $88.43_{\pm0.53}$ | $90.48_{\pm0.13}$ | $93.75_{\pm0.32}$ |
| APPNP | $\mathbf{79.95}_{\pm0.72}$ | $\underline{\mathbf{65.56}}_{\pm0.64}$ | $84.00_{\pm0.22}$ | $83.83_{\pm0.78}$ | $\underline{\mathbf{90.50}}_{\pm0.59}$ | $91.90_{\pm0.12}$ | $\underline{\mathbf{94.84}}_{\pm0.08}$ |
| GPRGNN | $78.17_{\pm1.31}$ | $61.26_{\pm2.14}$ | $84.54_{\pm0.24}$ | $83.77_{\pm1.06}$ | $89.86_{\pm0.63}$ | $91.34_{\pm0.25}$ | $94.63_{\pm0.26}$ |
| $^{1.0}$GNN | $77.11_{\pm0.39}$ | $63.17_{\pm0.89}$ | $83.14_{\pm0.46}$ | $82.64_{\pm0.98}$ | $89.60_{\pm0.69}$ | $92.53_{\pm0.22}$ | $\underline{\mathbf{94.86}}_{\pm0.24}$ |
| $^{1.5}$GNN | $78.69_{\pm0.43}$ | $63.14_{\pm0.93}$ | $83.97_{\pm0.04}$ | $\mathbf{84.64}_{\pm1.42}$ | $\mathbf{90.67}_{\pm0.67}$ | $\mathbf{92.93}_{\pm0.14}$ | $\underline{\mathbf{94.93}}_{\pm0.12}$ |
| $^{2.0}$GNN | $79.06_{\pm0.41}$ | $63.92_{\pm1.14}$ | $84.24_{\pm0.27}$ | $\underline{\mathbf{84.57}}_{\pm0.96}$ | $90.17_{\pm0.88}$ | $\underline{\mathbf{92.74}}_{\pm0.26}$ | $\mathbf{95.05}_{\pm0.09}$ |
| $^{2.5}$GNN | $79.15_{\pm0.39}$ | $63.16_{\pm1.25}$ | $\mathbf{84.88}_{\pm0.09}$ | $83.84_{\pm0.71}$ | $89.05_{\pm0.85}$ | $92.31_{\pm0.19}$ | $94.92_{\pm0.10}$ |

## F.10 TRAINING CURVES FOR $p = 1$

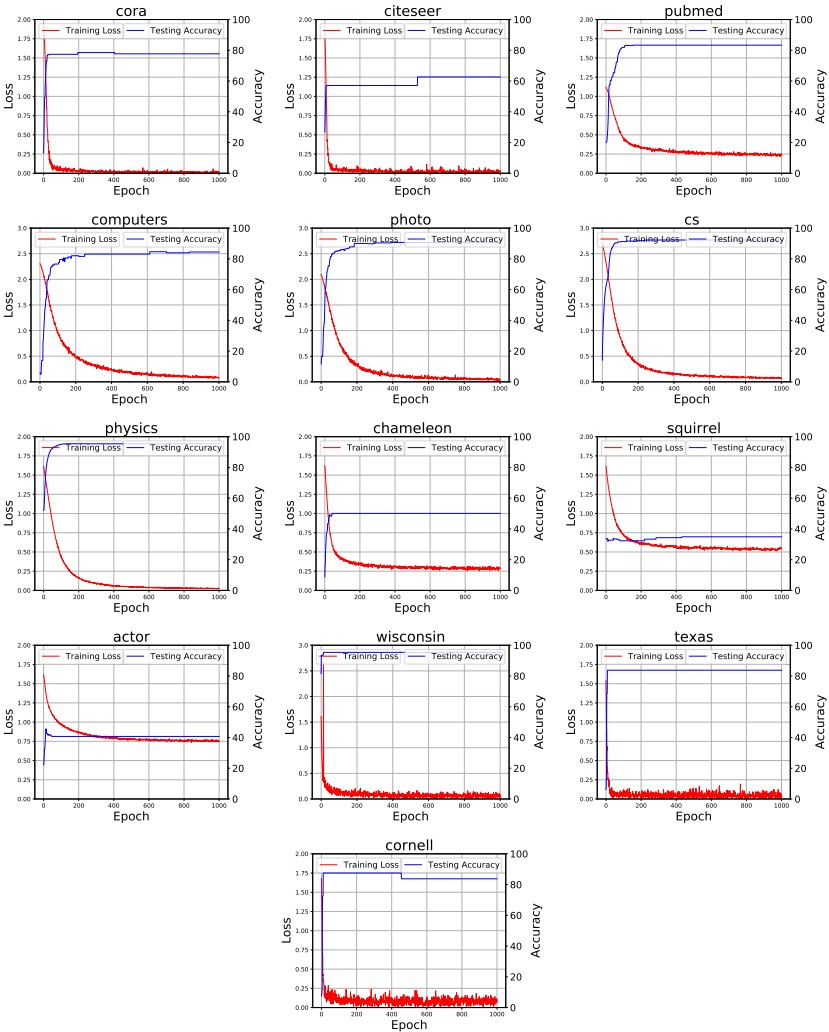

Figure 7: The curves of training loss and testing accuracy for $p = 1$.

### F.11  Visualization Results of Node Embeddings

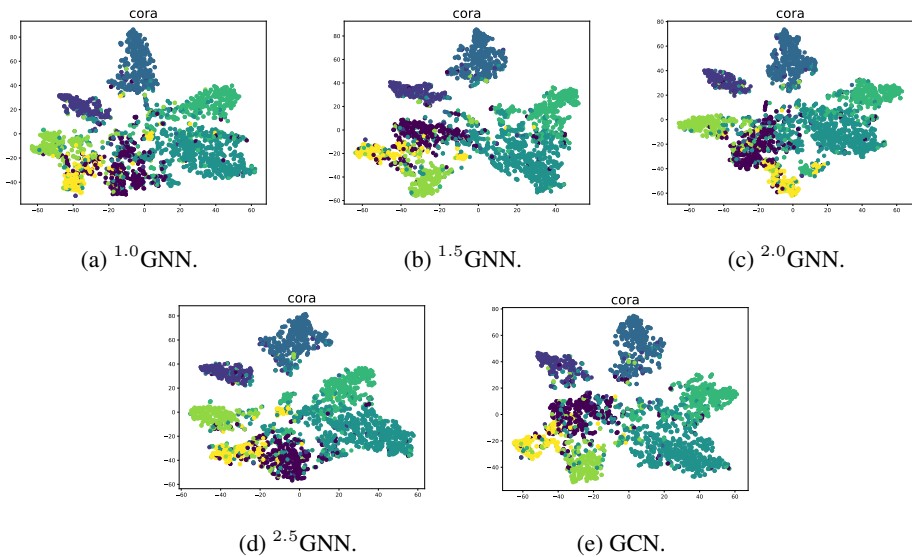

(a) $^{1.0}$GNN.  (b) $^{1.5}$GNN.  (c) $^{2.0}$GNN.

(d) $^{2.5}$GNN.  (e) GCN.

Figure 8: Visualization of node embeddings for Cora dataset using t-SNE (van der Maaten & Hinton, 2008)

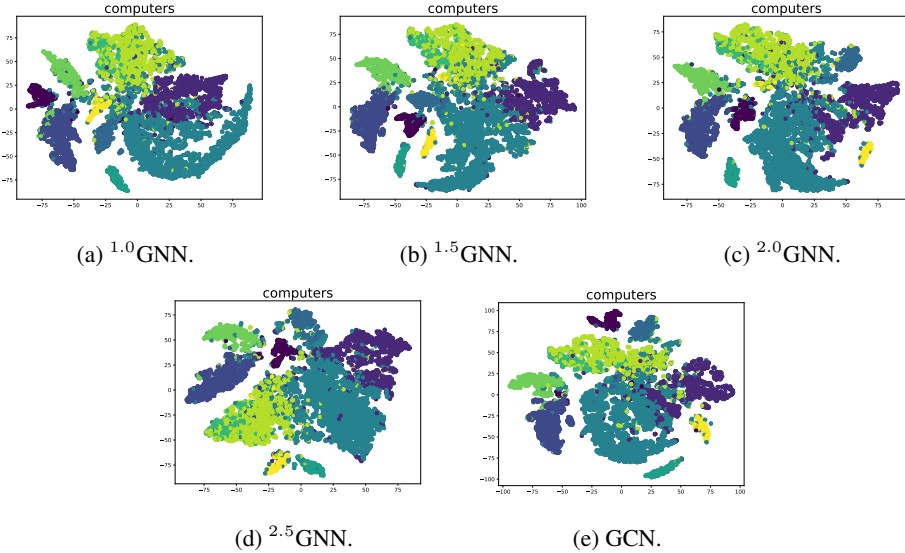

(a) $^{1.0}$GNN.  (b) $^{1.5}$GNN.  (c) $^{2.0}$GNN.

(d) $^{2.5}$GNN.  (e) GCN.

Figure 9: Visualization of node embeddings for Computers dataset using t-SNE.

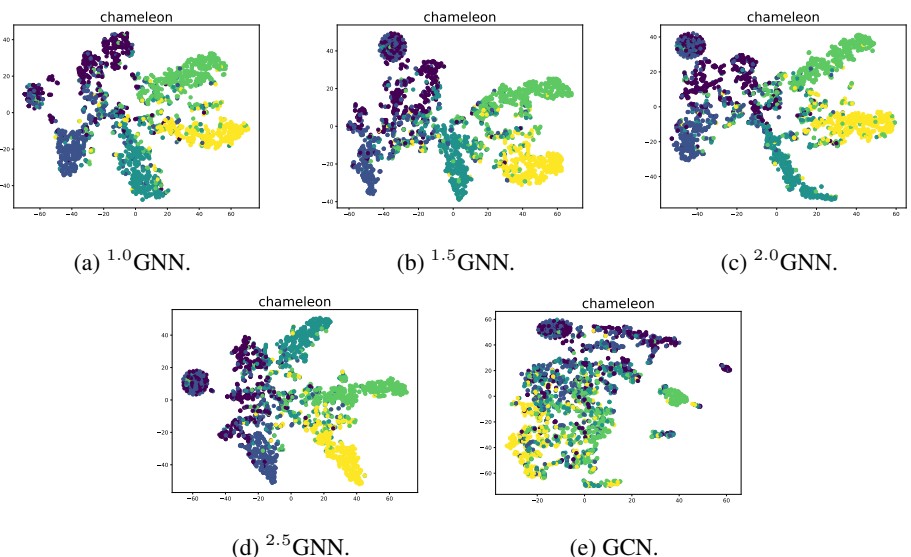

(a) $^{1.0}$GNN.  (b) $^{1.5}$GNN.  (c) $^{2.0}$GNN.

(d) $^{2.5}$GNN.  (e) GCN.

Figure 10: Visualization of node embeddings for Chameleon dataset using t-SNE.

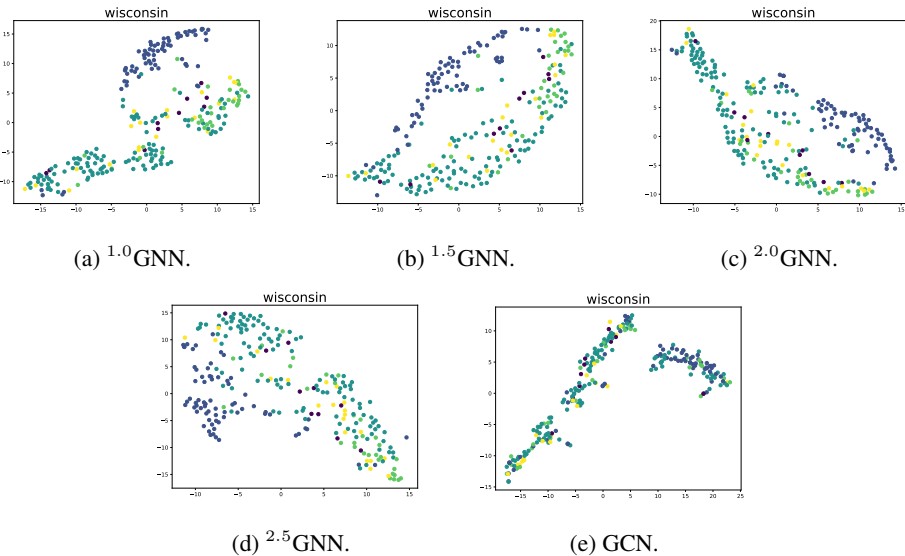

(a) $^{1.0}$GNN.  (b) $^{1.5}$GNN.  (c) $^{2.0}$GNN.

(d) $^{2.5}$GNN.  (e) GCN.

Figure 11: Visualization of node embeddings for Wisconsin Dataset using t-SNE.

