# OpenReview forum: "$p$-Laplacian Based Graph Neural Networks"
_ICLR.cc/2022/Conference — ICLR 2022 Submitted_

### Official Review · Reviewer_4UH6 · 2021-10-26

**Correctness:** 3
**Technical Novelty And Significance:** 2
**Empirical Novelty And Significance:** 3
**Recommendation:** 5
**Confidence:** 4

**Main Review:**

## Pros:

(+) The idea of designing new message-passing based on p-Laplacian is interesting.

(+) The empirical evaluation of cSBM is great.

(+) The high-level idea on the effect of different choices of p is inspiring.

## Cons:

(-) Most theorems derived in this paper are either well-known (Theorem 1) or not meaningful (Theorem 3).

(-) The overall clarity of the paper can be improved, especially in section 4.

(-) Some concerns regarding the choice of hyperparameters (see detail below).

## Detail comments:

-	Theorem 1 is well-known in the graph learning community, (see [4], equation (3) for example). I think there should be a proper reference for it.
-	Theorem 3 is not very meaningful, as the derived risk bound is greater than discarding the graph part ($\beta = 1,\; \alpha = 0$). This is because the analysis does not take the correlation of graphs into account. See [3] on how the authors show the graph convolution is helpful under cSBM assumption. I feel like the analysis and result here can be further improved for meaningful bound.
-	The clarity of section 4 and explanation of Theorem 2 should be improved. Regarding the explanation of Theorem 2, I cannot understand why the “weight effect” of p-Laplacian based message passing can help for heterophilic graphs. What does the “boundary of the lined nodes” refer to in the explanation? On the other hand, while I like the explanation of Proposition 1, I feel like the authors treat the matrix M as fixed in this section. However, according to their definition, it should depend on the input feature matrix. Please elaborate more on this point and improve the clarity of the text.
-	I find it weird for the choice of hyperparameter for all baseline methods. More precisely, the authors choose the hidden unit = 16 while the default and common choice is 64. I wonder why the authors detour from the standard choice.

### Typo:

1.	Page 2, the reference of the cSBM is incorrect. Should be [1].

2.	Page 5, in Remark 2: “In contrast, it is is similar with SGC…”, redundant “is” here.

### Additional questions:

1.	The matrix M in Theorem 2 depends on the node embedding F right? I assume it follows the specific aggregation rule of equation (12).

2.	Does the iteration (11)-(13) guarantee to converge?

## Reference:

[1] Contextual stochastic block models. Deshpande et al. NeurIPS 2018.

[2] Optimizing generalized pagerank methods for seed-expansion community detection, Li et al. NeurIPS 2019.

[3] Graph Convolution for Semi-Supervised Classification: Improved Linear Separability and Out-of-Distribution Generalization, Baranwal et al. ICML 2021.

[4] PREDICT THEN PROPAGATE: GRAPH NEURAL NETWORKS MEET PERSONALIZED PAGERANK, Klicpera et al. ICLR 2019.


**Summary Of The Paper:**

The authors propose a new message-passing design based on p-Laplacian. They demonstrate that their method works better in heterophilic graphs.

**Summary Of The Review:**

I have mixed feelings about this paper. On one hand, I like the newly designed message passing scheme and their implicit bias demonstrated in Proposition 1. Also, conducting experiments on cSBM is great for studying problems of heterophilic graphs. On the other hand, the drawbacks of the paper prevent me from supporting acceptance of it. I think the paper has great potential but requires a major revision.

---

> ### Author Response · Authors · 2021-11-16
> **Response to all concerns raised by the reviewer**
>
> We thank the reviewer for her/his valuable feedback and recommendations for improving the manuscript. We have done a major revision in the updated manuscript, the main changes are summarized in the "Common replies to all reviewers".
> We address the concerns raised in detail below.
>
> >C1: Theorem 1 is well-known in the graph learning community, (see [4], equation (3) for example). I think there should be a proper reference for it.
>
> Thanks a lot for pointing out the reference for Theorem 1! We have added [4] as a reference for Theorem 1. We also would like to clarify that Theorem 1 serves as a connection between our $p$-Laplacian message passing scheme and the personal PageRank and it is not our main contribution.
>
>
> >C2: Theorem 3 is not very meaningful, as the derived risk bound is greater than discarding the graph part ($\beta=1; \alpha=0$). This is because the analysis does not take the correlation of graphs into account. See [3] on how the authors show the graph convolution is helpful under cSBM assumption. I feel like the analysis and result here can be further improved for meaningful bound.
>
> We thank the reviewer for the suggestion of derivating a risk bound of $^p$GNNs that related to some graph properties, such as homophily levels, linear separability of original node features (as considered in [1]). We believed this is a valuable direction and deserves further investigations for future work.
>
> Meanwhile, our upper-bounding risk theorem of $^p$GNNs is helpful because it provides a glance at the effect of the hyperparameter $\mu$ on the performance of $^p$GNNs. It shows that the risk of $^p$GNNs is upper-bounded by the sum of three terms: the risk of label prediction using only the original node features $\mathbf{X}$, the norm of $p$-Laplacian diffusion on $\mathbf{X}$, and the magnitude of the noise in $\mathbf{X}$ and $\mu$ controls the trade-off between these three terms.
>
> We also would like to clarify that our main focus is on proposing a new message passing scheme based on $p$-Laplacian and showing how it could work as both low-pass and high-pass filters by spectral analysis. Please note that we do not take the upper-bounding risk theorem as one of our major contributions. Considering it as an aside contribution, we move it to Appendix C1. Besides, a bound considering some graph properties is beyond the scope of this work and will be left as important future work.
>
>
> >C3: The clarity of section 4 and explanation of Theorem 2 should be improved. Regarding the explanation of Theorem 2, I cannot understand why the “weight effect” of p-Laplacian based message passing can help for heterophilic graphs. What does the “boundary of the lined nodes” refer to in the explanation?
>
> We are aware that the explanation of Theorem 2 is not clear. When we say "the boundary of the linked nodes", it actually refers to "the distance between the linked nodes in the embedding space". We have updated Theorem 2 and used it to demonstrate the convergency of iteration (11)-(13) and we removed the old explanation of Theorem 2.
>
>
> >C4: On the other hand, while I like the explanation of Proposition 1, I feel like the authors treat the matrix M as fixed in this section. However, according to their definition, it should depend on the input feature matrix. Please elaborate more on this point and improve the clarity of the text.
>
> We thank the reviewer for pointing out the ambiguity of $\mathbf{M}$ in section 4. The matrix $\mathbf{M}$ is not fixed there and it is depends on the node embeddings, or more precisely, it is depends on the graph gradients as we defined in equation (12): $M{i,j} = W{i,j}\left\|\sqrt{\frac{W{i,j}}{D{i,i}}}\mathbf{F}{i,:} - \sqrt{\frac{W{i,j}}{D{j,j}}}\mathbf{F}{j,:}\right\|^{p-2} = W{i,j}\|\nabla f([i,j])\|^{p-2}$.
>
> We have updated section 4 and Proposition 1 to make sure the presentation more clear for $\mathbf{M}$.

---

> > ### Author Response · Authors · 2021-11-16
> > **Response to all concerns raised by the reviewer (part 2)**
> >
> > >C5: I find it weird for the choice of hyperparameter for all baseline methods. More precisely, the authors choose the hidden unit = 16 while the default can common choice is 64. I wonder why the authors detour from the standard choice.
> >
> > The reason we used 16 hidden units instead of 64 is that we found the experimental results for 16 hidden units are similar to the results for 64 hidden units. Our method is superior on heterophilic graphs for both 16 and 64 hidden units. Due to the limited budget of time and computational resources, especially we have conducted many experiments on a number of datasets with many repeating times (100 or 20) as well as we have conducted many experiments to verify the potential extensions of our model, we used 16 hidden units in our experiments. Here we present some experimental results for 64 hidden units to provide the reviewer a glance at the performance of all models on homophilic graphs and heterophilic benchmark graphs.
> >
> > Results on heterophilic benchmark datasets for 64 hidden units. Averaged accuracy (\%) for 20 runs.
> >
> > |Method|Chameleon|Squirrel|Actor|Wisconsin|Texas|Cornell|
> > |-|-|-|-|-|-|-|
> > | MLP | $46.55_{\pm 0.90}$ | $\mathbf{33.83}_{\pm 0.59}$ | $38.40_{\pm 0.76}$ | $93.91_{\pm 2.47}$ | $\mathbf{\underline{87.51}}_{\pm 8.53}$ | $\mathbf{\underline{86.75}}_{\pm 8.22}$ |
> > | GCN | $34.74_{\pm 2.62}$ | $25.68_{\pm 1.17}$ | $30.86_{\pm 1.51}$ | $65.93_{\pm 5.47}$ | $58.56_{\pm 13.28}$ | $46.81_{\pm 4.28}$ |
> > | SGC | $34.57_{\pm 4.71}$ | $24.39_{\pm 1.54}$ | $35.50_{\pm 2.09}$ | $62.87_{\pm 8.92}$ | $50.62_{\pm 5.60}$ | $29.44_{\pm 14.83}$ |
> > | GAT | $43.33_{\pm 1.53}$ | $30.07_{\pm 0.99}$ | $33.44_{\pm 2.45}$ | $66.57_{\pm 4.69}$ | $50.69_{\pm 12.89}$ | $42.62_{\pm 13.37}$ |
> > | JKNet | $32.69_{\pm 4.47}$ | $27.18_{\pm 0.76}$ | $25.72_{\pm 2.75}$ | $66.57_{\pm 10.53}$ | $43.88_{\pm 17.10}$ | $47.69_{\pm 3.25}$ |
> > | APPNP | $35.09_{\pm 3.18}$ | $28.15_{\pm 0.93}$ | $32.28_{\pm 1.75}$ | $66.30_{\pm 1.60}$ | $69.00_{\pm 4.53}$ | $54.88_{\pm 3.85}$ |
> > | GPRGNN | $34.65_{\pm 2.86}$ | $28.56_{\pm 1.35}$ | $34.58_{\pm 1.45}$ | $93.70_{\pm 3.12}$ | $86.50_{\pm 6.04}$ | $84.75_{\pm 8.38}$ |
> > | $^{1.0}$GNN | $\mathbf{\underline{49.51}}_{\pm 1.32}$ | $32.67_{\pm 1.00}$ | $\mathbf{40.70}_{\pm 0.88}$ | $\mathbf{95.23}_{\pm 1.60}$ | $84.12_{\pm 7.39}$ | $82.56_{\pm 6.97}$ |
> > | $^{1.5}$GNN | $\mathbf{49.52}_{\pm 1.15}$ | $33.14_{\pm 1.10}$ | $39.82_{\pm 1.54}$ | $94.03_{\pm 2.26}$ | $86.94_{\pm 6.99}$ | $\mathbf{86.89}_{\pm 6.63}$ |
> > | $^{2.0}$GNN | $49.19_{\pm 0.81}$ | $\mathbf{\underline{33.78}}_{\pm 0.87}$ | $39.75_{\pm 1.26}$ | $94.49_{\pm 1.81}$ | $\mathbf{87.62}_{\pm 6.64}$ | $85.56_{\pm 7.25}$ |
> > | $^{2.5}$GNN | $48.93_{\pm 0.74}$ | $33.31_{\pm 1.27}$ | $39.47_{\pm 1.20}$ | $92.13_{\pm 2.16}$ | $\mathbf{\underline{87.25}}_{\pm 5.57}$ | $80.56_{\pm 5.28}$ |

---

> > > ### Author Response · Authors · 2021-11-16
> > > **Response to all concerns raised by the reviewer (part 3)**
> > >
> > > Results on homophilic benchmark datasets for 64 hidden units. Averaged accuracy (\%) for 20 runs.
> > >
> > > | Method | Cora | CiteSeer | PubMed | Computers | Photo | CS | Physics|
> > > |-|-|-|-|-|-|-|-|
> > > | MLP | $49.05_{\pm 0.82}$ | $50.67_{\pm 1.25}$ | $80.32_{\pm 0.40}$ | $70.58_{\pm 0.82}$ | $79.44_{\pm 0.79}$ | $89.48_{\pm 0.50}$ | $92.84_{\pm 0.62}$ |
> > > | GCN | $77.65_{\pm 0.42}$ | $64.72_{\pm 0.52}$ | $84.13_{\pm 0.12}$ | $\mathbf{\underline{84.56}}_{\pm 0.79}$ | $90.16_{\pm 0.88}$ | $91.14_{\pm 0.10}$ | $94.75_{\pm 0.04}$ |
> > > | SGC | $70.32_{\pm 1.87}$ | $\mathbf{65.77}_{\pm 0.99}$ | $76.27_{\pm 0.94}$ | $83.24_{\pm 0.81}$ | $89.43_{\pm 1.03}$ | $91.11_{\pm 0.10}$ | OOM |
> > > | GAT | $76.97_{\pm 1.18}$ | $61.28_{\pm 1.62}$ | $83.57_{\pm 0.23}$ | $83.84_{\pm 1.93}$ | $\mathbf{\underline{90.54}}_{\pm 0.56}$ | $89.68_{\pm 0.42}$ | $93.91_{\pm 0.20}$ |
> > > | JKNet | $78.77_{\pm 0.79}$ | $64.62_{\pm 0.80}$ | $82.82_{\pm 0.16}$ | $82.22_{\pm 1.32}$ | $88.43_{\pm 0.53}$ | $90.48_{\pm 0.13}$ | $93.75_{\pm 0.32}$ |
> > > | APPNP | $\mathbf{79.95}_{\pm 0.72}$ | $\mathbf{\underline{65.56}}_{\pm 0.64}$ | $84.00_{\pm 0.22}$ | $83.83_{\pm 0.78}$ | $\mathbf{\underline{90.50}}_{\pm 0.59}$ | $91.90_{\pm 0.12}$ | $\mathbf{\underline{94.84}}_{\pm 0.08}$ |
> > > | GPRGNN | $78.17_{\pm 1.31}$ | $61.26_{\pm 2.14}$ | $84.54_{\pm 0.24}$ | $83.77_{\pm 1.06}$ | $89.86_{\pm 0.63}$ | $91.34_{\pm 0.25}$ | $94.63_{\pm 0.26}$ |
> > > | $^{1.0}$GNN | $77.11_{\pm 0.39}$ | $63.17_{\pm 0.89}$ | $83.14_{\pm 0.46}$ | $82.64_{\pm 0.98}$ | $89.60_{\pm 0.69}$ | $92.53_{\pm 0.22}$ | $\mathbf{\underline{94.86}}_{\pm 0.24}$ |
> > > | $^{1.5}$GNN | $78.69_{\pm 0.43}$ | $63.14_{\pm 0.93}$ | $83.97_{\pm 0.04}$ | $\mathbf{84.64}_{\pm 1.42}$ | $\mathbf{90.67}_{\pm 0.67}$ | $\mathbf{92.93}_{\pm 0.14}$ | $\mathbf{\underline{94.93}}_{\pm 0.12}$ |
> > > | $^{2.0}$GNN | $79.06_{\pm 0.41}$ | $63.92_{\pm 1.14}$ | $84.24_{\pm 0.27}$ | $\mathbf{\underline{84.57}}_{\pm 0.96}$ | $90.17_{\pm 0.88}$ | $\mathbf{\underline{92.74}}_{\pm 0.26}$ | $\mathbf{95.05}_{\pm 0.09}$ |
> > > | $^{2.5}$GNN | $79.15_{\pm 0.39}$ | $63.16_{\pm 1.25}$ | $\mathbf{84.88}_{\pm 0.09}$ | $83.84_{\pm 0.71}$ | $89.05_{\pm 0.85}$ | $92.31_{\pm 0.19}$ | $94.92_{\pm 0.10}$ |
> > >
> > > As we can see from the tables, we have obtained similar results for 64 hidden units with the cases for 16 hidden units. $^p$GNNs significantly outperform several state-of-the-art GNN architectures on heterophilic benchmarks while achieving competitive performance on homophilic benchmarks. Therefore, considering the limited budget of time and computational resources, we use 16 hidden units in our experiments.
> > >
> > > We have included the results on heterophilic and homophilic benchmark datasets for 64 hidden units in Appendix F9. Please also refer to Tables 14 and 15 in Appendix F9.
> > >
> > > ##Typo:
> > > 1. Page 2, the reference of the cSBM is incorrect. Should be [1].
> > >
> > > 2. Page 5, in Remark 2: “In contrast, it is is similar with SGC…”, redundant “is” here.
> > >
> > > We thank the reviewer for pointing out these typos. We have corrected all of them in the new version.
> > >
> > >
> > > ##Additional questions:
> > > >A1: The matrix M in Theorem 2 depends on the node embedding F right? I assume it follows the specific aggregation rule of equation (12).
> > >
> > > Yes, your understanding is correct. We have updated Theorem 2 to make sure there is clear for $\mathbf{M}$.
> > >
> > >
> > > >A2: Does the iteration (11)-(13) guarantee to converge?
> > >
> > > We give a proof in Theorem 2 to show that for $p > 1$ and some proper $\mu$ that is chosen depends on the input graph and $p$, the loss of the objective function (8) is guaranteed to decline after taking one step $p$-Laplacian message passing. The theorem demonstrates that the iteration (11)-(13) is guaranteed to converge for $p > 1$ and some proper $\mu$ that chosen depends on the input graphs and $p$.
> > >
> > > Reference
> > >
> > > [1] Graph Convolution for Semi-Supervised Classification: Improved Linear Separability and Out-of-Distribution Generalization, Baranwal et al. ICML 2021.

---

> > > > ### Comment · Reviewer_4UH6 · 2021-11-25
> > > > **Re:**
> > > >
> > > > Thanks for the detail response. Partial of my concerns are addressed yet some still persist.
> > > >
> > > > ## Inconsistent experimental results compare to Chien et al. 2021.
> > > >
> > > > As also pointed out my Reviewer Pzid, the experiment results seem to be inconsistent to those in Chien et al. 2021. More specifically, Chien et al. 2021 showed that GPRGNN achieves accuracy around 50% on Squirrel and 67% on Chameleon, which are both much **higher** than MLP. In contrast, the authors show that GPRGNN only achieves 31% and 43% on Squirrel and Chameleon respectively. Both of which are much **lower** than MLP. I wonder why this is the case? I've tried my best to look into the code and checked the github repo of GPRGNN paper [1]. I find that both results are reproducible. That being said, using the exact code from GPRGNN github repo can reproduce their Table 2 and using the code provided by the authors can reproduce the results in Table 1 of this paper. It would be great if the authors can provide reasons why there's such an inconsistency between these two implementations.
> > > >
> > > > ## Regarding the theoretical contribution
> > > >
> > > > First, I thank the genuine explanation by the authors. Unfortunately, it does not change my opinion on the theoretical contributions. The authors admitted that Theorem 1 is known to graph learning community. Regarding Theorem 3, I still think the explanation is inadequate. Note that one should be really careful when explaining relations of different terms using **upper bound**, as it could be a very loose (void) upper bound. For example, I can easily add a (positive) irrelevant term to the upper bound in Theorem 3, which obviously still makes the bound hold. However, any conclusion drawn from that irrelevant term is inadequate. As I mentioned, the derived risk bound is greater than discarding the graph part. Hence, the stated conclusion on $\mu$ or the effect on the graph part is meaningless. I think the paper should be improved along this direction if the authors want to claim their work with theoretical contribution.
> > > >
> > > > According to these two points, my judgement on this paper remain the same for now.
> > > >
> > > > ### Reference
> > > >
> > > > [1] https://github.com/jianhao2016/GPRGNN

---

> > > > > ### Author Response · Authors · 2021-11-27
> > > > > **Reply to the further comments raised by the reviewer 4UH6 (part 1)**
> > > > >
> > > > > Thank you very much for your further comments! We address the raised concerns and the misunderstanding on the upper-bounding risk below.
> > > > >
> > > > > >Inconsistent experimental results compare to [1].
> > > > >
> > > > > **The inconsistency comes from the different data preprocessing ways between our work and [1].** As mentioned by reviewer 4UH6, both the experimental results of our work and [1] are reproducible. We have carefully checked the public codes of GPRGNN [2] and found that the inconsistency between our work and [1] comes from the different data preprocessing ways adopted. We used the original version of the benchmark datasets from the Pytorch Geometric 1.7.1 [3] where they did **not** transform the directed graphs Chameleon and Squirrel into **undirected**.
> > > > > On the contrary, [1] has transformed Chameleon and Squirrel into undirected in their data preprocessing procedure (please check `line 157` in `dataset_utils.py ` from [2] and the ``last paragraph on Page 19`` of the GPRGNN paper [1]). As a result, our homophily levels of Chameleon and Squirrel are different from [1]. The homophily levels of all graphs used in our work were given in Table 2 in Appendix E2 where we showed that the homophily levels of Chameleon and Squirrel are **0.247** and **0.216**, which is much different from the homophily levels in [1] where they are **0.024** and **0.055**. The homophily levels of Chameleon and Squirrel in our case indicate they are **weak heterophilic** graphs (the homophily level $\mathcal{H}_\mathcal{G} \approx \frac{1}{|\mathcal{Y}|}$), while the homophily levels in the case of [1] indicate they are **strong heterophilic** graphs (the homophily level $\mathcal{H}_\mathcal{G} \ll \frac{1}{|\mathcal{Y}|}$). Therefore, it is reasonable for MLP to work well on Chameleon and Squirrel in our case and it was significantly outperformed by GPRGNN on Chameleon and Squirrel in [1].
> > > > >
> > > > > **Consistent experimental results on cSBM compared to [1].** On the other hand, our experimental results on cSBM datasets are consistent with the results in [1] where we both showed GPRGNN dominates MLP and other GNN baselines on **strong heterophilic** ($\phi \leq 0.5$) graphs generated by cSBM. Moreover, the results on cSBM also demonstrated the superior performance of $^p$GNNs on heterophilic graphs.
> > > > >
> > > > > >Regarding the theoretical contribution.
> > > > >
> > > > > **Theorem 1 and the Theorem of upper-bounding risk (termed as Theorem 4 in our new manuscript) are not our major theoretical contributions.** Please note that the theoretical contributions we claimed are **Proposition 1** which demonstrates $p$-Laplacian message passing can adaptively work as low-pass and high-pass filters and **Theorem 3** which demonstrates $p$-Laplacian message passing is equivalent to a polynomial filter defined on the spectral domain of $p$-Laplacian.
> > > > >
> > > > > **Theorem 1 serves as a connection between the personal PageRank used by APPNP [3] and $p$-Laplacian message passing.** As we explained in the previous response, we used Theorem 1 to show the personal PageRank is a special case of our message passing scheme with $p=2$. Based on Theorem 1 we further discussed the connection of our method and APPNP. We did not claim Theorem 1 as a theoretical contribution in our manuscript.

---

> > > > > > ### Author Response · Authors · 2021-11-27
> > > > > > **Reply to the further comments raised by the reviewer 4UH6 (part 2)**
> > > > > >
> > > > > > > As I mentioned, the derived risk bound is greater than discarding the graph part ($\beta_{i,i}=1; \alpha_{i,i}=0$). Hence, the stated conclusion on or the effect on the graph part is meaningless.
> > > > > >
> > > > > > We would like to first recall our upper-bounding risk with $K=1$ for better illustration and denote the bound by $UPB(\alpha_{i,i}, \beta_{i,i})$, i.e.
> > > > > > <
> > > > > > $$\frac{1}{N}\sum_{i=1}^N\left|y_i - \tilde{y}_i\right| \leq UPB(\alpha_{i,i}, \beta_{i,i}) := \frac{1}{N}\sum_{i=1}^N\beta_{i,i}\left|y_i - \sigma(X_{i,:})\right| + \frac{L}{N}\sum_{i=1}^N\alpha_{i,i}\left\|\Delta_pF_{i,:}\right\| + \frac{L}{N}\sum_{i=1}^N(1 -\beta_{i,i})\left\|\epsilon_{i,:}\right\|.$$>
> > > > > >
> > > > > > Please note that the comment that "the derived risk bound is greater than discarding the graph part" is a bit weird since it is obviously incorrect based on the literal understanding. Therefore, we kindly took this comment as a typo in the first round of rebuttal. Since this comment was raised again by the reviewer, we would like to address it here in detail based on its literal understanding, i.e. $UPB(0, 1)$ means the risk bound that discards the graph part and the comment means that $UPB(\alpha_{i,i}, \beta_{i,i}) > UPB(0, 1)$ for all $\alpha_{i,i} > 0$. Below we will explain in detail why this statement is incorrect.
> > > > > >
> > > > > > Note that $\alpha_{i,i} = 1/\left(\sum_j\frac{M_{i,j}}{d} + \frac{2\mu}{p}\right) > 0$, $\beta_{i,i} = \frac{2\mu}{p}\alpha_{i,i} < 1$, and $\alpha{i,i}\sum_jM{i,j}/d + \beta{i,i} = 1, \tilde{y}i = \alpha{i,i}\sum_j\frac{M{i,j}}{d}\sigma(X{j,:}) + \beta{i,i}\sigma(X{i,:})$ .
> > > > > >
> > > > > > 1. When $\alpha_{i,i}, \rightarrow 0$ then $\beta_{i,i} \rightarrow 1$, the upper-bounding risk becomes $UPB(0, 1) = \frac{1}{N}\sum_{i=1}^N\left|y_i - \sigma(X_{i,:})\right|$ and it has discarded the graph part.
> > > > > >
> > > > > > 2. When $\alpha_{i,i} > 0$ then $\beta_{i,i} < 1$, we have
> > > > > > $$UPB(\alpha_{i,i}, \beta_{i,i}) - UPB(0, 1) = \frac{L}{N}\sum_{i=1}^N\alpha_{i,i}\left\|\Delta_pF_{i,:}\right\| + \frac{L}{N}\sum_{i=1}^N(1 -\beta_{i,i})\left\|\epsilon_{i,:}\right\| - \frac{1}{N}\sum_{i=1}^N(1-\beta_{i,i})\left|y_i - \sigma(X_{i,:})\right|,$$
> > > > > > and it is not always positive, i.e. $UPB(\alpha_{i,i}, \beta_{i,i}) > UPB(0, 1)$ for all $\alpha_{i,i} > 0$ is **not** true.
> > > > > >
> > > > > > Therefore, the comment that "*the derived risk bound is greater than discarding the graph part. Hence, the stated conclusion on or the effect on the graph part is meaningless.*" is **incorrect**.
> > > > > >
> > > > > > Please let us know if we misunderstood your comment, thank you!
> > > > > >
> > > > > >
> > > > > > Reference
> > > > > >
> > > > > > [1] Adaptive Universal Generalized PageRank Graph Neural Network. Chien et al. ICLR 2021.
> > > > > >
> > > > > > [2] https://github.com/jianhao2016/GPRGNN
> > > > > >
> > > > > > [3] https://pytorch-geometric.readthedocs.io/en/latest/_modules/torch_geometric/datasets/wikipedia_network.html#WikipediaNetwork

---

> > > > > > > ### Comment · Reviewer_4UH6 · 2021-11-27
> > > > > > > **Re**
> > > > > > >
> > > > > > > Sorry I may not convey my point very clearly. It is also possible that I misinterpret the results. Yet, let me rephrase my question in a more clear way.
> > > > > > >
> > > > > > > 1. According to the Theorem 3, it seems like there's no specific relation between $\alpha,\beta$. I understand that in the paper, the authors mentioned that one can choose $\beta = \frac{2\mu}{p}\alpha$ in equation (13). Is this the necessary condition for the Theorem 3 to hold? Note that in the original description of the Theorem 3, there's no specific requirement on $\alpha, \beta$. The authors did not specify that their $\alpha, \beta$ is based on equation (13) or not. Hence, I simply assume that they are in the range $\alpha,\beta \in (0,1)$ and no further constraints.
> > > > > > >
> > > > > > > 2. If there's indeed a relation between $\alpha, \beta$, why not plug in this relation to reduce the redundant parameters? If not, my point is that by keeping the same $\beta$, one seems can make $\alpha = 0$ to make the bound smaller comparing to the case $\alpha > 0$. That's what I mean by "discarding the graph part".
> > > > > > >
> > > > > > >
> > > > > > > Please let me know if I still misunderstand the results. I'm more than happy to correct my statements and judgement. Thanks again for the timely response.

---

> > > > > > > > ### Author Response · Authors · 2021-11-27
> > > > > > > > **Reply to the Reviewer 4UH6**
> > > > > > > >
> > > > > > > > Thank you so much for your quick reply! Your suggestions have been very helpful for improving our work, and we have further refined the manuscript accordingly. We would like to address your questions below.
> > > > > > > >
> > > > > > > > >1. According to your Theorem 3, it seems like there's no specific relation between $\alpha$ and $\beta$. I understand that in the paper, the authors mentioned that one can choose in $\beta = \frac{2\mu}{p}\alpha$ equation (13). Is this the necessary condition for the Theorem 3 to hold? Note that in the original description of the Theorem 3, there's no specific requirement on $\alpha, \beta$. The authors did not specify that their  is $\alpha, \beta$ based on equation (13) or not. Hence, I simply assume that they are in the range $\alpha, \beta \in (0, 1)$ and no further constraints.
> > > > > > > > >
> > > > > > > > >2. If there's indeed a relation between $\alpha, \beta$, why not plug in this relation to reduce the redundant parameters? If not, my point is that by keeping the same $\beta$, one seems can make $\alpha = 0$ to make the bound smaller comparing to the case $\alpha > 0$. That's what I mean by "discarding the graph part".
> > > > > > > >
> > > > > > > > Yes, the relationship between $\alpha$ and $\beta$ given by equation (13) is a necessary condition for Theorem 3 to hold. We are sorry for the missing condition for $\alpha$ and $\beta$ in Theorem 3 in our old manuscript so that it causes your misunderstanding. We have added equation (13) as the condition on $\alpha$ and $\beta$, i.e.
> > > > > > > > $$\alpha_{i,i} = 1 \big/ \left(\sum_{j=1}^N\frac{M_{i,j}}{D_{i,i}} + \frac{2\mu}{p}\right), \quad \beta_{i,i} = \frac{2\mu}{p}\alpha_{i,i},$$
> > > > > > > > in our new manuscript. Please refer to the upper-bounding risk theorem (termed as Theorem 4) in Appendix B.1 in the updated version.
> > > > > > > >
> > > > > > > > Many thanks again for your valuable comments!

---

> > > > > > > > ### Author Response · Authors · 2021-11-30
> > > > > > > > **Thanks again for your comments!**
> > > > > > > >
> > > > > > > > Dear Reviewer:
> > > > > > > >
> > > > > > > > Thanks again for your valuable comments!   It has been very helpful for improving the work.
> > > > > > > > Would be great if you can let us know if your questions (1, 2) above have been properly addressed.
> > > > > > > > We are looking forward to your feedback.
> > > > > > > >
> > > > > > > > Best,
> > > > > > > > The authors.

---

> ### Author Response · Authors · 2021-11-25
> **Reply to Reviewer 4UH6**
>
> Dear Reviewer 4UH6,
>
> Thank you very much for your valuable comments! We are wondering if your concerns have been addressed properly. Please let us know if you have any further questions after reviewing our answers.
>
> Best regards,
>
> The authors

---

> ### Author Response · Authors · 2021-11-29
> **Reply to Reviewer 4UH6 (any new concerns?)**
>
> Dear reviewer 4UH6,
>
> Thank you again for your constructive comments! We want to know if your concerns have been properly addressed. If you have any other (unresolved or new) concerns after reviewing our feedback and paper, please let us know. We are very happy to answer them.
>
> Best,
>
> The authors

---

### Official Review · Reviewer_Pzid · 2021-11-03

**Correctness:** 2
**Technical Novelty And Significance:** 2
**Empirical Novelty And Significance:** 2
**Recommendation:** 3
**Confidence:** 4

**Main Review:**

The paper proposes general p-GNNs architecture under the p-Laplacian regularization framework
with comprehensive experimental results. The paper is well written and the theoretical part is well
organized, though not significant. However, we still have some concerns.

Strength:
(1) The author proposes a new general GNN architecture based on p-Laplacian message passing,
which claims to work on both heterophily and homophily settings.
(2) Relations of p-Laplacian message passing with low-pass and high-pass filters have been
explained.
(3)Experiments are comprehensive and inclusive.

Weakness:
(1) More insights on how to choose parameter p should be given. From the upper bound risk theorem, we only know that for K=1, the risk can be controlled by \mu which further depends on whether the topological information of graphs is useful or not. However, for K >=2, we do not see the relations. More insights are welcomed.

(2) The stationary point for the objective function in equation 8 is problematic when $p < 2$ and $\nabla f =0$. In particular, when $p=1$  and $\nabla f =0$, the Laplacian diffusion should have a step function that is not continuous at $\nabla f =0$. The authors seem not to have given a good discussion on this aspect.

(3) Moreover, when $p<2$, the graph diffusion operator is not Lip-continuous, which violates the assumption made by the upper-bounding risks theorem. So I do not see how that theorem can be applied here.

(4) For p=1, the diffusion becomes a step function (piecewise constant), how can the model be trained? Can the authors also provide the training curve for the p=1 case?

(5) P-GNN can only get comparable performances on homophilic benchmark datasets (both
accuracy and entropy experiments).

(6) The authors claim that p-lap can handle the heterophilic case. And the explanation is that the p-lap allows big change on the class boundary which can be viewed as a high-pass filter. I do not think this argument is very convincing. Can the authors have TSNE of the node embeddings for the heterophilic case so that we can get some visualization of the results?


Some typos:
(1)Page 2, last paragraph: Heterphily -> Heterophily
(2)Section 3.1, R^N should be changed into N

**Summary Of The Paper:**

The paper derives the p-Laplacian message passing formula under the p-Laplacian based regularization framework and further proposes p-GNN architecture. Authors further justify the relations of p-Laplacian message passing with low and high-pass filters and the upper bound of
one layer risk of p-GNNs. Experiments show the superiority of p-GNNs on both heterophilic and homophilic settings. However, we still have some concerns for the paper before further evaluation.

**Summary Of The Review:**

See the main review.

---

> ### Author Response · Authors · 2021-11-16
> **Response to all concerns raised by the reviewer**
>
> We thank the Reviewer for the valuable feedback for improving the manuscript.
> We would like to note that there are major misunderstandings regarding
> some of the critical points raised by the reviewer.
>
> We would like to start by addressing the concerns on the Laplacian diffusion operator raised by the reviewer:
>
> >C1: The stationary point for the objective function in equation 8 is problematic when $p < 2$ and $\nabla f = 0$. In particular, when $p=1$ and $\nabla f=0$, the Laplacian diffusion should have a step function that is not continuous at $\nabla f=0$. The authors seem not to have given a good discussion on this aspect.
>
> Note that this comment is mainly due to a major misunderstanding of the $p$-Laplacian diffusion.
> For $1 < p < 2$, the raised concern would never happen. When $p=1$, there could be a subtle problem which will be explained in detail below.
>
> When $1 < p < 2$, the Laplacian diffusion is continuous and the stationary point for the objective function is not problematic.
>
> When $p = 1$, the stationary point could be problematic when $f$ is a real-valued function, i.e. one channel graph signals. Specifically, when the node embeddings $\mathbf{F}$ are one dimensional vectors (one channel signals), $\Delta_1 f$ is a step function, which could make the stationary condition of the objective function in equation (8) become problematic. Additionally, $\Delta_1f$ is not continuous at $\|(\nabla f)([i, j])\| = 0$. Therefore, $p = 1$ is not allowed when the embeddings of nodes are one dimensional vectors.
>
> On the other hand, note that there is a Frobenius norm in $\Delta_pf$. When $f$ is a vector-valued function, i.e. node embeddings $\mathbf{F}$ are multi-dimensional vectors, the step function in $\Delta_1f$ only exists along the axes. The stationary condition could be fine if the node embeddings $\mathbf{F}$ are not vectors that have only one non-zero element, which is true for many graphs in practice. $p = 1$ may work for these graphs, which is also verified by the empirical studies.
>
> Overall, we suggest to use $p > 1$ in practice but $p = 1$ may work for graphs with multiple channel signals. In order to provide a comprehensive empirical study, we conduct experiments for $p > 1$ (e.g., $p = 1.5, 2, 2.5$) and $p = 1$.
>
>
> >C2: Moreover, when $p < 2$, the graph diffusion operator is not Lip-continuous, which violates the assumption made by the upper-bounding risks theorem. So I do not see how that theorem can be applied here.
>
> The reviewer may have misunderstood the assumption made by the upper-bounding risk theorem. First, the graph diffusion is Lip-continues when $1 < p < 2$. Second, the Lip-continuous assumption made by the upper-bounding risk theorem is on the prediction function $\sigma(\cdot)$ instead of the graph diffusion operator. The Lip-continuous assumption in the theorem of upper-bounding risk of $^p$GNNs has not been violated. To make sure there is no violation of our assumption, we restricted $p > 1$ in our updated version of the upper-bounding risk theorem.
>
> Please also note that the theorem of the upper-bounding risk of $^p$GNNs is moved to Appendix and is referred to as Theorem 4 in our updated manuscript.
>
> >C3: For $p=1$, the diffusion becomes a step function (piecewise constant), how can the model be trained? Can the authors also provide the training curve for the $p=1$ case?
>
> The diffusion does not become a step function for $p=1$ for the datasets used in our experiments. Note that there is a Frobenius norm in the Laplacian diffusion. When $p = 1$, the diffusion does not become a step function when $f$ is a vector-valued function and the step function in $\Delta_1f$ only exists on the axes. The stationary condition could be fine if the node embeddings $\mathbf{F}$ are not a matrix of vectors that has only one non-zero element, which is true for the datasets used in our experiments.
>
> We have added the training curves (in terms of training loss and testing accuracy) for $p = 1$ in Appendix F10. Please refer to Figure 7. From the results, one can observe that the training loss declines stably and the test accuracy increases smoothly for all datasets.

---

> > ### Author Response · Authors · 2021-11-16
> > **Response to all concerns raised by the reviewer (part 2)**
> >
> > >C4: More insights on how to choose parameter p should be given. From the upper bound risk theorem, we only know that for K=1, the risk can be controlled by \mu which further depends on whether the topological information of graphs is useful or not. However, for K >=2, we do not see the relations. More insights are welcomed.
> >
> > We have updated the upper-bounding risk so that it works for all $K >= 1$. Similar to the cases that $K=1$, the risk for $K \geq 2$ can be controlled by $\mu$ which provides a trade-off between the risk of label prediction based on only the node features, the norm of $p$-Laplacian diffusion on the node features, and the magnitude of the noise in the node features.
> >
> > Please note that the main contributions of this work lie in proposing a new methodology and spectral analysis. We did not claim the upper-bounding risk theorem to be the main contribution and it was moved to Appendix C.1 in the
> > updated version.
> >
> >
> > >C5: P-GNN can only get comparable performances on homophilic benchmark datasets (both accuracy and entropy experiments).
> >
> > We would like to kindly remind the reviewer that the main focus of this paper is on heterophilic graphs. Homophilic graphs are not our major focus. The experimental results demonstrated our method significantly outperforms baselines on heterophilic benchmarks while achieving competitive performance on homophilic benchmarks. We think this should be an advantage of our method instead of a drawback.
> >
> >
> > >C6: The authors claim that p-lap can handle the heterophilic case. And the explanation is that the p-lap allows big change on the class boundary which can be viewed as a high-pass filter. I do not think this argument is very convincing. Can the authors have TSNE of the node embeddings for the heterophilic case so that we can get some visualization of the results?
> >
> > Proposition 1 has rigorously demonstrated how $p$-Laplacian message passing acts as low-pass filters or both as low-pass and high-pass filters for different $p$s. We present some visualization results of $^p$GNNs using TSNE on homophilic graphs (Cora and Computers datasets) and heterophilic graphs (Chameleon and Wisconsin datasets) in Appendix F11. Please refer to Figures 8 to 11. From the results, one can observe that the node embeddings generated by $^p$GNNs are much better than GCN.
> >
> >
> > Some typos:
> > 1. Page 2, last paragraph: Heterphily -> Heterophily
> > 2. Section 3.1, R^N should be changed into N
> >
> > We thank the reviewer for pointing out the typos in our manuscript. We have corrected all of the mentioned topos in the new version.

---

> > > ### Comment · Reviewer_Pzid · 2021-11-17
> > > **Further comments**
> > >
> > > Thank the authors for the detailed explanations. Based on the authors' response, I understand the diffusion operator when p=1 and 1<p<2 better. However, I still have the following four concerns.
> > >
> > > 1. Proposition 1 is not very clear to me. I do not completely understand what low-pass and high-pass filters mathematically mean. To the best of my knowledge, this is not a commonly-used definition. We either use low-pass or high-pass instead of both. So, it causes confusion. Moreover the statement also says when p=2, it is always low-pass and high-pass, which I am very confused. Because, the case with p=2 is well known to be low-pass as it corresponds to PageRank-type diffusion and gives the model APPNP [5].
> > >
> > > 2. Although the experimental setting used in this paper is the same as [1], the performance listed in the table is greatly different from that in [1]. Especially for the heterophilic ones, MLP performs best over all datasets in this work. However, in [1] the non-MLP baselines work much better. Can the authors explain the reason? Also for those heterophilic networks, the network structures are not non-informative as claimed by the authors. Many of these networks essentially correspond to the node structural role classification case, where the network structures are crucial as illustrated in [2].
> > >
> > > 3. The p=1 case was also investigated in [3] but this work does not refer to [3] or discuss the relation between this work and [3]. Moreover, [3] particularly focuses on the homophilic case and shows advantage in the homophilic case instead of the heterophilic case, which is a mismatch between [3] and this work. I doubt the experiments in either of this work or [3]. I expect the authors to give an in-depth comparison and explain the findings.
> > >
> > > 4. Previous works discussed or considered p-Lap (1<=p<2) because the spectrum of p-Lap gives a tighter estimation of the graph conductance [4], which is a notion related to network community detection / strong homophilic network structures, while this work claims its benefit to perform node classification over heterophilic networks, which is rather counterintuitive. Can the authors give any explanations on this aspect? Thanks!
> > >
> > > [1] Chien et al., Adaptive Universal Generalized PageRank Graph Neural Network
> > > [2] Suresh et al., Breaking the Limit of Graph Neural Networks by Improving the Assortativity of Graphs with Local Mixing Patterns
> > > [3] Liu et al., Elastic Graph Neural Networks
> > > [4] Bühler & Hein, Spectral clustering based on the graph p-Laplacian
> > > [5] Klicpera et al. Predict then Propagate: Graph Neural Networks meet Personalized PageRank

---

> > > > ### Author Response · Authors · 2021-11-19
> > > > **Response to the further comments raised by the reviewer (part 1)**
> > > >
> > > > Thanks for the further comments on our manuscript! Please note that there are
> > > > still several major misunderstandings regarding this work, for which we will
> > > > provide detailed clarifications below. At the same time, we have
> > > > also significantly refined the manuscript based on the comments from all
> > > > reviewers, which is helpful for a better understanding.
> > > >
> > > > >C1: Proposition 1 is not very clear to me. I do not completely understand what low-pass and high-pass filters mathematically mean. To the best of my knowledge, this is not a commonly-used definition. We either use low-pass or high-pass instead of both. So, it causes confusion. Moreover the statement also says when p=2, it is always low-pass and high-pass, which I am very confused. Because, the case with p=2 is well known to be low-pass as it corresponds to PageRank-type diffusion and gives the model APPNP [5].
> > > >
> > > >
> > > > **Regarding the notion of "low-pass and high-pass filters":**
> > > > We thank the reviewer for pointing out the ambiguity of this statement (or this notion). We would like to clarify that we say that the filter works both as low-pass and high-pass filters if the low frequencies and high frequencies dominate the middle frequencies (the frequencies that are around the cutoff frequency). We have included the explanation as a footnote in where we first mention low-pass and high-pass filters in Section 1.
> > > > Probably a more appropriate notion for  "both as low-pass and high-pass filters" could be "low-high-pass filter".
> > > >
> > > > **Why it is both low-pass and high-pass when $p=2$:**
> > > > The case with $p = 2$ in our settings acts as both low-pass and high-pass filters, which is different from GCN, SGC, can be explained by similar spectral analysis in [6]. When $p=2$, as shown in [6], the diffusion operator is equivalent to a graph filter $(1 - \lambda)^k$ defined on the spectral domain of the Laplacian. The self-loops used by GCN, SGC, APPNP will shrink the spectral range of the Laplacian from $[0, 2]$ to approximately [0, 1.5], As a result, the amplitudes of middle and high frequencies will be small and $(1 - \lambda)^k$ are around 0 for $k > 1$. Therefore, the low frequencies will dominate the others so that GCN, SGC, APPNP act like low-pass type filters. Different from GCN, SGC, and APPNP, our diffusion operator does not use self-loops and when $p = 2$ the spectral range of the Laplacian is $[0, 2]$. Therefore, the low frequencies and the high frequencies will dominate the middle frequencies for $k > 1$ so that our diffusion operator acts as both low-pass and high-pass filters.
> > > >
> > > > Similar to the spectral analysis for $p=2$, we have shown that the $p$-Laplacian diffusion operator on node $i$ is approximatly equivalent to a graph filter $(1 - \tilde{\alpha}i\lambda i)^k$ defined on the spectral of $p$-Laplacian, where $\tilde{\alpha}_i = 1 / \sum_j\frac{W{i,j}}{D{ii}}\|(\nabla f)([i,j])\|^{p-2}$. The behaviour of the graph filter depends on the spectral range of $p$-Laplacian and the local graph gradients around $i$. Following the similar analysis to the case with $p = 2$, we conclude that $p$-Laplacian message passing can adaptively act as low-pass or both low-pass and high-pass filters on nodes in terms of the norm of their local graph gradients and the spectral range of $p$-Laplacian (which is given in Theorem 7 in Appendix C.4).
> > > >
> > > > (PS: Note that the graph gradients of nodes here are learnable in our $^p$GNN architectures because we use the first-layer to learn the node embeddings.)

---

> > > > > ### Author Response · Authors · 2021-11-19
> > > > > **Response to the further comments raised by the reviewer (part 2)**
> > > > >
> > > > > >C2: Although the experimental setting used in this paper is the same as [1], the performance listed in the table is greatly different from that in [1]. Especially for the heterophilic ones, MLP performs best over all datasets in this work. However, in [1] the non-MLP baselines work much better. Can the authors explain the reason? Also for those heterophilic networks, the network structures are not non-informative as claimed by the authors. Many of these networks essentially correspond to the node structural role classification case, where the network structures are crucial as illustrated in [2].
> > > > >
> > > > > **Performance of MLPs**: The reviewer may have overlooked some important experimental results, i.e. the results of aggregation weight entropy and the results on cSBM. MLP did not obtain the best performance on heterophilic graphs, $^p$GNNs ($p=1, 1.5$) are the best.
> > > > > The results that MLP worked well on heterophilic benchmark datasets in our settings are reasonable and can be well supported by the empirical analysis of aggregation weight entropy (as shown in Figure 3 in Section 5 or Figure 6 in Appendix F.2 for more datasets) as well as the results on cSBM (as shown in Figure 2 in Section 5 or Table 4 in Appendix F.3 for more details). Moreover, the results on cSBM have shown MLP work significantly bad (i.e. close to random guessing) on strong heterophilic graphs, where the graph topology is very helpful for label prediction.
> > > > >
> > > > > As the results of aggregation weight entropy for heterophilic benchmark datasets are shown in Figure 6 in Appendix F.2, the entropy of the aggregation weights learned by $^p$GNNs (especially for $p=1, 1.5$) is around zero, which means that most aggregation weights are on one single source node.
> > > > > MLP and $^p$GNNs obtained similar performance on these graphs and significantly outperformed the others. It indicates that most edges in these heterophilic benchmark graphs are not helpful for label prediction in our settings.
> > > > >
> > > > > Meanwhile, the results on heterophilic benchmark datasets are similar to the results on weak heterophilic graphs ($\phi = -0.25, 0$) generated by cSBM. We can observe from Table 4 in Appendix F.3 that MLP obtained superior performance on weak heterophilic graphs, where the graph topology is not much informative for label prediction. It significantly outperformed GNN baselines and is comparable with $^p$GNNs, which is similar to the results on heterophilic benchmark graphs. This again illustrates that the topology is not very helpful for label prediction for these heterophilic benchmark graphs in our settings.
> > > > >
> > > > > **On the "non-informative" statement:** Please note that we did not claim that the graph topology is non-informative for label prediction for all heterophilic graphs. The results on cSBM have shown that MLPs worked significantly bad (close to random guessing) on strong heterophilic graphs ($\phi <= -0.5$), and it did not work better than GNN baselines and was significantly dominated by $^p$GNNs and GPRGNN. This implies that the graph topology is informative for label prediction.
> > > > >
> > > > > Additionally, please note that when we say "the graph topology is not informative for label prediction", the statement is restricted to the label prediction in our settings and it does not mean the graph topology is not meaningful for a graph. It may provide information for other aspects, e.g., community detection on the graph, the node structural role classification.

---

> > > > > > ### Author Response · Authors · 2021-11-19
> > > > > > **Response to the further comments raised by the reviewer (part 3)**
> > > > > >
> > > > > > >C3: The p=1 case was also investigated in [3] but this work does not refer to [3] or discuss the relation between this work and [3]. Moreover, [3] particularly focuses on the homophilic case and shows advantage in the homophilic case instead of the heterophilic case, which is a mismatch between [3] and this work. I doubt the experiments in either of this work or [3]. I expect the authors to give an in-depth comparison and explain the findings.
> > > > > >
> > > > > > **Misunderstanding on $p=1$ case:**
> > > > > > Notably, the $p=1$ case in our work is `totally different` to the $l_1$ graph smoothing term  in [3]: when $p=1$, the $\mathcal{S}_p(f)$ in Eq. 2 becomes a $l_2$ norm, while [3] merely considered $l_1$ norm based graph smoothing. Hence this comment reflects a major misunderstanding of the technical details of related papers.
> > > > > >
> > > > > > Meanwhile, we are still happy to explain in greater detail here:
> > > > > >
> > > > > > In our work, when $p = 1$, the $\mathcal{S}_1(f) =\frac{1}{2}\sum_i\sum_j||(\nabla f)([i,j])||_2 = \frac{1}{2}\sum_i\sum_j||\sqrt{\frac{W{i,j}}{D{i,i}}}f_j - \sqrt{\frac{W{i,j}}{D{i,i}}}f_i||_2$, which is a $l_2$ norm. However, the $l_1$ norm graph smoothing term in [3] is defined to be $||\Delta f||_1 = \sum_i,_j\in_\mathcal{E}|f_i - f_j|$. They are obviously different to each other.
> > > > > >
> > > > > > **No mismatch between [3] and this work:**
> > > > > > Moreover, our experimental results did not mismatch the results in [3] because we also show the cases with $p=1$ (even though our case with $p=1$ is different from [3]) work highly effectively on homophilic benchmark datasets.
> > > > > >
> > > > > > The results of $^p$GNN for $p=1$ on both homophilic benchmark graphs (As shown in Table 3 in Appendix F.1) and strong homophilic graphs ($\phi >= 0.5$) generated by cSBM (as shown in Table 4 in Appendix F.3) can work effectively on homophilic graphs. Specifically, from Table 3 in Appendix F.1 one can observe that $^p$GNN for $(p=1)$ outperformed all baselines on Computers, Photo, CS (for CS dataset, it slightly worse than APPNP), Physics datasets, and work comparable on Cora, CiteSeer, and PubMed. From Tables 4 and 5 in Appendix F.3 one can observe that it also worked very effectively on strong homophilic graphs ($\phi \geq 0.5$) generated by cSBM.
> > > > > >
> > > > > > Besides homophilic graphs, our work demonstrates that it can also work effectively on heterophilic graphs upon our proposed message passing scheme and GNN architectures.
> > > > > >
> > > > > > We thank the reviewer for pointing out the reference [3] and we have included the discussion of [3] in the related work in Appendix A.
> > > > > >
> > > > > > >C4: Previous works discussed or considered p-Lap (1 <= p < 2) because the spectrum of p-Lap gives a tighter estimation of the graph conductance [4], which is a notion related to network community detection / strong homophilic network structures, while this work claims its benefit to perform node classification over heterophilic networks, which is rather counterintuitive. Can the authors give any explanations on this aspect? Thanks!
> > > > > >
> > > > > > **Not counterintuitive to previous work of $p$-Laplacians:** Please note that $^p$GNNs works effectively on both homophilic and heterophilic graphs for $1\leq p < 2$. We have shown that $^p$GNNs with $1 \leq p < 2$ worked highly effectively on strong homophilic graphs and obtained competitive performance against state-of-the-art GNNs in Table 3, 4, and 5 in Appendix F. As a result, the effectiveness of $^p$GNNs on homophilic graphs is exactly consistent with previous work of $p$-Laplacian.
> > > > > >
> > > > > > On the other hand, the effectiveness of $^p$GNNs on heterophilic graphs comes from their ability of adaptively learning aggregation weights, i.e. $M{i,j} = W{i,j}\|\sqrt{\frac{W{i,j}}{D{i,i}}}F{i,:} - \sqrt{\frac{W{i,j}}{D{j,j}}}F{j,:}\|^{p-2} = W{i,j}\|(\nabla f)([i,j])\|^{p-2}$. The spectral analysis given in Proposition 1 demonstrates that the behaviour of the $p$-Laplacian diffusion operator is related to $M$. The $p$-Laplacian diffusion operator can adaptively act as low-pass filters or both as low-pass and high-pass filters on nodes in terms of the graph gradients $(\nabla f)([i,j])$. Note that the graph gradients are learnable in our $^p$GNN architectures because we use the first-layer to learn the node embeddings.
> > > > > >
> > > > > > **Clarification on the relevant discussions in the old manuscript:**
> > > > > > The old version of our manuscript has heuristically discussed
> > > > > > why $^p$GNNs are effective for heterophilic graphs when $1 \leq p < 2$,
> > > > > > which might have caused ambiguity for the readers, so we carefully refined this part in the updated version.
> > > > > >
> > > > > >
> > > > > > Reference
> > > > > >
> > > > > > [1] Chien et al., Adaptive Universal Generalized PageRank Graph Neural Network
> > > > > >
> > > > > > [2] Suresh et al., Breaking the Limit of Graph Neural Networks by Improving the Assortativity of Graphs with Local Mixing Patterns
> > > > > >
> > > > > > [3] Liu et al., Elastic Graph Neural Networks
> > > > > >
> > > > > > [4] Bühler & Hein, Spectral clustering based on the graph p-Laplacian
> > > > > >
> > > > > > [5] Klicpera et al. Predict then Propagate: Graph Neural Networks meet Personalized PageRank
> > > > > >
> > > > > > [6] Simplifying graph convolutional networks. Wu et al. ICML 2019.

---

> > > > > > > ### Author Response · Authors · 2021-11-30
> > > > > > > **Thanks again for your comments!**
> > > > > > >
> > > > > > > Dear Reviewer:
> > > > > > >
> > > > > > > Thank you again for your valuable comments! We are wondering if your concerns have been addressed properly. Please let us know if you have any further questions.
> > > > > > >
> > > > > > > Best,
> > > > > > >
> > > > > > > The authors.

---

> ### Author Response · Authors · 2021-11-25
> **Reply to Reviewer Pzid**
>
> Dear Reviewer Pzid,
>
> Thank you very much for your valuable comments! We are wondering if your concerns have been addressed properly. Please let us know if you have any further questions after reviewing our answers.
>
> Best regards,
>
> The authors

---

> ### Author Response · Authors · 2021-11-29
> **Reply to Reviewer Pzid (any new concerns?)**
>
> Dear reviewer Pzid,
>
> Thank you again for your valuable comments! We want to know if your concerns have been properly addressed. If you have any other (unresolved or new) concerns after reviewing our feedback and paper, please let us know. We are very happy to answer them.
>
> Best,
>
> The authors

---

### Official Review · Reviewer_rMNu · 2021-11-06

**Correctness:** 4
**Technical Novelty And Significance:** 3
**Empirical Novelty And Significance:** 3
**Recommendation:** 8
**Confidence:** 3

**Main Review:**

-- Strengths:

1. The paper is very well-written and easy to follow. It was a pleasure to read the novel theoretical material presented in this paper.

2. The paper proposes a novel architecture of p-GNN that introduces p-Laplacian message passing by leveraging and infusing existing architectures like SGC, APPNP and GPRGCN.

3. The theoretical connection drawn between the “p-Laplacian message passing scheme” and “the polynomial graph filter defined on the spectral domain of the p-Laplacian” is particularly interesting.

-- Weaknesses:

1. It would have been interesting to see a theoretical justification as to why p-GNN always outperforms say GCNs, GATs, etc. (models that stack their message passing layers) on heterophilic graphs. Can the authors please elaborate on this point in their rebuttal? I feel this would give more insights into the behavior of p-GNN and why it is outperforming standard GNN models.

2. Much of the material in Sections 2 and 3 is obtained from Zhou and Scholkopf 2005. Less space could have been used for this explanation.

3. The function signature for the graph gradient operator in Definition 1 does not look right. The domain of this operator has to be a space of functions, it cannot be the set of vertices. Can you please relook into this?

4. In regards to the experiments conducted on noisy edges, it would be interesting to see how p-GNN performs on some stochastic block models based on edge dropping and some other random graph percolations. The random edge drop model seems a bit simplistic and it's not easy to see how this sort of noise is affecting the graph inputs and why p-GNN should do better than other robust GNNs?

5. In regards to the following statement in the paper on Pg 6.

“Therefore, for graphs whose topological information is not helpful for label prediction, we could impose more weights on the first term in Eq. (18) by using a large μ so that P-GNNs work more like MLPs which simply learn on node features. While for graphs whose topological information is helpful for label prediction, we could impose more weights on the second term by using a small μ so that P-GNNs can benefit from p-Laplacian smoothing on node features.”

How does one determine when the graph’s topological information is helpful for
label prediction, as opposed to when it’s not?


**Summary Of The Paper:**

This paper proposes a p-Laplacian based GNN to handle heterophilic graphs and graphs with non-informative topologies. Both the above cases are assumed in most existing GNN architectures and hence this work breaks away from the norm. This work proposes a discrete p-Laplacian message passing scheme which is derived from a discrete regularization framework. The authors do a spectral analysis of their novel p-Laplacian message passing scheme and show that it works as both a low-pass and high-pass filter, which is then applicable to both homophilic and heterophilic graphs. More specifically, they show that p-GNN with p>2, works well for graphs which exhibit strong homophily, while for p \in [1,2) p-GNN works effectively on heterophilic graphs. The empirical results support the theoretical justifications and outperform the baselines quite significantly especially on heterophilic and non-informative topology bearing graphs.

**Summary Of The Review:**

The paper is well written and presents a novel p-GNN architecture on top of other PageRank based GNN architectures. I found the theoretical justifications and connections drawn between the message passing scheme and the polynomial graph filter particularly interesting. Also, they had an interesting bound on the risk (Theorem 3). I am overall quite positive about this paper's acceptance, as it makes novel contributions to handle a difficult scenario like heterophilly in graphs and also the fact that they challenge the implicit assumption made by previous works that the "shape of node neighborhoods" helps distinguish nodes.. of course this can be extended to graph shapes too.

---

> ### Author Response · Authors · 2021-11-16
> **Response to all concerns raised by the reviewer**
>
> We thank the reviewer for her/his constructive evaluation and comments for improving the manuscript. We answer the questions raised below:
>
> >C1: It would have been interesting to see a theoretical justification as to why p-GNN always outperforms say GCNs, GATs, etc. (models that stack their message passing layers) on heterophilic graphs. Can the authors please elaborate on this point in their rebuttal? I feel this would give more insights into the behavior of p-GNN and why it is outperforming standard GNN models.
>
> We thank the reviewer for pointing out a very good question of under what conditions, e.g., what heterophily levels of graphs, $^p$GNNs (or more generally, GNNs that can act as high-pass filters) are guaranteed to outperform GNNs that act low-pass filters on node classification tasks.
> We believe this is an exciting direction that is worth further investigation. However, rigorous proof could be very hard, which is beyond the workload of one or two weeks. So we will leave it as important future work and continue working on it during the rebuttal period.
>
> Instead, here we give some informal explanations of why $^p$GNNs can outperform GCN, GAT on heterophilic graphs. The paper [1] demonstrates that the self-loop in the adjacency matrix will shrink its spectral range from $[0, 2]$ to approximately $[0, 1.5]$. As a result, the graph convolution of GCN in each layer acts as a low-pass filter, which limits GCN to work well on heterophilic graphs. On the contrary, $^p$GNN can work both as low-pass and high-pass filters and therefore outperforms GCN. Additionally, the attention coefficients of GAT are learned based on the similarity between linked nodes. The more similar of two linked nodes the larger the attention coefficient on their edge, which indicates GAT implicitly endows the homophily assumption. Therefore, it can not work well on heterophilic graphs as well.
>
>
> > C2: Much of the material in Sections 2 and 3 is obtained from Zhou and Scholkopf 2005. Less space could have been used for this explanation.
>
> We thank the reviewer's suggestion. We have simplified the explanations that have been discussed in previous literature. Besides, in order to avoid misunderstanding, we also add some statements to point out the difference between our definitions of $\mathcal{S}_p$ and $p$-Laplacian with the previous work in section 2.
>
>
> >C3: The function signature for the graph gradient operator in Definition 1 does not look right. The domain of this operator has to be a space of functions, it cannot be the set of vertices.
>
> We thanks the reviewer for pointing out the topos. We have corrected the topos in the definitions where the graph gradient operator should be $\Delta: \mathcal{F}_\mathcal{V} \rightarrow \mathcal{F}_\mathcal{E}$, the graph divergence operator should be $\mathrm{div}: \mathcal{F}_\mathcal{E} \rightarrow \mathcal{F}_\mathcal{V}$, and the graph $p$-Laplacian should be $\Delta_p: \mathcal{F}_\mathcal{V} \rightarrow \mathcal{F}_\mathcal{V}$, where $\mathcal{F}_\mathcal{V}$ denotes the Hilbert space of functions endowed with the inner product $\langle f, \tilde{f} \rangle := \sum_i f(i)\tilde{f}(i)$ (Note that $i \in \mathcal{V}$, there is some problems to show the full equation here. Please refer to our definition in Section 2). Similarly for defining $\mathcal{F}_\mathcal{E}$. We have updated them in the new version of our manuscript.
>
>
> >C4: In regards to the experiments conducted on noisy edges, it would be interesting to see how p-GNN performs on some stochastic block models based on edge dropping and some other random graph percolations. The random edge drop model seems a bit simplistic and it's not easy to see how this sort of noise is affecting the graph inputs and why p-GNN should do better than other robust GNNs?
>
> We thank the reviewer for her/his suggestion. We are happy to include the noisy edge experiments on graphs generated by the stochastic block models. We conjecture that $^p$GNN is more robust to the change of graph homophily levels compare to other GNN baselines. We are working on the experiments to verify the conjecture. We design the experiments by randomly adding edges to the graphs generated by the stochastic block models and randomly removing the same number of original edges. Then, we study the relationship between the gap of the graph homophily levels after adding and removing some edges and the gap of performance of GNNs. We will include the results in our manuscript once we finish the experiments.

---

> > ### Author Response · Authors · 2021-11-16
> > **Response to all concerns raised by the reviewer (part 2)**
> >
> > >C5: In regards to the following statement in the paper on Pg 6. How does one determine when the graph’s topological information is helpful for label prediction, as opposed to when it’s not?
> >
> > We say that the graph's topological information is helpful if the performance of the prediction based on only the node features (e.g., simply applying MLPs on each node) is significantly worse than the prediction based on both the node features and the graph topology (e.g., using GNNs or graph based machine learning algorithms). Otherwise, we say that the graph's topological information is not helpful. In practice, to choose a proper value of $\mu$ one may first simply apply MLPs on the node features to have a glance at the helpfulness of the node features. If MLPs work very well, there is not much space for the graph's topological information to further improve the prediction performance and we may choose a large $\mu$. Otherwise, there could be a large chance for the graph's topological information to further improve the performance and we should choose a small $\mu$.
> >
> > Please also note that the theorem of the upper-bounding risk of $^p$GNNs is moved to Appendix C1 and is referred to be Theorem 4 in our updated manuscript.
> >
> > Reference
> >
> > [1] Simplifying graph convolutional networks. Wu et al. ICML 2019.

---

> > > ### Author Response · Authors · 2021-11-27
> > > **Updating additional experimental results**
> > >
> > > >Updating experimental results for C4 raised by Reviewer rMNu
> > >
> > > We update our experimental results here to provide the reviewer with a glance at the performance of $^p$GNNs on cSBM graphs with noisy edges. We design the experiments by randomly adding edges to the graphs generated by the stochastic block models and randomly removing the same number of original edges. From the results, we can observe that adding random edge into the graphs will change their homophily levels and $^p$GNNs are better than other methods when there are a few noisy edges in graphs ($r \leq 0.5$). We are aware that it will be better to use some stochastic block models based on edge dropping and some other random graph percolations to study the behaviors of $^p$GNNs on noisy graphs. Due to the time limitation, we will leave it as future work.
> > >
> > > Results on cSBM (sparse splitting) with random edges. Averaged accuracy (\%) for 20 runs.
> > > Best results are outlined in bold and the results within $95\%$ confidence interval of the best results are outlined in underlined bold.
> > >
> > > |noise level ||$r$=0.25||||$r$=0.5||||$r$=1||
> > > |-|-|-|-|-|-|-|-|-|-|-|-|
> > > ||$\phi$=-1|$\phi$=0|$\phi$=1||$\phi$=-1|$\phi$=0|$\phi$=1||$\phi$=-1|$\phi$=0|$\phi$=1|
> > > |Homophily| $\mathcal{H}_{\mathcal{G}}$ = 0.1506 | $\mathcal{H}_{\mathcal{G}}$ = 0.4944 | $\mathcal{H}_{\mathcal{G}}$ = 0.8500 || $\mathcal{H}_{\mathcal{G}}$ = 0.2486 | $\mathcal{H}_{\mathcal{G}}$ = 0.4993 | $\mathcal{H}_{\mathcal{G}}$ = 0.7519 || $\mathcal{H}_{\mathcal{G}}$ = 0.5052 | $\mathcal{H}_{\mathcal{G}}$ = 0.4963 | $\mathcal{H}_{\mathcal{G}}$ = 0.4981 |
> > > | MLP | $49.49_{\pm 0.31}$ | $\mathbf{\underline{61.76}}_{\pm 0.20}$ | $50.06_{\pm 0.53}$ || $49.60_{\pm 0.34}$ | $\mathbf{\underline{61.74}}_{\pm 0.27}$ | $50.31_{\pm 0.63}$ || $\mathbf{\underline{49.52}}_{\pm 0.25}$ | $\mathbf{\underline{61.86}}_{\pm 0.46}$ | $50.06_{\pm 0.44}$ |
> > > | GCN | $51.07_{\pm 0.75}$ | $52.60_{\pm 0.28}$ | $66.70_{\pm 1.08}$ || $49.83_{\pm 0.36}$ | $50.07_{\pm 0.77}$ | $59.66_{\pm 1.56}$ || $\mathbf{\underline{49.61}}_{\pm 0.72}$ | $49.98_{\pm 0.64}$ | $49.57_{\pm 0.37}$ |
> > > | SGC | $50.95_{\pm 0.12}$ | $52.72_{\pm 0.06}$ | $64.36_{\pm 1.17}$ || $49.82_{\pm 0.04}$ | $49.91_{\pm 0.05}$ | $60.16_{\pm 0.75}$ || $\mathbf{\underline{49.51}}_{\pm 0.69}$ | $49.49_{\pm 0.20}$ | $49.93_{\pm 0.15}$ |
> > > | GAT | $51.25_{\pm 1.45}$ | $51.54_{\pm 1.28}$ | $66.41_{\pm 1.81}$ || $49.58_{\pm 0.36}$ | $50.43_{\pm 0.60}$ | $57.38_{\pm 3.13}$ || $48.77_{\pm 0.56}$ | $49.97_{\pm 0.36}$ | $49.71_{\pm 0.22}$ |
> > > | JKNet | $50.56_{\pm 0.78}$ | $50.95_{\pm 1.03}$ | $74.86_{\pm 2.85}$ || $49.66_{\pm 0.45}$ | $50.55_{\pm 1.07}$ | $61.77_{\pm 2.59}$ || $48.73_{\pm 0.43}$ | $50.43_{\pm 0.55}$ | $49.56_{\pm 0.24}$ |
> > > | APPNP | $49.52_{\pm 0.32}$ | $61.57_{\pm 0.48}$ | $61.58_{\pm 1.94}$ || $49.31_{\pm 0.31}$ | $61.62_{\pm 0.57}$ | $54.83_{\pm 2.66}$ || $\mathbf{49.67}_{\pm 0.51}$ | $61.54_{\pm 0.45}$ | $49.70_{\pm 0.15}$ |
> > > | GPRGNN | $79.82_{\pm 0.80}$ | $59.91_{\pm 1.84}$ | $76.02_{\pm 4.70}$ || $59.17_{\pm 4.60}$ | $59.85_{\pm 1.46}$ | $62.56_{\pm 0.85}$ || $\mathbf{\underline{49.49}}_{\pm 0.67}$ | $59.97_{\pm 0.98}$ | $49.89_{\pm 0.49}$ |
> > > | $^{1.0}$GNN | $60.83_{\pm 3.23}$ | $60.89_{\pm 0.32}$ | $\mathbf{87.56}_{\pm 1.91}$ || $50.93_{\pm 0.68}$ | $\mathbf{\underline{61.77}}_{\pm 0.38}$ | $56.45_{\pm 3.72}$ || $\mathbf{\underline{49.54}}_{\pm 0.56}$ | $\mathbf{\underline{61.79}}_{\pm 0.24}$ | $49.91_{\pm 0.27}$ |
> > > | $^{1.5}$GNN | $69.25_{\pm 2.89}$ | $61.66_{\pm 0.29}$ | $\mathbf{\underline{87.51}}_{\pm 1.12}$ || $52.98_{\pm 2.01}$ | $\mathbf{\underline{61.73}}_{\pm 0.30}$ | $\mathbf{63.99}_{\pm 1.38}$ || $\mathbf{\underline{49.54}}_{\pm 0.53}$ | $\mathbf{61.90}_{\pm 0.32}$ | $49.93_{\pm 0.23}$ |
> > > | $^{2.0}$GNN | $\mathbf{83.85}_{\pm 1.52}$ | $\mathbf{61.81}_{\pm 0.35}$ | $86.67_{\pm 0.81}$ || $56.18_{\pm 1.80}$ | $\mathbf{\underline{61.78}}_{\pm 0.34}$ | $60.35_{\pm 1.45}$ || $\mathbf{\underline{49.45}}_{\pm 0.48}$ | $\mathbf{\underline{61.73}}_{\pm 0.31}$ | $50.10_{\pm 0.42}$ |
> > > | $^{2.5}$GNN | $81.49_{\pm 1.09}$ | $61.54_{\pm 0.41}$ | $78.97_{\pm 1.85}$ || $\mathbf{61.10}_{\pm 2.00}$ | $\mathbf{61.91}_{\pm 0.24}$ | $52.59_{\pm 3.50}$ || $\mathbf{\underline{49.44}}_{\pm 0.30}$ | $61.61_{\pm 0.30}$ | $\mathbf{50.26}_{\pm 0.49}$ |

---

### Public Comment · ~Benedek_Andras_Rozemberczki1 · 2021-11-14
**The Chameleon and Squirrel Datasets were introduced in this paper - NOT the Geom-GCN paper.**

This is the right reference:

@article{rozemberczki2021multi,
  title={Multi-scale attributed node embedding},
  author={Rozemberczki, Benedek and Allen, Carl and Sarkar, Rik},
  journal={Journal of Complex Networks},
  volume={9},
  number={2},
  pages={cnab014},
  year={2021},
  publisher={Oxford University Press}
}

---

> ### Author Response · Authors · 2021-11-16
> **Thank you for pointing out the right reference for Chameleon and Squirrel datasets**
>
> The reference has been added to our manuscript.

---

### Author Response · Authors · 2021-11-16
**Common replies to all reviewers**

We thank all the reviewers for their constructive suggestions on improving our manuscript. We have carefully incorporated all the feedback and made significant changes in the updated manuscript. Specifically, we have updated some theoretical results and added more clarifications and new experiments to address reviewers' concerns. Detail changes are are summarized as follows:

1. Add a discussion on $p = 1$ (Remark 1) in Section 3 (to address the concerns raised by Reviewer 2).

1. Update Theorem 2 (Shrinking Property of $p$-Laplacian Message Passing) to prove the convergence of our iteration algorithm and remove the old explanations of Theorem 2 (to address the concerns raised by Reviewer 3).

1. Move the upper-bounding risk theorem to the Appendix to highlight our main focus on the proposed new methodology and spectral analysis (to address the concerns raised by Reviewer 3).

1. Update the upper-bounding risk theorem for all $K\geq 1$ (to address the concerns raised by Reviewer 2).

1. Add experimental results on heterophilic and homophilic benchmark datasets for 64 hidden units in Appendix (to address the concerns raised by Reviewer 3).

1. Add some visualization results of node embeddings using tSNE and the training curve for $p=1$ (to address the concerns raised by Reviewer 2).

1. Improve the clearity of $\mathbf{M}$ in Theorem 2, Section 4, and Proposition 1 (to address the concerns raised by Reviewer 3).

1. Correct some function signatures for the definition of graph gradient (to address the concerns raised by Reviewer 1).

1. Add a reference to Theorem 1 (to address the concerns raised by Reviewer 3).

1. Add a footnote of the explanation of low-pass and high-pass filters on Page 2 (to address the concerns raised by Reviewer 2).

1. Correct some typos and add some related references.

---

### Author Response · Authors · 2021-11-22
**Main changes marked**

Dear Reviewers,

Thanks a lot for your valuable comments that help us to improve our work! We are wondering whether your concerns have been addressed properly. We would be glad to answer any further questions you may have after reviewing the answers.

We have marked the main changes in the updated manuscript
in blue color, in order to better illustrate the refinement. They are mainly in Sections 2, 3, 4, Appendices C.1, F9, F10, F11.


Best regards,

The authors

---

### Decision · Program_Chairs · 2022-01-20

**Decision:**

Reject

**Comment:**

This work studies a variant of a message-passing scheme,  aiming to improve the efficiency of GNNs to heterophilic graphs, as well as improving its stability to noise. The authors provide a new architecture, called $p$-Laplacian message passing, as well as some theoretical analysis and empirical evaluation.
Reviewers highlighted several positive aspects on this work, such as the general idea of considering p-Laplacians, as well as the extensive empirical evaluation. However, during the review discussions, several important issues arose, namely important concerns regarding the theoretical contributions, as well as concerns in calibrating the baselines in some empirical evaluations. Overall, the AC is of the opinion that this paper requires a further iteration before it can be considered for publication, and encourages the authors to take the time to address the comments raised by the reviewers.